# Tracing water masses with $^{129}$I and $^{236}$U in the subpolar North Atlantic along the GEOTRACES GA01 section

Maxi Castrillejo[1,2], Núria Casacuberta[1,3], Marcus Christl[1], Christof Vockenhuber[1], Hans-Arno Synal[1], Maribel I. García-Ibáñez[4,5], Pascale Lherminier[6], Géraldine Sarthou[7], Jordi Garcia-Orellana[2,8] and Pere Masqué[2,8,9]

[1]Laboratory of Ion Beam Physics, ETH-Zurich, Otto Stern Weg 5, Zurich, 8093, Switzerland

[2]Institut de Ciència i Tecnologia Ambientals, Universitat Autònoma de Barcelona, Bellaterra, 08193, Spain

[3]Institute of Biogeochemistry and Pollutant Dynamics, Environmental Physics, ETH-Zurich, Universitätstrasse 16, Zurich, 8092, Switzerland

[4]Uni Research Climate, Bjerknes Centre for Climate Research, Bergen 5008, Norway

[5]Instituto de Investigaciones Marinas (IIM-CSIC), Eduardo Cabello 6, 36208 Vigo, Spain

[6]Ifremer, Univ. Brest, CNRS, IRD, Laboratoire d' Océanographie Physique et Spatiale, IUEM, Plouzané, France

[7]Laboratoire des Sciences de l'Environnement Marin (LEMAR), UMR 6539, IUEM, Technopôle Brest Iroise, 29280 Plouzané, France

[8]Departament de Fisica, Universitat Autònoma de Barcelona, Bellaterra, 08193, Spain

[9]School of Science, Centre for Marine Ecosystems Research, Edith Cowan University, Joondalup, WA 6027, Australia

*Correspondence to*: Maxi Castrillejo (maxic@phys.ethz.ch)

**Abstract.**

Pathways and time scales of water mass transport in the Subpolar North Atlantic Ocean (SPNA) have been investigated by many studies due to their importance for the Meridional Overturning Circulation and thus for the global ocean. In this sense, observational data on geochemical tracers provide complementary information to improve the current understanding of the circulation in the SPNA. To this end, we present the first simultaneous distribution of artificial $^{129}$I and $^{236}$U in 14 depth profiles and in surface waters along the GEOVIDE section covering a zonal transect through the SPNA in spring 2014. Our results show that the two tracers are distributed following the water mass structure and that their presence is largely influenced by the global fallout (GF) and liquid effluents discharged to north western European coastal waters by the Sellafield and La Hague nuclear reprocessing plants (NRPs). As a result, $^{129}$I concentrations and $^{236}$U/$^{238}$U atom ratios and $^{129}$I/$^{236}$U atom ratios display a wide range of values: (0.2 – 256) × 10$^7$ at/kg, (40 – 2350) × 10$^{-12}$ and 0.5 – 200, respectively. The signal from NRPs, that is characterized by higher $^{129}$I concentrations and $^{129}$I/$^{236}$U atom ratios compared to GF, is transported by Atlantic Waters (AWs) into the SPNA, notably by the East Greenland Current (EGC)/Labrador Current (LC) at the surface and by waters overflowing the Greenland-Scotland passage at greater depths. Nevertheless, our results show that the effluents from NRPs may also directly enter the surface eastern SPNA through the Iceland-Scotland passage or the English Channel/Irish Sea. The use of the $^{236}$U/$^{238}$U and $^{129}$I/$^{236}$U dual tracer approach further serves to discern Polar Intermediate Water (PIW) of Canadian origin from that of Atlantic origin which carries comparably higher tracer levels due to NRPs (particularly $^{129}$I). The cascading of these waters appears to modify the water mass composition in the bottom of the Irminger and Labrador Seas, which are dominated by Denmark Strait Overflow Water (DSOW). Indeed, PIW-Atlantic which has a high level of $^{129}$I compared to $^{236}$U appears to contribute to the deep Irminger Sea rising the $^{129}$I concentrations in the realm of DSOW. A similar observation can be made for $^{236}$U for PIW entering through the Canadian archipelago into the Labrador Sea. Several depth profiles also show an increase in $^{129}$I concentrations in near bottom waters in the Iceland and the West European Basins that are very likely associated to the transport of the NRP signal by the Iceland-Scotland Overflow Water (ISOW). This novel result would support current modelling studies indicating the transport of ISOW into the eastern SPNA. Finally, our tracer data from 2014 is combined with published $^{129}$I data for the deep

central Labrador Sea between 1993 and 2013. The results obtained from comparing simulated and measured $^{129}$I concentrations support the previously suggested two major transport pathways for the AWs in the SPNA, i.e. a short loop through the Nordic Seas into the SPNA and a longer loop which includes recirculation of the AWs in the Arctic Ocean before entering the western SPNA.

# 1 Introduction

The subpolar North Atlantic (SPNA) is a key region for the global ocean circulation (see all acronyms in Table 1). The North Atlantic Current (NAC) carries warm subtropical waters northwards to the SPNA, where they are transformed into cold Subpolar Mode Water (SPMW) and ultimately into Labrador Sea Water (LSW), which circulates southwards along with the overflow waters from the Nordic Seas (Figure 1). These water mass formation processes constitute the starting point of the Atlantic Meridional Overturning Circulation (AMOC). Among other studies, the repeated hydrographic cruises along the Greenland-Portugal OVIDE line shed light on the decadal variability of the AMOC (Daniault et al., 2016; Lherminier et al., 2010; Mercier et al., 2013) and its relevance to climate by, for example, controlling the ocean uptake of $CO_2$ (Pérez et al., 2013). The GEOVIDE cruise carried out in spring 2014 covered the OVIDE line, extending further to the Labrador Sea revealing an intense AMOC over a cold and fresh SPNA (Zunino et al., 2017). The observed strong AMOC was linked to an intensified poleward transport of subtropical waters, as well as, to the increased equatorward transport of Iceland-Scotland Overflow Water (ISOW), Irminger-SPMW (IrSPMW) and Polar Intermediate Water (PIW) in 2014 relative to mean 2002 - 2010 (García-Ibáñez et al., 2018).

Anthropogenic tracers provide complementary information about the above water mass circulation changes in the SPNA. For example, chlorofluorocarbons (CFCs) and sulphur hexafluoride ($SF_6$) from industrial activities, or tritium ($^3H$) from atmospheric nuclear weapon tests conducted in the 1950s and 1960s (global fallout, GF), provide information on the ventilation of the interior Atlantic Ocean (Doney and Jenkins, 1994; Sy et al., 1997; Tanhua et al., 2005). Contrary to CFCs or $^3H$, which were introduced into the surface ocean from a rather well mixed atmosphere, nuclear reprocessing plants (NRPs) represent point-like sources of artificial radionuclides. The NRPs located near La Hague and Sellafield discharge(d) liquid effluents to the English Channel and the Irish Sea, respectively, over the past $50 - 60$ years, thereby tagging Atlantic Waters (AWs) passing by these locations from the 1960s on (Kershaw and Baxter, 1995). This allowed investigating AW spreading pathways and time scales downstream of these nuclear facilities (e.g. Aarkrog et al., 1983, 1987, Alfimov et al., 2004, 2013; Beasley et al., 1998; Casacuberta et al., 2016,

2018; Christl et al., 2015b; Dahlgaard, 1995; Edmonds et al., 2001; Holm et al., 1983; Smith et al., 1998, 2005, 2011, 2016). The schematic transport of NRP effluents and water masses in the SPNA-Artic Ocean region is displayed in Figure 1. NRP-labelled AWs are first transported by surface currents into the North Sea and then carried poleward by the Norwegian Coastal Current (NCC) into the Nordic Seas (Edmonds

et al., 1998; Raisbeck and Yiou, 2002) while mixing with the Norwegian Atlantic Current (NwAC) (Gascard et al., 2004; Kershaw and Baxter, 1995). The current splits in two branches north of Norway, one branch entering the Barents Sea as Barents Sea Branch Water (BSBW) and the other branch approaching the Fram Strait west of Spitsbergen where it bifurcates again. One branch joins the East Greenland Current (EGC) and recirculates southwards as Return Atlantic Water (RAW) (Fogelqvist et

al., 2003) mixing with IrSPMW and PIW (modified AW that has recirculated in the Arctic Ocean; Rudels et al., 1999b). The other branch, the West Spitsbergen Current (WSC), transports the remaining AWs at shallow to intermediate depths into the Arctic Ocean via the Fram Strait Branch Water (FSBW), where they recirculate in the Arctic Eurasian Basin before outflowing back through the Fram Strait and continuing southwards carried by the EGC (Rudels, 2015). The NRP signal also penetrates deep in the

water column due to the formation of dense water north of the Greenland-Iceland and Iceland-Scotland passages, providing means of tracing the deep overflows that ventilate the deep North Atlantic Ocean (e.g. Smith et al., 2005). Thus, radionuclides discharged from European NRPs are particularly well suited for studying the water mass circulation in the SPNA (Figure 1).

Among the set of radionuclides discharged from NRPs, the [129]I is regarded as a robust circulation tracer for investigating water mass transport pathways, advection and mixing, and for testing the performance of ocean circulation models in the Nordic Seas, the Arctic Ocean and the Atlantic Ocean (Karcher et al., 2012; Orre et al., 2010; Smith et al., 2016). Firstly, [129]I can be detected at all oceanic levels far away from the source thanks to its conservative behaviour in seawater, its long half-life ($T_{1/2} = 15.7$ Ma) and the low

detection limits obtained with accelerator mass spectrometry (AMS) (e.g. Vockenhuber et al., 2015). Secondly, the presence of [129]I in those regions is dominated by the liquid discharge from European NRPs, which has a well-documented release history (> 5700 kg; He et al., 2013a; Raisbeck et al., 1995), while the contribution from GF is comparably negligible (~ 90 kg worldwide release; Hou, 2004; Raisbeck and

Yiou, 1999; Wagner et al., 1996). Consequently, the seawater affected by NRPs may present $^{129}I$ concentrations $1 - 4$ orders of magnitude above the background due to GF ($\sim 2.5 \times 10^7$ at/kg; Edmonds et al., 1998). In the western SPNA, most of the $^{129}I$ is nowadays present in the EGC flowing on the Greenland Shelf (Alfimov et al., 2004) and in the Denmark Strait Overflow Water (DSOW) that fills

bottom depths of the Irminger and Labrador Seas (Smith et al., 2005, 2016). The other overflow water, ISOW, enters the North Atlantic through the Iceland-Scotland Sill (Hansen and Osterhus, 2000) and is present notably along the Reykjanes Ridge or overlying DSOW in the Irminger and Labrador Seas. ISOW carries comparably less $^{129}I$ (Edmonds et al., 2001), yet its $^{129}I$ concentrations at the sill have increased 900 % over the last 20 years (Alfimov et al., 2013; Edmonds et al., 2001; Vivo-Vilches et al., 2018)

implying that this tracer may also provide a chronological marker for spreading pathways of such overflow water in the eastern SPNA.

Uranium-236 ($T_{1/2} = 23.5$ Ma) is a long-lived conservative radionuclide similar to $^{129}I$ and a novel ocean circulation tracer investigated in the last decade (e.g. Casacuberta et al., 2014, 2016, 2018; Castrillejo et

al., 2017; Christl et al., 2012; Sakaguchi et al., 2012; Winkler et al., 2012). Surface seawaters in the northern hemisphere present $^{236}U/^{238}U$ atom ratios of about $1000 \times 10^{-12}$ (e.g. Christl et al., 2012) in the unique presence of GF (about 900 kg released worldwide; Sakaguchi et al., 2009). However, the $^{236}U/^{238}U$ ratios can be significantly higher in the Arctic and North Atlantic Oceans due to the liquid discharge of $^{236}U$ from European NRPs (about 100 kg, Christl et al., 2015a). This made possible tracing the waters

carrying NRP-$^{236}U$ with $^{236}U/^{238}U$ ratios up to $3800 \times 10^{-12}$ in the Arctic Ocean in $2011 - 2012$ (Casacuberta et al., 2016), and up to $1400 \times 10^{-12}$ in LSW and DSOW in the western SPNA in 2010 (Casacuberta et al., 2014). In addition, both $^{236}U$ and $^{129}I$ can be combined as the dual tracer, $^{129}I/^{236}U$ - $^{236}U/^{238}U$, to identify the radionuclide source(s) present in a given water mass (Casacuberta et al., 2016; Christl et al., 2015b). This is possible because the GF and the European NRPs introduced different

amounts of $^{236}U$ and $^{129}I$ (see above) into the environment and tagged the waters with characteristic $^{129}I/^{236}U$ and $^{236}U/^{238}U$ atom ratios depending on the proximity from the source(s). Consequently, this dual tracer could also help to better understand the mixing in the SPNA between AWs tagged by European NRPs and water masses carrying mainly GF (e.g. Arctic-Canadian water).

In this study, we aim at using artificial $^{129}$I and $^{236}$U to investigate the transport pathways and time scales of water mass circulation in the SPNA. To this end, we present the first simultaneous distribution of $^{129}$I and $^{236}$U along the GEOVIDE cruise track in spring 2014 (Figure 1). The study pursues three specific

objectives. Firstly, we study the zonal distribution of $^{129}$I and $^{236}$U and their relationship with the water mass structure. Although the distribution of $^{129}$I in the Irminger and Labrador Seas has been well studied in the last 30 years, there is a significant data gap east of the Reykjanes Ridge for $^{129}$I, and for most of the section for $^{236}$U. Secondly, we use the dual $^{129}$I/$^{236}$U - $^{236}$U/$^{238}$U tracer approach to distinguish the sources contribution to the presence of $^{129}$I and $^{236}$U in the SPNA. This information is then valuable to study the

origin, mixing and spreading pathways of water masses participating in the AMOC. The combined use of $^{129}$I and $^{236}$U allows tracing circulation features that received significant attention in earlier modelling, tracer and physical studies, and helps to validate recent interpretations on the ventilation of the North Atlantic by overflow waters. Thirdly, tracer data from 2014 are combined with the extensive $^{129}$I time series in the central Labrador Sea to further investigate the circulation time scales of AWs downstream

of European NRPs.

## 2 Materials and methods

### 2.1 The cruise, study area and sample collection

The GEOVIDE cruise (Figure 1) covered the OVIDE line from Lisbon (Portugal) to the southern tip of Greenland (Cape Farewell), and from Cape Farewell to St. John's (Newfoundland, Canada) onboard the

French *R/V Pourquoi pas?* between May 15$^{th}$ and June 30$^{th}$, 2014. This cruise is part of the GEOTRACES program (section GA01: http://www.geotraces.org/cruises/cruise-summary) and contributes from a geochemical perspective to the decade-long biannual sampling (2002 to 2018) of the OVIDE line (http://www.umr-lops.fr/Projets/Projets-actifs/OVIDE).

This study is based on concentrations of $^{129}$I and $^{236}$U, and on $^{236}$U/$^{238}$U ratios determined in about 150 seawater samples collected from 14 depth profiles (Figure 1) using a rosette equipped with conductivity-

temperature-depth sensors and 24 Niskin bottles of 12 L each. Sampling depths were chosen to collect water from the main water masses and circulation features at each station by considering conductivity, temperature and oxygen profiles. From east to west (Figure 1), depth profiles were located in the West European Basin (WEB: stations 1, 13, 21 and 26), the central Icelandic Basin (station 32), above the

Reykjanes Ridge (station 38), the Irminger Sea (stations 44 and 60), the Labrador Sea (stations 64, 69 and 77), and on the shelf/slope of Greenland (stations 53 and 61) and Canada (station 78). Additional surface samples were obtained using a 'FISH' device that allowed the collection of seawater from about 2 m depth at 8 locations placed between depth profiles. Samples for $^{129}I$ of ~ 0.5 L were collected in dark plastic bottles and sealed with parafilm. The $^{236}U$ samples of $5 - 7$ L were collected in plastic cubitainers.

Bottles and cubitainers were rinsed 3 times with seawater before sample collection to avoid potential contamination.

**2.2 Iodine-129 purification and AMS measurement**

The radiochemistry of $^{129}I$ was done following a method described in Michel et al. (2012) at EAWAG

(Switzerland). About $300 - 450$ mL of sample was spiked with ~ 1.5 mg of Woodward stable iodine ($^{127}I$) carrier. All iodine was oxidized to iodate adding 2 % Ca(ClO)$_2$, then reduced to iodide using Na$_2$S$_2$O$_5$ and 1 M NH$_3$O·HCl. The purification of iodine was carried out using columns filled with DOWEX® 1 × 8 ion exchange resin. The column was conditioned with deionized water and diluted KNO$_3$ solution before the elution of iodine with 2.25 M KNO$_3$ solution. The iodine was precipitated as AgI by adding AgNO$_3$,

then mixed with 4 - 5 mg of Ag and pressed into AMS cathodes. The compact 0.5 MV Tandy AMS system at ETH-Zurich was used to measure the $^{129}I/^{127}I$ atom ratios (Vockenhuber et al., 2015). The $^{129}I/^{127}I$ ratios were normalized with the ETH-Zurich in-house standard D22 with a nominal $^{129}I/^{127}I$ value of $(50.35 \pm 0.16) \times 10^{-12}$ (Christl et al., 2013b). Radiochemistry blanks (n = 24) were prepared with deionized water and processed together with seawater samples following the same analytical procedures.

These blanks presented $^{129}I/^{127}I$ ratios of $(0.7 - 4) \times 10^{-13}$ corresponding to $(0.5 - 3) \times 10^6$ atoms of $^{129}I$. The $^{129}I$ concentrations were calculated based on measured $^{129}I/^{127}I$ ratio and the know amounts of $^{127}I$

carrier added to each sample. The detection limit of $< 0.3$ fg $^{129}I$ depended on the measured $^{129}I/^{127}I$ ratio of the Woodward iodine carrier which is typically at the order of $\sim 10^{-13}$.

## 2.3 Uranium-236 purification and AMS measurement

Each seawater sample (5 – 7 L) was weighed, acidified to pH below 2 using concentrated suprapure $HNO_3$ and spiked with $\sim 3$ pg of $^{233}U$ (IRMM - 051). The uranium was co-precipitated with iron hydroxides upon addition of $\sim 200$ mg of U-free $Fe^{2+}$ solution and concentrated suprapure $NH_4OH$ by rising the pH to $\sim 8$. The iron precipitate was syphoned, evaporated to dryness and re-dissolved using 8 M $HNO_3$. The purification of uranium was carried out using UTEVA columns (Triskem). The eluate was co-precipitated

with $\sim 1$ mg of the U-free $Fe^{2+}$ solution and evaporated to dryness. All uranium was converted to oxide form by heating the iron precipitates to 650 ºC, then mixed with 2 – 3 mg of niobium and pressed into AMS cathodes. The compact 0.5 MV Tandy AMS system at ETH-Zurich was used to measure $^{233}U$, $^{236}U$ and $^{238}U$ following Christl et al. (2013a). The measured $^{236}U/^{233}U$ and $^{236}U/^{238}U$ ratios were normalized to the ZUTRI ETH-Zurich in-house standard with nominal values of $(4055 \pm 200) \times 10^{-12}$ and $(33170 \pm$

$830) \times 10^{-12}$, respectively (Christl et al., 2013a). Radiochemistry blanks (n = 19) were prepared onboard and in the land-based laboratory with deionized water and processed together with seawater samples following same analytical procedures. The blanks presented $^{236}U/^{233}U$ ratios $< 10^{-4}$, corresponding to $<$ 40 ag of $^{236}U$. The compact TANDY AMS system has an abundance mass sensitivity of $\sim 10^{-12}$ for the mass range of actinides, corresponding to an estimated instrumental background level at the order of

$^{236}U/^{238}U \sim 10^{-14}$. Due to a mistake in the laboratory, a total of 34 samples were accidentally cross-contaminated with a very high $^{236}U$ standard and therefore had to be background corrected for cross talk. The uncertainty of this additional background correction led to higher errors reported for those samples (marked with '*' in Table 2).

# 3 Results and discussion

## 3.1 Water mass structure in 2014

The water mass structure in spring 2014 is described using the zonal sections for salinity, potential temperature and dissolved oxygen concentrations (Figure 2) to facilitate the understanding of [129]I and [236]U distributions in section 3.2. The assessment of the water mass structure from GEOVIDE was performed using an extended Optimum Multi-Parameter analysis (OMP) (further details are found in García-Ibáñez et al. 2018).

In the upper water column (< 500 m), warm and saline Central Waters dominate the eastern part of the section between the coast off Portugal and station 26 (Figures 2A and 2B). Central waters (or East North Atlantic Central Water, ENACW) are characterized by the highest potential temperatures and relatively high salinities as they have been transported from subtropical latitudes into the eastern part of the section by the northeast-flowing branches of the NAC. Part of ENACW recirculates into the Iceland Basin and the Irminger Sea, where air-sea fluxes transform them into colder and fresher SPMWs (McCartney and Talley, 1982) that occupy equivalent depths between the Subarctic Front (SAF; roughly at 22.5 °W, station 26) and Greenland (Figure 2A and 2B). The upper water column on the continental shelves and slopes of Greenland and Canada is occupied by PIW, which presents very low salinities (< 34) and potential temperatures usually < 0 °C (Figures 2A and 2B). PIW originates from the Arctic Ocean, enters through the Fram (PIW-Atlantic) and Nares Straits (PIW-Canada), and joins the shallow western boundary transport in the EGC and LC (Figure 1).

At intermediate depths, LSW is the most abundant water mass and fills the entire section from the upper water-levels to about 2000 m depth (Figure 2). LSW is formed in the Labrador Sea and, to less extent, in the Irminger Sea by transformation of SPMWs via winter convection (e.g., Jong and Steur, 2016). Then, it flows south as part of the DBWC (e.g. Bersch et al., 2007) or east into the Irminger, Iceland and West European Basins (Figure 1). LSW is characterized by a relative minimum in salinity (< 34.9) and temperature (~ 3 °C) at its formation region, and warmer and saltier values as it mixes with surrounding waters along its equatorward and eastward transport (Figures 2A and 2B). Station 26 is also affected by

the Subarctic Intermediate Water (SAIW), which presents lower salinities (~ 34.9) and potential temperatures (4 – 7 ºC) than the other water masses surrounding it at similar depths, i.e. ENACW and SPMW from the Iceland Basin (IcSPMW), respectively (Figures 2A and 2B). SAIW forms in the western boundary of the SPNA (i.e. the LC) by mixing between LSW and subtropical waters carried by the NAC

(Arhan, 1990; Read, 2000), before subducting at about 400 m depth and being advected within the northern branch of NAC (Figure 2). Depths around 1000 m in the WEB (stations 1 and 13) are also influenced by the northward-flowing Mediterranean Water (MW) (Figure 1) which is characterized by a maximum in salinity (> 36) and minimum in oxygen (~ 180 µmol/kg) (Figures 2A and 2C).

In the deep-water column (> 2000 m), the lower North East Atlantic Deep Water ($NEADW_L$) dominates the section in the WEB (east of 20 ºW), while in the western part, the most abundant water masses are the dense overflow waters. $NEADW_L$ is generally saltier, colder and older than the overlying LSW due to the major contribution of the northward flowing Antarctic Bottom Water (AABW) (Figures 2A – 2C). Dense overflow waters dominate the bottom depths on both sides of the Reykjanes Ridge and in the Irminger

and Labrador Seas (Figure 1). ISOW is best identified thanks to its local salinity maximum (~ 34.92) on the flanks of the Reykjanes Ridge and between the LSW and DSOW in the Irminger and Labrador Seas (Figure 2A). This water mass is produced by mixing of old Norwegian Sea waters that overflow the Iceland–Scotland Sill and entrain SPMW and LSW in the SPNA (e.g., van Aken and De Boer, 1995). ISOW mainly flows along the eastern flank of the Reykjanes Ridge into the Irminger and Labrador Seas

(Figure 1), yet increasing studies point towards the eastward return flow of this water mass through passages in the Mid Atlantic Ridge (Xu et al., 2018) or directly along the flanks of the Rockall Through into the eastern part of the GEOVIDE section (e.g. Zou et al., 2017) (Figure 1). Finally, DSOW is present between ISOW and the seafloor in the Irminger and Labrador Seas (notably stations 44 and 69). This overflow water can be distinguished from ISOW by its lower salinities (< 34.90), potential temperatures

(< 2 ºC) and higher oxygen concentration (> 290 µmol/kg) (Figures 2A – 2C) due to the more recent ventilation and rapid advective flow of DSOW from the formation region north of the Denmark Strait into the GEOVIDE section (e.g. Read, 2000) (Figure 1).

## 3.2 The relationship of $^{129}$I and $^{236}$U with water masses in 2014

The concentrations of $^{129}$I and $^{236}$U, and atom ratios of $^{236}$U/$^{238}$U and $^{129}$I/$^{236}$U are reported in Table 2. Detailed depth profiles for these radionuclides are displayed along with salinity, potential temperature and dissolved oxygen concentrations in the Supporting Information (Figure S1). The $^{129}$I concentrations range from $(0.20 \pm 0.20) \times 10^7$ to $(256 \pm 4) \times 10^7$ at/kg. The $^{236}$U/$^{238}$U ratios range from $(40 \pm 20) \times 10^{-12}$ to $(2350 \pm 370) \times 10^{-12}$. The $^{129}$I/$^{236}$U ratios range from $0.50 \pm 0.50$ to $200 \pm 60$.

The results from this study are best described in relation to the water mass structure described in section 3.1, which has been represented with overlaid isohalines on the radionuclide distribution plots (Figure 3). The $^{129}$I concentrations (Figure 3A) and $^{236}$U/$^{238}$U ratios (Figure 3B) are generally higher west of $\sim 25\,°$W in southward flowing waters than in northward flowing low-latitude waters dominating the eastern SPNA. This is also clear when comparing tracer values of stations 1 - 26 with those from stations 32 - 78 (Table 2). Such general radionuclide distribution is largely due to the fact that southward flowing northern waters are located downstream of Sellafield and La Hague, and therefore present additional radionuclide contributions (especially for $^{129}$I) from these facilities. The highest $^{129}$I concentrations ($\sim 250 \times 10^7$ at/kg) and $^{236}$U/$^{238}$U ratios ($\sim 2300 \times 10^{-12}$) are present in PIW and RAW carried by the EGC and LC over the shelves and slopes of Greenland and Canada. This water admixture, largely influenced by NRPs (e.g. Alfimov, 2004), can mix with DSOW precursor waters through winter convection in the Greenland Sea (e.g. Gascard et al., 2002) or directly intrude DSOW by cascading in the Labrador Sea (e.g. Falina et al., 2012). Consequently, the DSOW core, found at the lowermost 100 m in the Irminger and Labrador Seas, presents $^{129}$I concentrations in the $(85 - 100) \times 10^7$ at/kg range and $^{236}$U/$^{238}$U ratios of $(1300 - 2300) \times 10^{-12}$, which is in agreement with high radionuclide levels previously reported for DSOW (Casacuberta et al., 2014; Orre et al., 2010; Smith et al., 2005, 2016). Intermediate $^{129}$I concentrations $((5 - 50) \times 10^7$ at/kg) and $^{236}$U/$^{238}$U ratios $((500 - 1500) \times 10^{-12})$ characterize the water masses filling most of the remaining GEOVIDE section. In the upper 500 m, ENACWs record rather uniform $^{129}$I concentrations ($\sim 10 \times 10^7$ at/kg) and $^{236}$U/$^{238}$U ratios ($\sim 1000 \times 10^{-12}$) in the WEB (stations 1 to 21). The $^{129}$I concentrations of SPMWs located further west almost double those of ENACW ($^{236}$U/$^{238}$U ratios are similar), indicating

that ENACW may be influenced by effluents from NRPs while recirculating in the northern SPNA or in the Nordic Seas. SAIW also presents relatively high $^{129}$I concentrations ($\sim 20 \times 10^7$ at/kg) at stations 26 and 32, probably because of the influence of waters carried by the LC. The 500 – 2000 m layer is dominated by LSW, which displays a wide range of $^{129}$I concentrations (($5-50$) $\times 10^7$ at/kg) and $^{236}$U/$^{238}$U

ratios (($700-1250$) $\times 10^{-12}$), with values decreasing downstream from its formation regions, the Labrador and Irminger Seas. Similar depths are also influenced by MW (stations 1 and 13), yet, its $^{129}$I concentrations ($\sim 3 \times 10^7$ at/kg) and $^{236}$U/$^{238}$U ratios ($\sim 1000 \times 10^{-12}$) in 2014 were significantly lower than average $^{129}$I concentrations ($9 \times 10^7$ at/kg) and $^{236}$U/$^{238}$U ratios ($1600 \times 10^{-12}$) reported in the outflow region of MW at the Strait of Gibraltar in 2013 (Castrillejo et al., 2017). Thus, 2014 data probably reflects

the dilution of MW, which is largely affected by inputs from the Marcoule nuclear facility (Castrillejo et al., 2017), with old LSW and SPMW carrying a diluted NRP signal. The deeper parts of the section west of 20 °W and below 2000 m are influenced by ISOW, which is characterized by relatively high $^{129}$I concentrations (($10-70$) $\times 10^7$ at/kg) and $^{236}$U/$^{238}$U ratios (($900-1700$) $\times 10^{-12}$). The lowest $^{129}$I concentrations (($0.2-2.0$) $\times 10^7$ at/kg) and $^{236}$U/$^{238}$U ratios (($40-350$) $\times 10^{-12}$) are found at depths greater

than 2000 m in the WEB and are associated with NEADW$_L$ (stations 1 to 13).

The distribution of $^{129}$I/$^{236}$U (Figure 3C) is notably driven by $^{129}$I concentrations, which display a greater range (3 orders of magnitude) than the $^{236}$U/$^{238}$U ratios (2 orders of magnitude). As noted before, this is probably due to the influence of NRPs, which released about 60 times more mass of $^{129}$I than of $^{236}$U to

the North Atlantic (further discussion in section 3.3). Thus, following the $^{129}$I patterns described above, the $^{129}$I/$^{236}$U ratios are particularly high (> 20) in the western part of the section and particularly contrasted in the Irminger and Labrador Seas as discussed in section 3.3 (Figure 3C). The highest $^{129}$I/$^{236}$U ratios (> 100) are present in waters transported by the shallow EGC and LC. Overflow waters are also distinguished by their relatively high $^{129}$I/$^{236}$U ratios (60 to 110 for DSOW, 15 to 40 for ISOW).

### 3.3 Sources of $^{129}I$ and $^{236}U$ in the SPNA

All samples show $^{129}I$ concentrations and $^{236}U/^{238}U$ ratios well above the lithogenic background (LB) or the natural values ($\sim 0.04 \times 10^7$ at/kg for $^{129}I$, Snyder et al., (2010); and $10^{-14}$ - $10^{-13}$ for $^{236}U/^{238}U$ atom ratios: Christl et al., (2012) and Steier et al., (2008)). This was also shown in previous studies, highlighting
the influence of artificial sources on the presence of $^{129}I$ (e.g. Edmonds et al., 2001) and $^{236}U$ (Casacuberta et al., 2014; Christl et al., 2012) in the North Atlantic.

As done in earlier studies (Casacuberta et al., 2016), we can estimate the contribution to our samples from the LB, GF and NRP by combining the $^{129}I/^{236}U$ and $^{236}U/^{238}U$ in a dual tracer approach (Figure 4). This
is possible because the atom ratios of $^{129}I/^{236}U$ and $^{236}U/^{238}U$ display a wide range of values due to the different input of $^{129}I$ and $^{236}U$ from the three sources. For example, the GF introduced about 10 times more $^{236}U$ than $^{129}I$, thus this endmember is characterized by $^{129}I/^{236}U < 1$ and $^{236}U/^{238}U$ surface ratios in the $(1000–2000) \times 10^{-12}$ range. On the contrary, the total amount of $^{236}U$ introduced from European NRPs was much smaller than for $^{129}I$. Therefore, a water mass with the additional influence of the European
NRPs may present $^{129}I/^{236}U$ on the 1–350 range and $^{236}U/^{238}U$ above the GF. The natural presence of $^{129}I$ and $^{236}U$ is negligible compared to artificial sources, yet the LB can be distinguished by a very small $^{236}U/^{238}U$ ($\sim 10^{-13}$) and a relatively large $^{129}I/^{236}U$ ($\sim 370$). The simple mixing model (Figure 4) considers the three aforementioned endmembers constant in time, for which values were estimated by Casacuberta et al. (2016) based on the literature or on their own calculations. The mixing lines between each
endmember represent all possible binary mixing scenarios, i.e. they delimit the range of $^{129}I/^{236}U$ and $^{236}U/^{238}U$ that a given water mass may show depending on the sources and on the different degrees of mixing. For instance, a sample falling in the 1 % value on the GF-NRP binary mixing line would be composed of waters carrying largely GF and about 1 % of the NRPs signal.

On top of the mixing model, we represent the results from the GEOVIDE cruise (Figure 4). Each data point represents a seawater sample collected at a certain station (Figure 4A) or assigned to a dominant water mass (Figure 4B to 4F). The results show that most of the samples fall along the GF-NRP binary mixing line with contributions from NRPs > 1 %. The largest NRP contribution, above 5 %, is observed

in the Irminger and Labrador Seas associated notably with PIW, DSOW and to lesser extent with ISOW. LSW also records NRP contributions > 1 % in the westernmost stations (e.g. 44 to 77), while LSW that has been transported further east on the section (stations 1 to 38) reflects the greater mixing or dilution with waters carrying notably GF. ENACWs are closer to the GF endmember, yet they show significant

contributions (~ 1 %) from NRP. This result is unexpected, given that the transport of ENACW occurs upstream and far away from the NRPs, from subtropical latitudes into the eastern SPNA (stations 1 to 26). At least 6 samples separate from the GF-NRP mixing line and plot towards the LB endmember. These are associated with the contribution of the northward flowing AABW to NEADW$_L$ (van Aken and Becker, 1996). AABW is the oldest water mass in the SPNA and has little or no influence from nuclear

activities given that it was not exposed to the surface or atmosphere for decades.

## 3.4 Time evolution of $^{129}$I in the SPNA

In this section we compare radionuclide concentrations reported in the literature with those measured at nearby stations during GEOVIDE (Figure 5). The assessment of the temporal evolution of radionuclide

distributions is important to identify the main circulation features highlighted by these tracers. The limited data on the novel $^{236}$U tracer prevents from studying any temporal evolution. In the case of $^{129}$I, the existing time series for the central Labrador Sea (1993–2013, Figure 5A) demonstrated that most of the tracer transport was carried by overflow waters (e.g. DSOW) and that the temporal evolution of $^{129}$I concentrations in those waters could be associated with the tracer release from the European NRPs few

20  years earlier (Edmonds et al., 2001; Orre et al., 2010; Smith et al., 2005, 2016). For instance, the literature on $^{129}$I shows a rise in tracer concentrations due to the increased $^{129}$I discharge rate from European NRPs in the whole water column, being more pronounced (about 10 times increase) in overflow waters (Figure 5A). The depth distribution of $^{129}$I concentrations in the Labrador Sea in 2014 (station 69) displays $^{129}$I concentrations in DSOW about 15 % lower (see section 3.5.4) than in 2012 – 2013 (Smith et al., 2016),

yet the general shape of the depth profile is comparable (Figure 5A). The main difference between the $^{129}$I depth profiles in the Irminger Sea (station 44, red squares in Figure 5A) and central Labrador Sea (station 69, red circles in Figure 5A) in 2014 is the surface $^{129}$I peak in the latter one. Considering the 30

m deep freshwater surface layer observed between station 69 and Greenland (not shown here), we suggest that waters carried by the West Greenland Current (continuation of the EGC) may have separated from the main western boundary transport and entered the Central Labrador Sea (Cuny and Rhines, 2002). This may also explain the peak in $^{236}U/^{238}U$ ratios observed at the same location (Figure S1).

A similar assessment of $^{129}I$ concentrations is now possible for the water column over the Reykjanes Ridge (station 38) and the Icelandic Basin (station 32) (Figure 5B), which were first studied in 1993 (Edmonds et al., 2001). The $^{129}I$ concentrations in the water column are 5 - 7 times higher in 2014 than in 1993. The most pronounced increase occurs in the upper 1000 m filled by SPMWs and in the deep

Icelandic Basin dominated by ISOW. This novel result shows that the $^{129}I$ tracer could potentially be used to trace the transformation of ENACWs into SPMWs and the evolution of ISOW. The depth profiles of $^{129}I$ concentration measured in the WEB in 2014 (particularly station 21) resemble the one sampled at the Porcupine Abyssal Plain (PAP) in 2012 by Vivo-Vilches et al. (2018) (Figure 5C). The $^{129}I$ distribution in the upper 1000 m at PAP is very similar to station 21 located 365 km to the southwest, while below

that depth $^{129}I$ concentrations are about $2.5 \times 10^7$ at/kg higher in the PAP. These results suggest a similar water mass composition for that region, yet the offset in deep $^{129}I$ concentrations would support the hypothesis that effluents from the nearby Sellafield and/or La Hague NRPs may enter directly into the SPNA without previous circulation in the Nordic Seas (see section 3.5.1).

**3.5 Tracing water mass circulation in the SPNA using $^{129}I$ and $^{236}U$**

We use the above information on the distribution, sources and time evolution of $^{129}I$ and $^{236}U$ to investigate the circulation of nuclear reprocessing effluents and in return, provide more insight on composition, spreading pathways and transport time scales of water masses in the SPNA.

### 3.5.1 Shallow water circulation in the eastern SPNA

The main transport of reprocessing effluents occurs poleward, yet the increasing observations and simulations on [129]I suggest that part of the NRP signal may enter directly the surface of the SPNA without previous circulation in the Nordic Seas. Such hypothesis is based on the fact that [129]I concentrations in surface waters of the northeastern SPNA can record values more than one order of magnitude above the GF level ($2.5 \times 10^7$ at/kg; Edmonds et al., 1998). He et al. (2013b) proposed that the outflow through the English Channel and the Irish Sea may lead to [129]I concentrations in surface waters above $20 \times 10^7$ at/kg and modify the isotopic iodine composition in the Bay of Biscay. Modelling of [129]I releases from the European NRPs also shows that tracers discharged from Sellafield may expand southwards (Villa et al., 2015), which could explain [129]I concentrations of $77 \times 10^7$ at/kg measured in surface waters of the Celtic Sea (Vivo-Vilches et al., 2018). GEOVIDE data (Table 2) of [129]I concentrations ($\sim 10 \times 10^7$ at/kg) and [129]I/[236]U ratios (7–20) in ENACWs confirms such influence from the NRPs. If that was the case, the different NAC branches could mix ENACWs with reprocessing-labeled local waters and be transported southward by surface currents (e.g. Lambelet et al., 2015; Lherminier et al., 2010; Ríos et al., 1992). Indeed, NAC branches west of station 21 recirculated anti-cyclonically into the WEB bringing waters south across the GEOVIDE section in 2014 (Zunino et al., 2017). Further, the transformation of ENACW into SPMW results in [129]I concentrations twice larger in the Icelandic Basin than in the WEB (Figures 5B and 5C), which suggests the near-surface transport of [129]I from European NRPs also occurs southward across Iceland-Scotland. Consequently, [129]I concentrations in shallow waters at stations 1, 13 and 21 strongly contrast with those at stations 26 and 32 located west of the SAF (Table 2). Such near surface tracer input would also explain the increase in surface [129]I concentrations (Figure S2) up to $10^8$–$10^9$ at/kg in the Icelandic Basin and northwest of the British Isles by 2010–2012 (Gómez-Guzmán et al., 2013; Vivo-Vilches et al., 2018). Thus, one could potentially use [129]I to trace ENACWs in the upper water column of the WEB and their transformation into SPMW. This is not clearly supported by [236]U levels ($\sim 10 \times 10^6$ at/kg) which are close to GF in the shallow eastern SPNA, yet European NRPs introduced about 60 times less [236]U than [129]I (Christl et al., 2015b).

### 3.5.2 Shallow water transport and cascading in the Irminger Sea and Labrador Sea

It is well known that the eastern coast of Greenland receives RAW and PIW-Atlantic injected in the EGC (Figure 1). Similarly, the shelf of Newfoundland is bathed by the LC, which carries EGC waters and PIW-Canada, this last one being supplied through the Nares Strait (Curry et al., 2014). The tracer levels are particularly high in such waters residing on the shelves, slopes and very deep waters around Greenland and Newfoundland (Figure 3). Further, the tracer content differs between Arctic waters of Atlantic and Canadian origin enriched in $^{129}$I and $^{236}$U, respectively. Thus, one may use them to distinguish key circulation features such as the EGC/LC and the DWBC in the Labrador and Irminger Seas (Figure 1). For example, at shallow depths, the EGC (stations 53 to 64) presents remarkably high $^{129}$I concentrations (up to $\sim 250 \times 10^7$ at/kg) and $^{129}$I/$^{236}$U ratios (up to 200), while both values are significantly lower in the LC (station 78) which is characterized by comparably higher $^{236}$U/$^{238}$U ratios (up to $2350 \times 10^{-12}$) (Figure 3). Such differences on the composition of $^{129}$I and $^{236}$U in the two shallow boundary currents are likely due to the fact that waters of Atlantic origin (PIW-Atlantic and RAW) have been largely influenced by NRP effluents (high $^{129}$I). On the contrary, the LC records lower $^{129}$I concentrations due to the influence of PIW-Canada waters with mainly GF signal (Ellis and Smith, 1999; Smith et al., 1998), and a large $^{236}$U content ($^{236}$U/$^{238}$U ratios are likely $> 2000 \times 10^{-12}$) from both the GF and unconstrained Arctic rivers inputs (Casacuberta et al., 2016).

Shelf waters carried by the EGC are thought to occasionally descend down the Greenland slope feeding the East Greenland Spill Jet and DSOW (von Appen et al., 2014; Falina et al., 2012; Harden et al., 2014; Koszalka and Haine, 2013; Pickart et al., 2005; Rudels et al., 1999a). The GEOVIDE section shows a rise of $^{129}$I concentrations at certain depths on the Greenland slope (e.g., station 60; Figure 3 and Figure S1), and particularly in bottom waters of the Irminger Sea (station 44), which are probably related to the cascading of $^{129}$I-rich waters from the Greenland Shelf. This finding would be supported by OMP analyses that estimate up to 20 % of PIW in the DSOW realm (García-Ibáñez et al., 2018). Our results also highlight that similar processes may be taking place in the Canadian shelf, but with PIW-Canada water cascading to the bottom of the Labrador Sea. This would explain the slightly higher $^{236}$U/$^{238}$U ratios near the Newfoundland Shelf (station 77; Figure 3 and Figure S1) compared to the offshore waters in the Labrador

Sea (e.g., station 69), as well as, the higher $^{236}$U/$^{238}$U ratios and lower $^{129}$I concentrations in the deep Labrador Sea (influenced by PIW-Canada) compared to the Irminger Sea (influenced by PIW-Atlantic) (Figure 3 and 5A).

### 3.5.3 Spreading pathways of ISOW in the eastern SPNA

The $^{129}$I and $^{236}$U tracers may help validate current interpretations of ISOW spreading pathways in the SPNA which are largely based on model outputs or on limited observations (Fleischmann et al., 2001; LeBel et al., 2008; Xu et al., 2010; Zou et al., 2017). The ISOW is best distinguished by its relative $^{129}$I concentration maxima and $^{129}$I/$^{236}$U ratios of 15 – 40 due to NRPs, that are significantly higher than in surrounding waters (e.g., LSW, NEADW$_L$). This is particularly visible for $^{129}$I concentrations (Figures 5B and 5C, and Table 2) in deeper parts of the Icelandic Basin (stations 32 and 38) and the WEB (stations 1 and 13), where the presence of ISOW has also been inferred from OMP analyses (García-Ibáñez et al., 2018). For $^{236}$U (Figure S1), such increase is not as pronounced as for $^{129}$I, probably due to the pre-existing $^{236}$U from the GF. In the coming years, one can expect a stronger $^{129}$I signal carried by ISOW because tracer concentrations in ISOW precursor waters have increased from $7 \times 10^7$ at/kg to $63 \times 10^7$ at/kg at the Iceland–Scotland Sill from 1993 to 2012 in response to releases from the NRPs (Alfimov et al., 2004; Edmonds et al., 2001; Vivo-Vilches et al., 2018). The overflow of ISOW through the Iceland-Scotland Sill has increasing implications for the deep ventilation of the SPNA and for the magnitude of the AMOC (García-Ibáñez et al., 2018). Thus, future time series of time-varying $^{129}$I concentrations at GEOVIDE stations and further upstream may also be used to investigate timescales of ISOW ventilation in the North Atlantic Ocean.

### 3.5.4 Transit times and dilution factors of reprocessing-labelled Atlantic Waters

The observations of $^{129}$I concentrations in DSOW filling the Central Labrador Sea between 1993 and 2013 (Figure 5A) have also been valuable to estimate transport times of Atlantic Waters carrying the NRP signal into the Arctic and the North Atlantic (Orre et al., 2010; Smith et al., 2005, 2011, 2016). The $^{129}$I

time series shows two tracer pulses, one in the early 2000s and the following about 10 years later (Figure 6A, data points). Smith et al. (2005) suggested that the first sharp increase in $^{129}$I concentrations in DSOW was related to the arrival of the tracer front observed in the late 1990s at 60 °N in the northern North Sea. The second peak in $^{129}$I concentrations has also been related to the same cause as in the late 1990s front, but in this case, it would correspond to the return flow of AWs that were transported northwards into the Arctic Ocean before returning to the western SPNA (Smith et al., 2011). According to these authors, the 10-year gap between the two tracer fronts would be related to such 'Arctic loop', i.e. the transport of AWs by the FSBW into the Arctic Eurasian Basin, the return flow along the Lomonosov Ridge and the incorporation via the EGC into DSOW.

To test this hypothesis, we took the $^{129}$I input function at 60 °N for the northern North Sea (Figure 6A, green dashed line) used in previous studies (e.g. Christl et al., 2015b; Orre et al., 2010; Smith et al., 2005). This input function is estimated assuming that the signal of both NRPs mixes in the North Sea and then is advected to 60 °N in 2 years from La Hague and in 4 years from Sellafield. Following earlier modelling studies (Smith et al., 2005), we firstly estimate a new $^{129}$I input function for DSOW in the Central Labrador Sea (Figure 6A, blue line) that should match the aforementioned first tracer front shown by $^{129}$I measurements between 1993 and 2001 (Edmonds et al., 2001; Smith et al., 2005). This is achieved by applying a time lag (6 years) that accounts for the transit time from 60 °N, and a dilution factor (DF = 50) that represents the mixing with waters carrying only GF signal. Secondly, we estimate a second $^{129}$I input function (Figure 6A, red line) that fits the $^{129}$I concentrations measured in 2012 and 2013 by Smith et al. (2016) and in 2014 by this study. We found that this is possible when the tracer input function (Figure 6A, green dashed line) is diluted 30 times and a delay-time of about 14 years is applied. Thus, 2014 data reported in this study supports the current interpretation on the 'Arctic loop' (e.g. Smith et al., 2016) and suggests that the second $^{129}$I front probably peaked before the GEOVIDE cruise. Note that the latter input function (Figure 6A, red line) only provides an upper estimate for the $^{129}$I transit times and the dilution factor because the observed $^{129}$I concentrations in 2012 – 2014 also contains water from the shorter loop (Figure 6A, blue line). According to our results (Figure 6A), AWs follow at least two paths before arriving to the Labrador Sea: i) one short path into the Nordic Sea and then to the SPNA that takes approximately

8–10 years from the NRPs, and ii) a long path which adds approximately 8 years of circulation in the Arctic Ocean resulting in 16–18 years of transit time from the NRPs to the central Labrador Sea. A similar exercise for $^{236}$U (Figure 6B) shows that the single $^{236}$U measurement available for overflow waters in the central Labrador Sea (2014, this work) is above the concentrations predicted using the two fits. Although

5    the $^{236}$U data would agree with the hypothesis of a second delayed $^{129}$I pulse arriving from the Arctic Ocean, there are significant inconsistencies between the simulated and measured concentrations. These might be attributed to, among other factors, the large uncertainty of the used $^{236}$U data point for 2014, uncertainties on the amount released by the Sellafield NRP (Christl et al., 2015), missing information on other sources, or unaccounted features on the water mass circulation downstream the NRPs.

## 4 Conclusions

The distribution of artificial $^{129}$I and $^{236}$U in the SPNA was governed by the main water mass circulation. The highest $^{129}$I concentrations and $^{236}$U/$^{238}$U ratios are associated with water masses originating from the Nordic Seas (DSOW, ISOW and surface currents) or the Arctic Ocean (PIW). On the other end, ENACW

and NEADW$_L$ transported from low latitudes north into the SPNA present $^{129}$I concentrations and $^{236}$U/$^{238}$U ratios of about $2-3$ orders of magnitude lower. The $^{236}$U/$^{238}$U - $^{129}$I/$^{236}$U dual tracer approach indicates that all water masses, except NEADW$_L$, are influenced by GF and NRPs. ENACW is also influenced by effluents from NRPs (e.g. $^{129}$I/$^{236}$U > 1), which suggests that part of the radioactive releases split off from the mainstream and enter the surface eastern SPNA either through direct exchange at the

English Channel/Irish Sea or at the passage between Iceland and Scotland. Other key circulation features such as the shallow transport of PIW and RAW by the EGC and LC, or the deep North Atlantic ventilation by overflow waters (DSOW, ISOW) are particularly visible due to the presence of reprocessing $^{129}$I and $^{236}$U. For example, ISOW is tagged with relatively high $^{129}$I and, therefore, it can be traced while spreading eastwards into the WEB among waters that have lower tracer amounts. The combined use of $^{236}$U/$^{238}$U -

$^{129}$I/$^{236}$U allows differentiating water mass composition and origin, and serves to confirm known circulation features and validating recent interpretations on water mass transport pathways and time scales. For example, the LC presents mainly the GF signal and unconstrained Arctic river inputs (more

$^{236}$U relative to $^{129}$I) indicating the contribution from PIW-Canada through the Canadian Archipelago, while the EGC, largely influenced by the NRPs (more $^{129}$I relative to $^{236}$U), indicates the contribution of RAW and PIW-Atlantic. The contribution of RAW/PIW-Atlantic and PIW-Canada to DSOW in the Irminger and Labrador Seas is also visible thanks to slight elevations of tracer values in near bottom

depths and specific $^{236}$U/$^{238}$U - $^{129}$I/$^{236}$U ratios, due to cascading events. This work also contributes to extend the existing $^{129}$I time series in the Labrador Sea/Irminger Sea and allows a first assessment of time-varying $^{129}$I concentrations east of Reykjanes Ridge. Increasing $^{129}$I concentrations are observed in the western part of the GEOVIDE section and in the Icelandic Basin. In the WEB, the short observation time (2012 – 2014) does not allow yet seeing temporal trends of $^{129}$I levels. The $^{129}$I data in overflow waters of

the central Labrador Sea (1993 – 2014) can be fitted with reprocessing $^{129}$I (and $^{236}$U) input functions following earlier modelling studies to better understand the transport time scales and dilution factors of AWs tagged by the NRP signal. This study supports the current interpretations on the circulation of AWs, which apparently follow a short loop trough the Nordic Seas (8 – 10 years in this study) and a longer loop including the circulation in the Arctic Eurasian Basin (16 – 18 years in this study). Further experimental

and modelling studies on $^{129}$I and $^{236}$U may confirm circulation features highlighted by these tracers and to shed more light on novel findings such as the transport of ISOW in to the eastern SPNA, which plays an important role on the ventilation of the deep SPNA.

**Acknowledgements**

We are grateful to the captain and crew of *R/V Pourquoi Pas?*, as well as the technical team for their work at sea (P. Branellec, F. Desprez de Gésincourt, M. Hamon, C. Kermabon, P. Le Bot, S. Leizour, O. Ménage, F. Pérault, and E. de Saint-Léger). A big thank to Yi Tang for help sampling and to Anita Schlatter for assistance in the laboratory. This manuscript has been notably improved thanks to the constructive comments from two anonymous reviewers. Thanks to J. -L. Menzel and the group of E.

Achterberg (GEOMAR, Kiel) for providing the FISH device to collect surface seawater. The work of C. Schmechtig with the LEFE-CYBER database management is also acknowledged. Some of the $^{129}$I data discussed from the literature were kindly provided by V. Alfimov, H. Edmonds and J. N. Smith. Several

figures were created using *Ocean Data View* (Schlitzer, 2017). We would also like to thank Natalie Dubois and Alfred Lück for providing the laboratory space and assistance at EAWAG. The GEOVIDE cruise was funded by the French National Research Agency (ANR-13-BS06-0014, ANR-12-PDOC-0025-01), the French National Center for Scientific Research (CNRS-LEFE-CYBER), the LabexMER

(ANR-10-LABX-19), and Ifremer. This work was funded by the Ministerio de Economía y Competitividad of Spain (MDM2015-0552), the Generalitat de Catalunya (MERS 2017 SGR-1588) and consortium partners of the ETH-Zurich Laboratory of Ion Beam Physics (EAWAG, EMPA, and PSI). M. Castrillejo was funded by a FPU PhD studentship (AP-2012-2901) from the Ministerio de Educación, Cultura y Deporte of Spain and the ETH Zurich Postdoctoral Fellowship Program (17-2 FEL-30), co-

funded by the Marie Curie Actions for People COFUND Program. N. Casacuberta's research was funded by the AMBIZIONE grant (PZ00P2_154805) from the Swiss National Science Foundation. M.I. García-Ibáñez was supported by the Spanish Ministry of Economy and Competitiveness through the BOCATS (CTM2013-41048-P) project co-funded by the Fondo Europeo de Desarrollo Regional 2014–2020 (FEDER).

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

Table 1. Acronyms used to define water masses, geographic locations and radionuclide sources.

| Acronym | |
| --- | --- |
| AABW | Antarctic Bottom Water |
| AMOC | Atlantic Meridional Overturning Circulation |
| AMS | Accelerator Mass Spectrometry |
| AW | Atlantic Water |
| BSBW | Barents Sea Branch Water |
| DSOW | Denmark Strait Overflow Water |
| DWBC | Deep Western Boundary Current |
| EGC | East Greenland Current |
| ENACW | East North Atlantic Central Water |
| FSBW | Fram Strait Branch Water |
| GF | Global Fallout |
| ISOW | Iceland-Scotland Overflow Water |
| LB | Lithogenic Background |
| LC | Labrador Current |
| LH | La Hague |
| LSW | Labrador Sea Water |
| MW | Mediterranean Water |
| NAC | North Atlantic Current |
| NCC | Norwegian Coastal Current |
| NEADW$_L$ | North East Atlantic Deep Water lower |
| NRP | Nuclear Reprocessing Plant |
| NwAC | Norwegian Atlantic Current |
| OMP | Optimum Multi-Parameter analysis |
| PAP | Porcupine Abyssal Plain |
| PIW | Polar Intermediate Water |
| RAW | Return Atlantic Water |
| SAF | Sub-Arctic Front |
| SAIW | Sub-Arctic Intermediate Water |
| SF | Sellafield |
| SPMW | Sub-Polar Mode Water |
| SPNA | Sub-Polar North Atlantic |
| WEB | West European Basin |
| WSC | West Spitsbergen Current |

Table 2. Concentrations of $^{129}$I and $^{236}$U, and atom ratios of $^{236}$U/$^{238}$U and $^{129}$I/$^{236}$U in seawater samples collected during the GEOVIDE cruise in spring 2014. Uncertainties of radionuclide concentrations and the $^{236}$U/$^{238}$U ratio are given as one sigma deviations. The uncertainty of the $^{129}$I/$^{236}$U ratio was propagated from the concentration uncertainty. *$^{236}$U data corrected for cross talk contamination.

| Station and location | ETH Code | Depth m | Dominant water mass | Salinity | Pot. Temp. °C | Oxygen μmol/kg | $^{129}$I ×10$^7$ at/kg | | | $^{236}$U ×10$^6$ at/kg | | | $^{236}$U/$^{238}$U ×10$^{-12}$ at/at | | | $^{129}$I/$^{236}$U at ·at$^{-1}$ | | |
|---|---|---|---|---|---|---|---|---|---|---|---|---|---|---|---|---|---|---|
| Station 1 | TU1141-H160122 | 5 | ENACW | 35.16 | 16.49 | 250 | 7.2 | ± | 0.1 | 9.2 | ± | 0.4 | 1130 | ± | 60 | 7.9 | ± | 0.4 |
| 40° 19.99' N | TU1151-H160120 | 25 | ENACW | 35.50 | 15.09 | 262 | 7.4 | ± | 0.1 | 10.6 | ± | 0.8 | 1330 | ± | 100 | 7.0 | ± | 0.5 |
| 10° 2.16' W | TU1152-H160119 | 50 | ENACW | 35.70 | 13.76 | 254 | 8.2 | ± | 0.1 | 9.2 | ± | 0.5 | 1110 | ± | 60 | 8.9 | ± | 0.5 |
| Bottom depth: 3536 m | TU1153-H160118 | 100 | ENACW | 35.74 | 13.03 | 239 | 8.3 | ± | 0.1 | 8.4 | ± | 0.4 | 1100 | ± | 50 | 9.9 | ± | 0.5 |
| | TU1154-H160110 | 200 | ENACW | 35.71 | 12.50 | 228 | 7.5 | ± | 0.1 | 8.9 | ± | 0.5 | 1050 | ± | 60 | 8.5 | ± | 0.5 |
| | TU1147-H160103 | 400 | ENACW | 35.65 | 11.63 | 212 | 6.6 | ± | 0.1 | 8.6 | ± | 0.5 | 1030 | ± | 70 | 7.7 | ± | 0.5 |
| | TU1134-H160080 | 1000 | MW | 36.15 | 11.13 | 177 | 3.4 | ± | 0.1 | 8.3 | ± | 0.4 | 1000 | ± | 60 | 4.1 | ± | 0.2 |
| | TU1135-H160079 | 1200 | MW | 36.11 | 10.42 | 182 | 3.1 | ± | 0.1 | 8.4 | ± | 0.4 | 990 | ± | 60 | 3.7 | ± | 0.2 |
| | TU1136-H160078 | 1600 | MW/LSW | 35.52 | 6.74 | 220 | 3.1 | ± | 0.1 | 10.0 | ± | 0.6 | 1010 | ± | 60 | 3.1 | ± | 0.2 |
| | TU1123-H160077 | 2000 | LSW | 35.09 | 4.07 | 248 | 2.1 | ± | 0.1 | 7.9 | ± | 0.4 | 940 | ± | 50 | 2.6 | ± | 0.1 |
| | TU1124-H160076 | 2500 | NEADW$_L$ | 35.00 | 3.16 | 246 | 1.0 | ± | 0.1 | 2.5 | ± | 0.6 | 340 | ± | 110 | 4.1 | ± | 0.9 |
| | TU1125-H160075 | 3000 | NEADW$_L$ | 34.95 | 2.60 | 244 | 0.2 | ± | 0.2 | 2.3 | ± | 0.3 | 270 | ± | 30 | 0.9 | ± | 0.9 |
| | TU1126-H160074 | 3506 | NEADW$_L$ | 34.92 | 2.28 | 243 | 0.2 | ± | 0.2 | 4.4 | ± | 0.3 | 540 | ± | 40 | 0.5 | ± | 0.5 |
| Surface 1 | TU1148-H160123 | 5 | ENACW | | | | 7.0 | ± | 0.1 | 9.3 | ± | 0.4 | 1160 | ± | 50 | 7.5 | ± | 0.4 |
| 40° 19.98' N | | | | | | | | | | | | | | | | | | |
| 9° 27.57' W | | | | | | | | | | | | | | | | | | |
| Station 13 | TU1155-H160211 | 10 | ENACW | 35.85 | 15.55 | 259 | 8.5 | ± | 0.3 | 8.5 | ± | 0.4 | 950 | ± | 40 | 10.0 | ± | 0.5 |
| 41° 2298' N | TU1156-H160085 | 30 | ENACW | 35.83 | 15.00 | 259 | 7.7 | ± | 0.3 | 8.6 | ± | 0.3 | 980 | ± | 40 | 8.9 | ± | 0.5 |
| 13° 53.26' W | TU1158-H160061 | 65 | ENACW | 35.76 | 13.16 | 251 | 7.9 | ± | 0.3 | 8.8 | ± | 0.4 | 1010 | ± | 50 | 8.9 | ± | 0.5 |
| Bottom depth: 5347 m | TU1159-H160084 | 120 | ENACW | 35.74 | 12.84 | 247 | 7.6 | ± | 0.3 | 8.8 | ± | 0.3 | 1030 | ± | 40 | 8.6 | ± | 0.5 |
| | TU1160-H160083 | 250 | ENACW | 35.69 | 12.39 | 245 | 8.0 | ± | 0.3 | 8.8 | ± | 0.3 | 1030 | ± | 40 | 9.1 | ± | 0.5 |
| | TU1149-H160082 | 800 | IcSPMW | 35.72 | 10.51 | 182 | 4.9 | ± | 0.3 | 8.3 | ± | 0.3 | 980 | ± | 40 | 5.9 | ± | 0.4 |
| | TU1137-H160081 | 1150 | IcSPMW | 35.86 | 9.45 | 188 | 3.9 | ± | 0.2 | 8.7 | ± | 0.3 | 970 | ± | 40 | 4.4 | ± | 0.3 |
| | TU1127-H160073 | 2000 | LSW | 35.03 | 3.82 | 258 | 2.6 | ± | 0.2 | 6.3 | ± | 0.3 | 720 | ± | 40 | 4.2 | ± | 0.4 |
| | TU1129-H160072 | 3000 | NEADW$_L$ | 34.95 | 2.65 | 249 | 0.2 | ± | 0.2 | 1.2 | ± | 0.2 | 140 | ± | 20 | 2 | ± | 2 |
| | TU1130-H160071 | 4000 | NEADW$_L$ | 34.91 | 2.19 | 243 | 0.2 | ± | 0.2 | 0.3 | ± | 0.2 | 40 | ± | 20 | 6 | ± | 7 |
| | TU1131-H160070 | 4885 | NEADW$_L$ | 34.90 | 2.05 | 243 | 0.5 | ± | 0.1 | | | | | | | | | |
| | TU1132-H160069 | 5334 | NEADW$_L$ | 34.90 | 2.04 | 244 | 1.6 | ± | 0.1 | | | | | | | | | |
| Surface 2 | TU1161-H160215 | 5 | ENACW | | | | 8.9 | ± | 0.3 | 8.5 | ± | 0.4 | 990 | ± | 40 | 10.4 | ± | 0.6 |
| 44° 43.46' N | | | | | | | | | | | | | | | | | | |
| 18° 10.34' W | | | | | | | | | | | | | | | | | | |
| Station 21 | TU1162-H160124 | 5 | ENACW | 35.69 | 14.45 | 277 | 8.7 | ± | 0.2 | 8.6 | ± | 0.5 | 990 | ± | 60 | 10.2 | ± | 0.6 |
| 46° 32.65' N | TU1163-H160214 | 12 | ENACW | 35.69 | 14.45 | 277 | 9.1 | ± | 0.3 | 8.5 | ± | 0.4 | 970 | ± | 50 | 10.7 | ± | 0.6 |
| 19° 40.32' W | TU1164-H160213 | 50 | ENACW | 35.62 | 12.72 | 266 | 9.9 | ± | 0.3 | 8.5 | ± | 0.4 | 980 | ± | 50 | 11.6 | ± | 0.6 |
| Bottom depth: 4515 m | TU1165-H160121 | 100 | ENACW | 35.66 | 12.52 | 254 | 8.8 | ± | 0.2 | 8.1 | ± | 0.3 | 940 | ± | 40 | 10.9 | ± | 0.5 |
| | TU1166-H160212 | 200 | ENACW | 35.64 | 12.03 | 248 | 9.8 | ± | 0.3 | 8.4 | ± | 0.4 | 960 | ± | 40 | 11.7 | ± | 0.6 |
| | TU1170-H160104 | 450 | ENACW | 35.51 | 11.06 | 252 | 10.3 | ± | 0.2 | 9.1 | ± | 0.5 | 1060 | ± | 60 | 11.2 | ± | 0.6 |
| | TU1167-H160105 | 800 | IcSPMW | 35.31 | 9.03 | 188 | 8.6 | ± | 0.2 | 8.6 | ± | 0.4 | 980 | ± | 40 | 10.0 | ± | 0.5 |
| | TU1168-H160096 | 1500 | LSW | 35.00 | 4.40 | 259 | 8.4 | ± | 0.2 | 7.9 | ± | 0.5 | 970 | ± | 60 | 10.6 | ± | 0.6 |
| | TU1138-H160068 | 2300 | LSW | 34.92 | 3.22 | 273 | 5.5 | ± | 0.2 | 7.9 | ± | 0.3 | 930 | ± | 40 | 7.0 | ± | 0.4 |
| | TU1150-H160067 | 2750 | ISOW | 34.94 | 2.86 | 267 | 4.4 | ± | 0.2 | 6.4 | ± | 0.3 | 780 | ± | 40 | 6.8 | ± | 0.5 |
| | TU1133-H160066 | 4000 | NEADW$_L$ | 34.92 | 2.22 | 242 | 0.6 | ± | 0.2 | 0.6 | ± | 0.2 | 80 | ± | 20 | 9 | ± | 4 |
| | TU1139-H160065 | 4474 | NEADW$_L$ | 34.91 | 2.16 | 240 | 1.1 | ± | 0.2 | 0.6 | ± | 0.2 | 80 | ± | 20 | 17 | ± | 6 |
| Surface 3 | TU1171-H160219 | 5 | ENACW | | | | 10.9 | ± | 0.4 | 7.9 | ± | 0.4 | 920 | ± | 50 | 13.9 | ± | 0.8 |
| 47° 17.39' N | | | | | | | | | | | | | | | | | | |
| 20° 15.71' W | | | | | | | | | | | | | | | | | | |
| Station 26 | TU1172-H160125 | 5 | ENACW | 35.31 | 11.52 | 282 | 13.3 | ± | 0.2 | 8.8 | ± | 0.4 | 1000 | ± | 50 | 15.1 | ± | 0.7 |
| 50° 16.67' N | TU1173-H160218 | 25 | ENACW | 35.31 | 11.52 | 282 | 12.9 | ± | 0.3 | 8.2 | ± | 0.5 | 950 | ± | 60 | 16 | ± | 1 |
| 22° 36.28' W | TU1174-H160217 | 50 | ENACW | 35.19 | 9.85 | 291 | 16.7 | ± | 0.3 | 8.1 | ± | 0.5 | 970 | ± | 60 | 21 | ± | 1 |
| Bottom depth: 4130 m | TU1205-H160117 | 100 | ENACW | 35.17 | 9.45 | 284 | 19.2 | ± | 0.3 | 8.1 | ± | 0.3 | 940 | ± | 40 | 23.7 | ± | 0.9 |
| | TU1206-H160216 | 250 | SAIW | 35.14 | 8.67 | 265 | 20.4 | ± | 0.4 | 9.3 | ± | 0.4 | 1060 | ± | 40 | 21.9 | ± | 0.9 |
| | TU1188-H160111 | 500 | IcSPMW | 34.99 | 6.69 | 211 | 15.9 | ± | 0.3 | 8.3 | ± | 0.4 | 970 | ± | 50 | 19 | ± | 1 |
| | TU1183-H160106 | 1000 | LSW | 34.96 | 4.40 | 263 | 13.4 | ± | 0.2 | 8.2 | ± | 0.4 | 1020 | ± | 50 | 16.3 | ± | 0.9 |
| Surface 4 | TU1184-H160222 | 5 | SAIW | | | | 14.9 | ± | 0.3 | 7.8 | ± | 0.4 | 890 | ± | 50 | 19 | ± | 1 |
| 52° 7.99' N | | | | | | | | | | | | | | | | | | |
| 24° 3.50' W | | | | | | | | | | | | | | | | | | |
| Station 32 | TU0971-H160126 | 5 | SAIW | 35.07 | 10.16 | 292 | 19.9 | ± | 0.3 | 7.4 | ± | 0.6 | 870 | ± | 80 | 27 | ± | 2 |
| 55° 30.336' N | TU1200-H160221 | 30 | SAIW | 35.06 | 10.02 | 292 | 20.5 | ± | 0.4 | 8.1 | ± | 0.3 | 950 | ± | 40 | 25 | ± | 1 |
| 26° 42.62' W | TU1207-H160116 | 60 | SAIW | 35.14 | 8.85 | 288 | 18.9 | ± | 0.3 | 9.2 | ± | 0.3 | 1040 | ± | 40 | 20.6 | ± | 0.7 |
| Bottom depth: 3235 m | TU1208-H160220 | 150 | SAIW | 35.11 | 8.11 | 274 | 20.1 | ± | 0.4 | 8.3 | ± | 0.3 | 980 | ± | 40 | 24.1 | ± | 0.9 |
| | TU1175-H160097 | 380 | SAIW | 35.04 | 6.70 | 222 | 15.7 | ± | 0.3 | 8.0 | ± | 0.4 | 970 | ± | 50 | 19.5 | ± | 0.9 |

| Station | Sample | Depth | Water mass | Sal | Temp | | | | | |
|---|---|---|---|---|---|---|---|---|---|---|
| | TU1201-H160098 | 600 | IcSPMW | 34.96 | 5.02 | 246 | 17.8 ± 0.3 | 9.0 ± 0.2 | 1060 ± 30 | 19.8 ± 0.6 |
| | TU1176-H160099 | 850 | LSW | 34.93 | 4.32 | 267 | 15.8 ± 0.3 | 9.1 ± 0.4 | 1097 ± 50 | 17.3 ± 0.8 |
| | TU1202-H160112 | 1200 | LSW | 34.91 | 3.84 | 277 | 18.5 ± 0.3 | 9.7 ± 0.3 | 1130 ± 40 | 19.1 ± 0.6 |
| | TU1177-H160113 | 1700 | LSW | 34.93 | 3.55 | 272 | 12.3 ± 0.2 | 8.8 ± 0.4 | 1030 ± 50 | 14.1 ± 0.7 |
| | TU1178-H160064 | 2250 | LSW | 34.93 | 3.11 | 274 | 10.7 ± 0.3 | 8.1 ± 0.3 | 960 ± 40 | 13.2 ± 0.6 |
| | TU1179-H160063 | 2650 | ISOW | 34.96 | 2.84 | 271 | 10.5 ± 0.3 | 8.0 ± 0.4 | 920 ± 50 | 13.2 ± 0.7 |
| | TU1180-H160062 | 3220 | ISOW | 34.96 | 2.53 | 263 | 13.2 ± 0.3 | 7.0 ± 0.3 | 820 ± 40 | 18.8 ± 0.9 |
| Station 38 58° 50.56' N 31° 15.97' W Bottom depth: 1345 m | TU1189-H160230 | 5 | IcSPMW | 35.06 | 9.24 | 296 | 16.7 ± 0.3 | 8.7 ± 0.4 | 1030 ± 50 | 19 ± 1 |
| | TU1190-H160229 | 20 | IcSPMW | 35.06 | 9.23 | 296 | 16.7 ± 0.3 | 9.8 ± 0.3 | 1170 ± 50 | 17.0 ± 0.7 |
| | TU1193-H160228 | 60 | IcSPMW | 35.08 | 7.95 | 291 | 17.4 ± 0.3 | 8.8 ± 0.3 | 1010 ± 40 | 19.7 ± 0.8 |
| | TU1203-H160227 | 110 | IcSPMW | 35.10 | 7.45 | 273 | 18.7 ± 0.4 | 8.7 ± 0.3 | 1010 ± 40 | 21.4 ± 0.9 |
| | TU1204-H160226 | 300 | IcSPMW | 35.11 | 7.19 | 273 | 19.1 ± 0.4 | 9.0 ± 0.2 | 1040 ± 30 | 21.3 ± 0.7 |
| | TU1209-H160225 | 650 | IcSPMW | 35.06 | 5.94 | 234 | 15.5 ± 0.3 | 9.3 ± 0.3 | 1050 ± 40 | 16.6 ± 0.7 |
| | TU1210-H160224 | 1000 | LSW | 34.99 | 4.33 | 262 | 18.3 ± 0.4 | 9.6 ± 0.2 | 1070 ± 30 | 19.1 ± 0.6 |
| | TU1213-H160223 | 1336 | ISOW | 34.99 | 3.90 | 267 | 21.4 ± 0.5 | 9.3 ± 0.3 | 1100 ± 40 | 23 ± 1 |
| Surface 5 59° 16.77' N 35° 33.64' W | TU1194-H160277 | 5 | IrSPMW | | | | 16.8 ± 0.3 | 8.9 ± 0.4 | 1030 ± 40 | 19.0 ± 0.8 |
| Station 44 59° 37.36' N 38° 57.234' W Bottom depth: 2929 m | TU1217-H160276 | 5 | IrSPMW | 34.85 | 6.86 | 318 | 19.4 ± 0.3 | 9.4 ± 0.3 | 1120 ± 40 | 20.6 ± 0.8 |
| | TU1260- | 5 | IrSPMW | 34.85 | 6.86 | 318 | | 9 ± 3 | *1040 ± 330 | |
| | TU1195-H160275 | 20 | IrSPMW | 34.85 | 6.85 | 318 | 17.2 ± 0.3 | 9.9 ± 0.4 | 1170 ± 50 | 17.4 ± 0.8 |
| | TU1218-H160274 | 40 | IrSPMW | 34.88 | 4.62 | 306 | 20.9 ± 0.3 | 9.2 ± 0.3 | 1090 ± 40 | 22.8 ± 0.9 |
| | TU1219-H160273 | 80 | IrSPMW | 34.90 | 4.30 | 295 | 19.5 ± 0.3 | 9.1 ± 0.3 | 1070 ± 40 | 21.5 ± 0.7 |
| | TU1196-H160272 | 150 | IrSPMW | 34.90 | 4.05 | 298 | 18.4 ± 0.3 | 9.2 ± 0.3 | 1090 ± 40 | 20.0 ± 0.8 |
| | TU1220-H160271 | 300 | LSW | 34.90 | 3.94 | 298 | 22.5 ± 0.4 | 9.9 ± 0.3 | 1130 ± 30 | 22.8 ± 0.7 |
| | TU1221-H160253 | 850 | LSW | 34.87 | 3.62 | 292 | 24.5 ± 0.4 | 9.6 ± 0.7 | 1160 ± 80 | 25 ± 2 |
| | TU1185-H160254 | 1400 | LSW | 34.93 | 3.57 | 273 | 10.5 ± 0.2 | 9.2 ± 0.4 | 1070 ± 50 | 11.4 ± 0.5 |
| | TU1186-H160255 | 1800 | LSW | 34.93 | 3.26 | 274 | 14.3 ± 0.2 | 9.5 ± 0.3 | 1070 ± 40 | 15.1 ± 0.5 |
| | TU1191-H160256 | 2250 | ISOW | 34.93 | 2.86 | 276 | 15.5 ± 0.3 | 8.6 ± 0.3 | 1000 ± 40 | 18.0 ± 0.7 |
| | TU1248-H160308 | 2600 | ISOW | 34.92 | 2.44 | 282 | 38.7 ± 0.6 | 10 ± 1 | *1180 ± 130 | 40 ± 4 |
| | TU1261-H160307 | 2875 | DSOW | 34.88 | 1.17 | 308 | 98 ± 2 | 9 ± 4 | *1090 ± 420 | 110 ± 40 |
| | TU1262-H160306 | 2909 | DSOW | 34.89 | 1.06 | 309 | 99 ± 2 | 11 ± 3 | *1280 ± 310 | 90 ± 20 |
| Station 53 59° 53.86' N 43° 0.33' W Bottom depth: 193 m | TU1265-H160285 | 5 | PIW | 31.90 | -0.68 | 423 | 256 ± 4 | 16 ± 2 | *2050 ± 310 | 160 ± 20 |
| | -H160284 | 25 | PIW | 32.10 | -1.18 | 417 | 247 ± 5 | | | |
| | TU1267-H160283 | 50 | PIW | 32.95 | -1.52 | 364 | 236 ± 8 | 12 ± 4 | *1570 ± 510 | 200 ± 60 |
| Station 60 59° 47.96' N 42° 00.78' W Bottom depth: 1719 m | TU1214-H160282 | 5 | IrSPMW | 34.83 | 6.94 | 325 | 19.0 ± 0.3 | 8.9 ± 0.2 | 1040 ± 30 | 21.4 ± 0.6 |
| | TU1197-H160281 | 25 | IrSPMW | 34.98 | 6.56 | 314 | 17.0 ± 0.3 | 8.6 ± 0.3 | 1030 ± 40 | 19.6 ± 0.7 |
| | TU1192-H160280 | 50 | IrSPMW | 35.01 | 6.27 | 303 | 16.6 ± 0.3 | 8.3 ± 0.5 | 1020 ± 60 | 20 ± 1 |
| | TU1198-H160279 | 100 | IrSPMW | 35.02 | 5.97 | 291 | 17.1 ± 0.3 | 8.9 ± 0.3 | 1020 ± 40 | 19.3 ± 0.7 |
| | TU1224-H160278 | 200 | IrSPMW | 34.96 | 5.20 | 287 | 18.4 ± 0.3 | 8.9 ± 0.4 | 1030 ± 50 | 20.8 ± 0.9 |
| | TU1215-H160268 | 500 | IrSPMW | 34.94 | 4.45 | 280 | 22.8 ± 0.5 | 9.5 ± 0.4 | 1140 ± 50 | 24 ± 1 |
| | TU1216-H160257 | 750 | IrSPMW | 34.90 | 3.85 | 286 | 21.3 ± 0.4 | 9.3 ± 0.3 | 1120 ± 40 | 22.8 ± 0.8 |
| | TU1225-H160258 | 1000 | LSW | 34.92 | 3.77 | 278 | 23.1 ± 0.5 | 13 ± 2 | *1600 ± 210 | 18 ± 2 |
| | TU1229-H160259 | 1190 | LSW/ISOW | 34.93 | 3.61 | 277 | 25.6 ± 0.4 | 9.5 ± 0.9 | *1210 ± 120 | 27 ± 3 |
| | -H160260 | 1500 | LSW/ISOW | 34.92 | 3.16 | 281 | 30.3 ± 0.5 | | | |
| | TU1242-H160305 | 1672 | LSW/ISOW | 34.92 | 3.00 | 281 | 36.1 ± 0.6 | 9 ± 1 | *1100 ± 180 | 40 ± 6 |
| | TU1243-H160304 | 1712 | LSW/ISOW | 34.91 | 2.94 | 284 | 37.4 ± 0.6 | 8.7 ± 0.9 | *1080 ± 120 | 43 ± 5 |
| Station 61 59° 45.21' N 45° 6.74' W Bottom depth: 144 m | TU1263-H160296 | 5 | SAIW | 32.40 | 0.07 | 412 | 219 ± 4 | 15 ± 3 | 1940 ± 340 | 140 ± 30 |
| | -H160290 | 25 | SAIW | 32.63 | -0.41 | 402 | 244 ± 4 | | | |
| | TU1269-H160289 | 50 | SAIW | 32.81 | -0.65 | 383 | 251 ± 4 | | | |
| Station 64 59° 4.30' N 46° 5.29' W Bottom depth: 2473 m | TU1236-H160288 | 5 | IrSPMW | 34.81 | 6.71 | 317 | 24.7 ± 0.4 | 9 ± 1 | *1060 ± 160 | 29 ± 4 |
| | TU1226-H160287 | 25 | IrSPMW | 34.81 | 6.68 | 317 | 21.5 ± 0.4 | 12 ± 1 | *1370 ± 140 | 19 ± 2 |
| | TU1227-H160286 | 50 | IrSPMW | 34.86 | 6.13 | 319 | 18.4 ± 0.3 | 11 ± 1 | *1480 ± 180 | 16 ± 2 |
| | TU1230-H160261 | 100 | IrSPMW | 34.95 | 5.37 | 294 | 22.2 ± 0.4 | 11 ± 1 | *1390 ± 140 | 20 ± 2 |
| | -H160262 | 200 | IrSPMW | 34.94 | 4.79 | 289 | 29.2 ± 0.5 | | | |
| | TU1231-H160267 | 450 | LSW | 34.92 | 4.22 | 288 | 26.8 ± 0.5 | 9 ± 1 | *1080 ± 140 | 30 ± 4 |
| | TU1232-H160100 | 900 | LSW | 34.87 | 3.60 | 295 | 24.7 ± 0.4 | 7.9 ± 0.8 | *970 ± 100 | 31 ± 3 |
| | -H160088 | 1150 | LSW | 34.91 | 3.71 | 281 | 23.7 ± 0.4 | | | |
| | TU1222-H160089 | 1400 | LSW | 34.93 | 3.55 | 274 | 21.0 ± 0.3 | 9.8 ± 0.3 | 1130 ± 40 | 21.3 ± 0.7 |
| | TU1238-H160090 | 1800 | LSW | 34.92 | 3.05 | 278 | 27.7 ± 0.4 | 9 ± 1 | *1070 ± 120 | 31 ± 4 |
| | TU1228-H160303 | 2150 | ISOW | 34.93 | 2.70 | 278 | 20.9 ± 0.3 | 11 ± 1 | *1300 ± 150 | 19 ± 2 |
| | TU1249-H160302 | 2462 | DSOW | 34.90 | 2.15 | 291 | 56.2 ± 0.9 | 9.5 ± 0.9 | *1160 ± 110 | 60 ± 6 |
| Surface 6 56° 44.79' N 47° 31.31' W | TU1239-H160231 | 5 | LSW | | | | 30.9 ± 0.5 | | | |
| Station 69 55° 50.50' N 48° 5.59' W Bottom depth: 3678 m | -H160232 | 5 | LSW | 34.62 | 6.24 | 326 | 48.8 ± 0.8 | | | |
| | TU1254-H160233 | 25 | LSW | 34.67 | 3.72 | 324 | 49.9 ± 0.8 | 9 ± 3 | 1060 ± 350 | 60 ± 20 |
| | TU1264-H160234 | 50 | LSW | 34.79 | 3.78 | 297 | 41.8 ± 0.7 | 5 ± 3 | *640 ± 320 | 80 ± 40 |
| | -H160239 | 100 | LSW | 34.83 | 3.81 | 296 | 32.9 ± 0.6 | | | |
| | TU1240-H160 | 200 | LSW | 34.84 | 3.62 | 298 | 29.8 ± 0.5 | 9 ± 2 | *1080 ± 190 | 32 ± 6 |
| | TU1272101-H160 | 500 | LSW | 34.86 | 3.49 | 296 | 29.3 ± 0.5 | 7 ± 2 | *820 ± 290 | 40 ± 20 |
| | TU1273-H109160 | 1000 | LSW | 34.85 | 3.39 | 296 | 26.0 ± 0.4 | | | |
| | TU1274-H160092 | 1800 | LSW | 34.92 | 3.57 | 274 | 17.6 ± 0.3 | 5 ± 2 | *620 ± 300 | 30 ± 20 |
| | TU1275-H160241 | 2200 | LSW | 34.92 | 3.22 | 275 | 20.0 ± 0.4 | 8 ± 3 | *940 ± 330 | 26 ± 9 |
| | TU1233-H160093 | 2800 | ISOW | 34.92 | 2.70 | 279 | 24.5 ± 0.4 | 8.9 ± 0.9 | *1070 ± 110 | 28 ± 3 |
| | TU1251-H160301 | 3500 | ISOW | 34.91 | 1.75 | 294 | 72 ± 1 | 15 ± 2 | *1770 ± 270 | 49 ± 8 |

| Station | Sample | Depth (m) | Water mass | Salinity | Temp | | | ± | | | ± | | | ± | | | ± | |
|---|---|---|---|---|---|---|---|---|---|---|---|---|---|---|---|---|---|---|
| | TU1255-H160300 | 3659 | DSOW | 34.90 | 1.47 | 299 | 85 | ± | 1 | 19 | ± | 3 | *2230 | ± | 340 | 44 | ± | 7 |
| Surface 7<br>53° 40.18' N<br>49° 29.49' W | TU1277-H160269 | 5 | LSW | | | | 31.6 | ± | 0.5 | 10.2 | ± | 0.5 | 1220 | ± | 70 | 31 | ± | 2 |
| Surface 8<br>52° 46.17' N<br>51° 48.26' W | TU1278-H160270 | 5 | SAIW/PIW | | | | 71 | ± | 1 | 15.7 | ± | 0.9 | 2090 | ± | 140 | 45 | ± | 3 |
| Station 77<br>52° 59.98' N<br>51° 6.01' W<br>Bottom depth: 2522 m | TU1245-H160242 | 5 | LSW | 34.47 | 7.28 | 324 | 37.9 | ± | 0.6 | 9 | ± | 1 | *1040 | ± | 170 | 45 | ± | 7 |
| | -H160243 | 25 | LSW | 34.63 | 5.30 | 334 | 32.2 | ± | 0.6 | | | | | | | | | |
| | TU1279-H160244 | 50 | LSW | 34.74 | 3.57 | 312 | 32.3 | ± | 0.6 | 10.4 | ± | 0.4 | 1230 | ± | 50 | 31 | ± | 1 |
| | TU1246-H160245 | 100 | LSW | 34.79 | 3.42 | 304 | 32.4 | ± | 0.6 | 9 | ± | 1 | *1040 | ± | 170 | 38 | ± | 6 |
| | TU1280-H160246 | 200 | LSW | 34.83 | 3.46 | 299 | 30.9 | ± | 0.5 | 10.4 | ± | 0.4 | 1210 | ± | 50 | 30 | ± | 1 |
| | TU1281-H160102 | 650 | LSW | 34.86 | 3.46 | 294 | 27.0 | ± | 0.4 | 10.3 | ± | 0.4 | 1190 | ± | 50 | 26 | ± | 1 |
| | TU1284-H160247 | 950 | LSW | 34.87 | 3.41 | 294 | 24.1 | ± | 0.4 | 10.6 | ± | 0.3 | 1240 | ± | 50 | 22.7 | ± | 0.8 |
| | TU1234-H160094 | 1250 | LSW | 34.91 | 3.54 | 279 | 22.8 | ± | 0.4 | 9.5 | ± | 0.9 | *1130 | ± | 100 | 24 | ± | 2 |
| | TU1282-H160299 | 1700 | LSW | 34.92 | 3.12 | 276 | 21.7 | ± | 0.4 | 9.8 | ± | 0.5 | 1140 | ± | 70 | 22 | ± | 1 |
| | TU1285-H160095 | 2200 | ISOW | 34.92 | 2.60 | 280 | 35.5 | ± | 0.6 | 10.6 | ± | 0.3 | 1270 | ± | 50 | 34 | ± | 1 |
| | TU1252-H160298 | 2477 | ISOW | 34.91 | 2.18 | 285 | 48.7 | ± | 0.8 | 8 | ± | 3 | *1030 | ± | 310 | 60 | ± | 20 |
| | TU1286-H160297 | 2500 | ISOW | 34.91 | 2.09 | 287 | 47.0 | ± | 0.8 | 12.1 | ± | 0.6 | 1400 | ± | 70 | 39 | ± | 2 |
| Station 78<br>51° 59.33' N<br>53° 40.01' W<br>Bottom depth: 377 m | TU1256-H160248 | 5 | PIW | 31.79 | 5.17 | 342 | 75 | ± | 1 | 16 | ± | 4 | *2110 | ± | 500 | 50 | ± | 10 |
| | TU1257-H160249 | 25 | PIW | 32.86 | -0.57 | 452 | 77 | ± | 1 | 19 | ± | 3 | *2350 | ± | 370 | 41 | ± | 6 |
| | TU1270-H160250 | 50 | PIW | 33.06 | -1.46 | 340 | 74 | ± | 1 | 12 | ± | 4 | | | | 60 | ± | 20 |
| | TU1258-H160251 | 200 | PIW | 34.08 | 0.77 | 308 | 64 | ± | 1 | 11 | ± | 2 | *1380 | ± | 280 | 60 | ± | 10 |
| | TU1287-H160252 | 367 | PIW | 34.62 | 2.92 | 283 | 44.2 | ± | 0.8 | 11.2 | ± | 0.5 | 1350 | ± | 70 | 40 | ± | 2 |

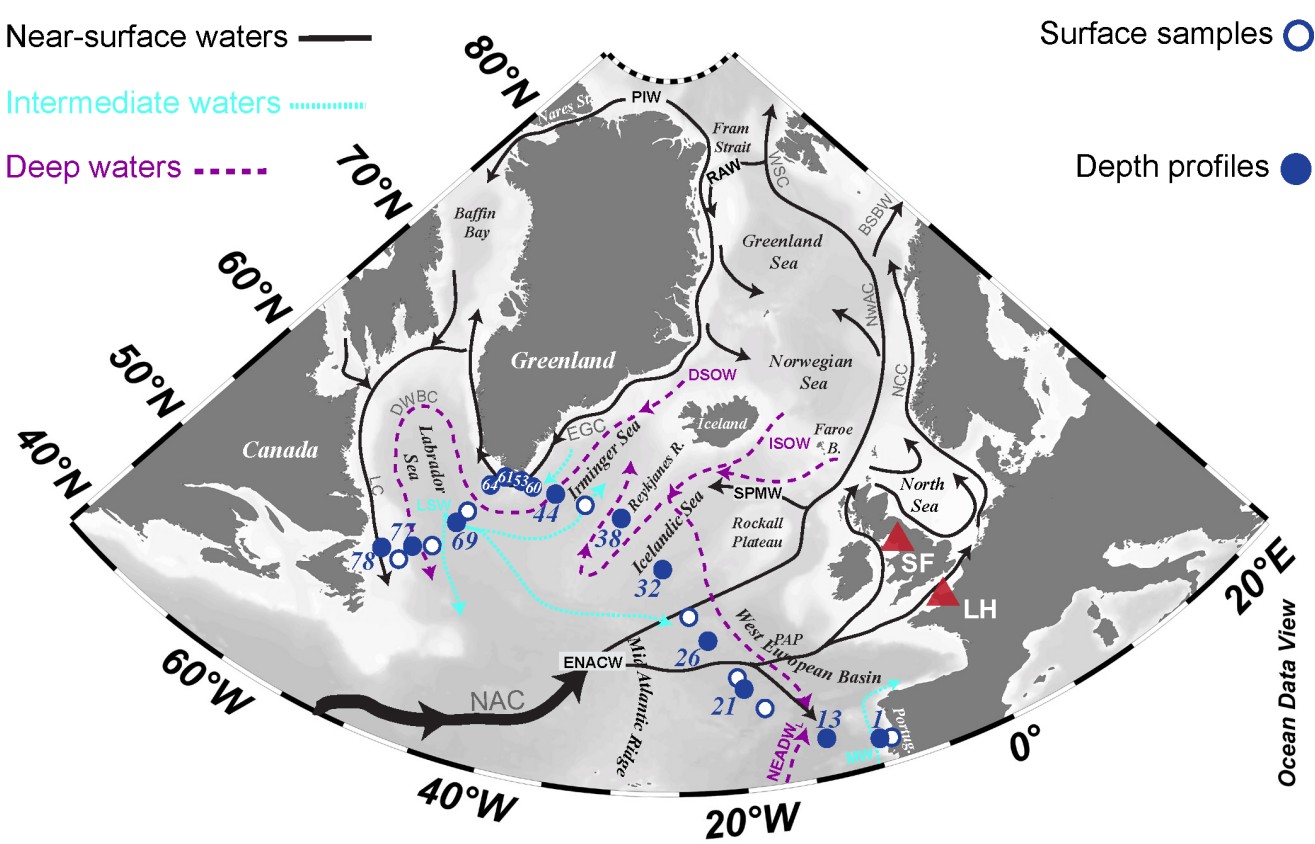

**Figure 1. Sampled locations in the subpolar North Atlantic during the GEOVIDE cruise in spring 2014. Nuclear fuel reprocessing plants of La Hague (LH) and Sellafield (SF) are represented along with main water masses and their schematic spreading pathways adapted from Daniault et al., (2016) and Smith et al., (2005). Acronyms are defined in Table 1.**

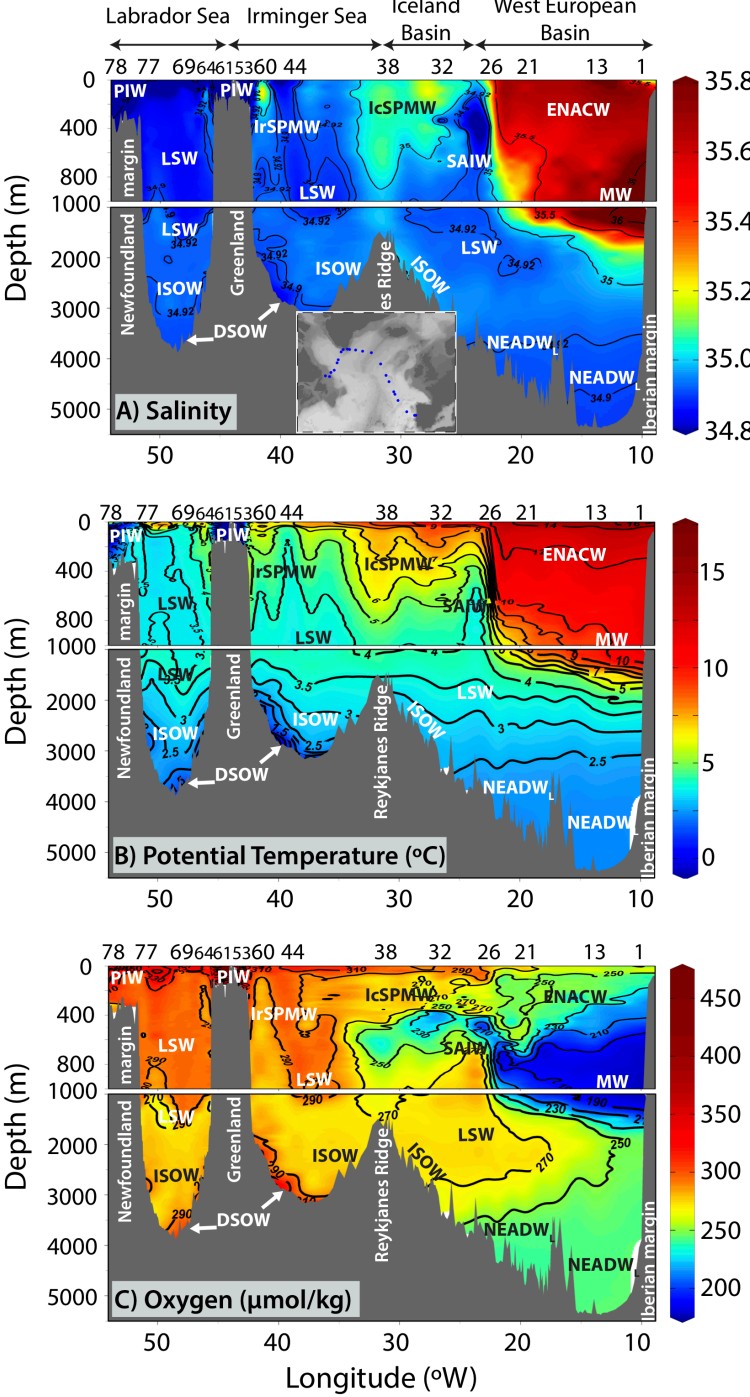

**Figure 2. Vertical distribution of (A) salinity, (B) potential temperature and (C) dissolved oxygen along the GEOVIDE section in spring 2014. Water mass acronyms are defined in Table 1.**

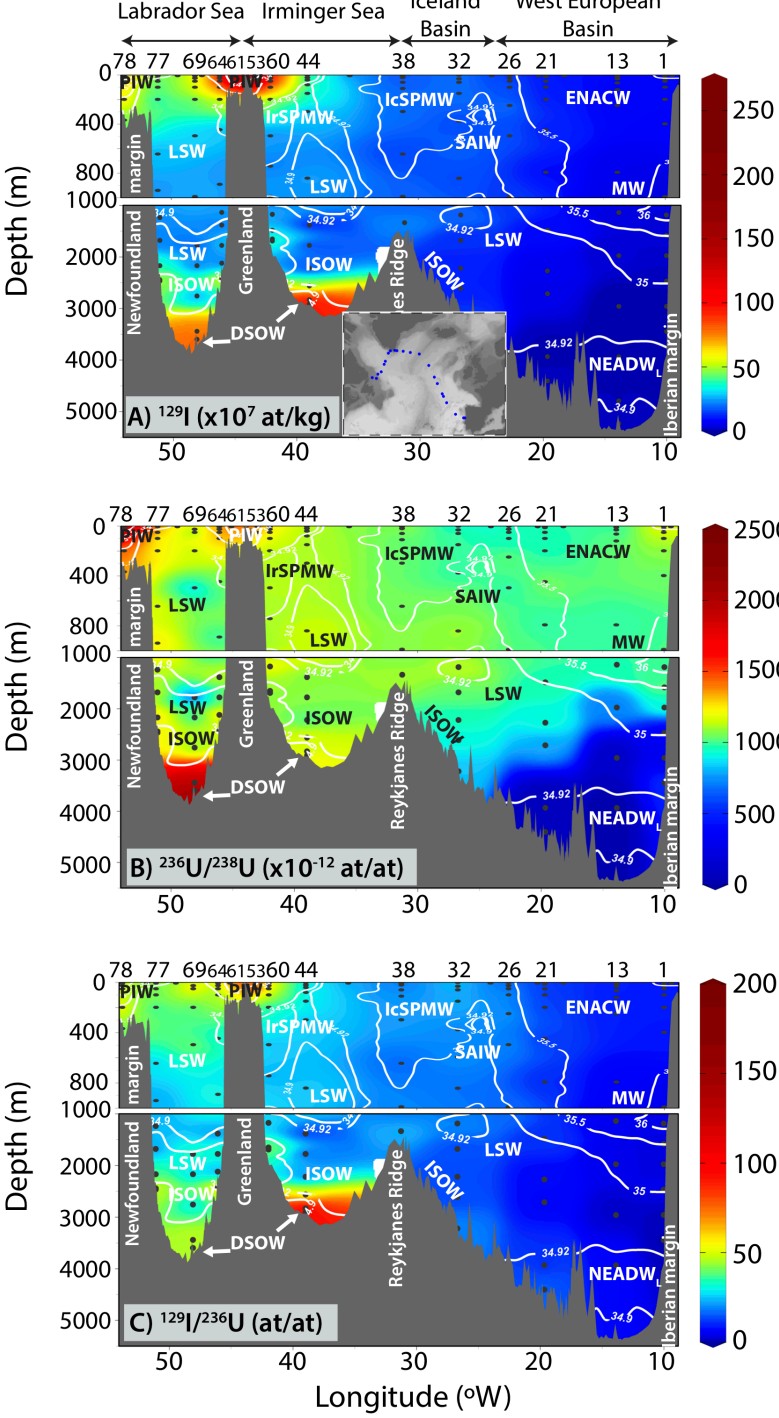

**Figure 3. Vertical distribution of (A) $^{129}$I concentrations, and atom ratios of (B) $^{236}$U/$^{238}$U and (C) $^{129}$I/$^{236}$U during the GEOVIDE cruise in spring 2014. Isohalines are overlaid to represent the water mass distribution. Acronyms are defined in Table 1.**

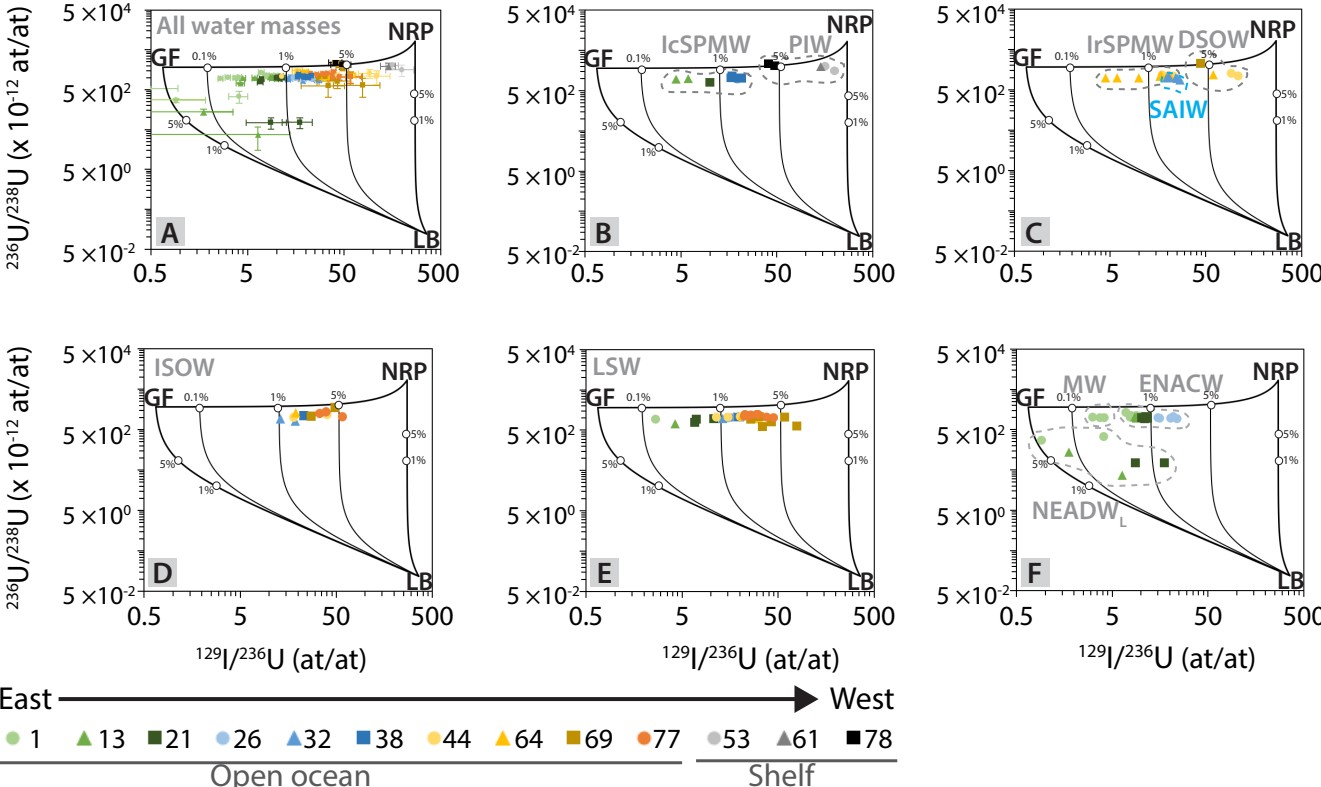

**Figure 4.** $^{129}$I/$^{236}$U and $^{236}$U/$^{238}$U atom ratios obtained during the GEOVIDE cruise in spring 2014 plotted on top of a binary mixing model (details in section 3.3) to estimate the contribution from the three sources to the subpolar North Atlantic (global fallout, GF; European nuclear fuel reprocessing plants, NRP; and the natural or lithogenic background, LB). Data are plotted for (A) each station, and (B to F) for the main water masses described in section 3.1. Diagram (A) also shows the data uncertainties. Explanation on how to interpret the results is provided in section 3.3. Further detail about the binary mixing model can be found in the original study (Casacuberta et al., 2016). Water mass acronyms are defined in Table 1.

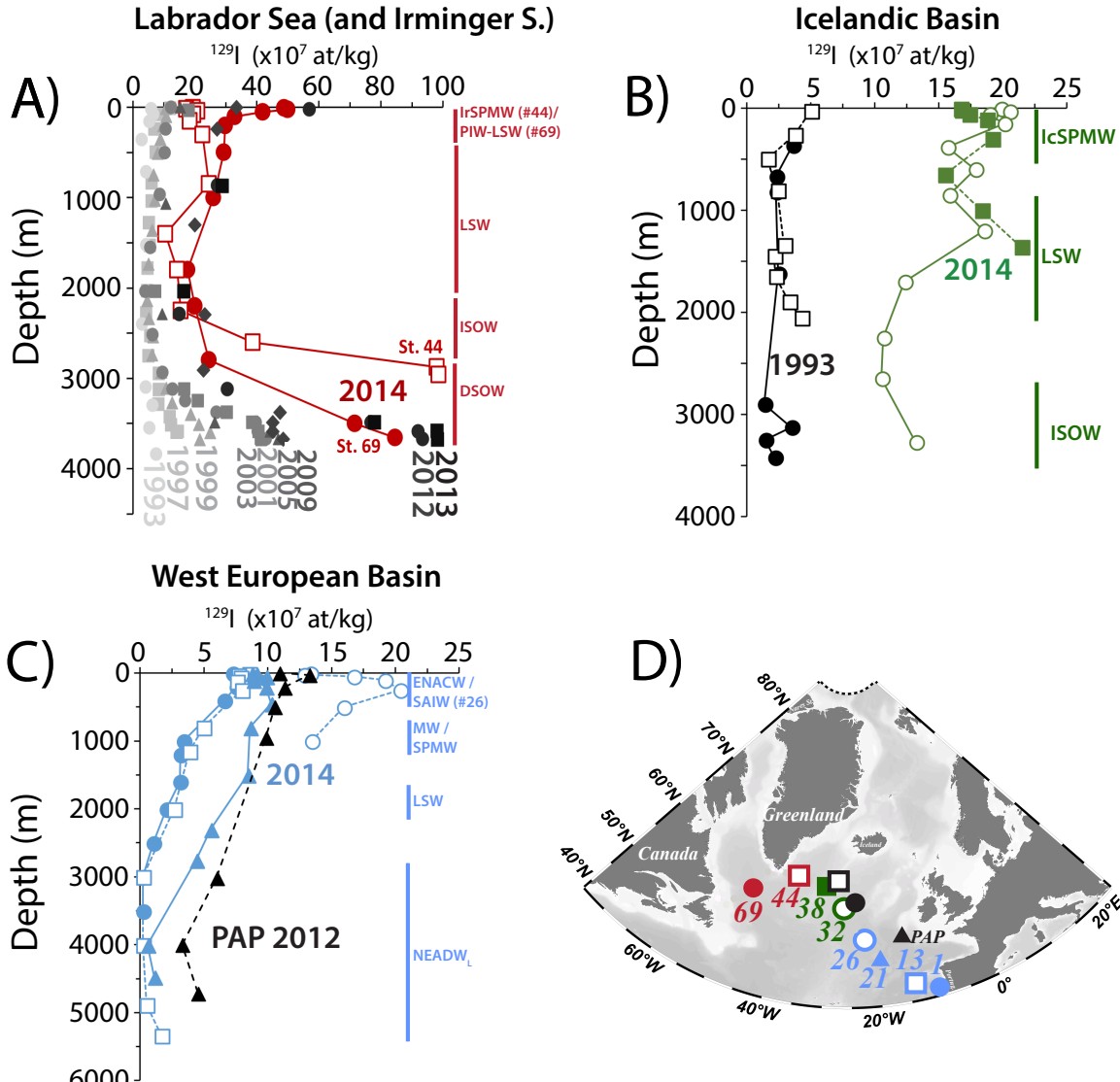

**Figure 5. Vertical profiles of $^{129}$I concentrations at locations shown in (D) of selected GEOVIDE stations and of those reported in nearby locations by earlier studies. (A) $^{129}$In the Labrador and Irminger Seas, red profiles represent data from this work (2014). Data from 1993 was reported southwest of GEOVIDE station 69 by Edmonds et al. (2001). Data from 1997 to 2013 were reported at Station 17 of the AR7W line: for 1997, 1999 and 2001 by Smith et al. (2005); for 2003, 2005 and 2009 by Orre et al. (2010); and for 2012 and 2013 by Smith et al. (2016). (B) In the Icelandic Basin, green profiles show data from GEOVIDE stations 32 and 38 in 2014, while black profiles represent data from 1993 reported by Edmonds et al. (2001). (C) In the West European Basin, blue profiles represent data from GEOVIDE stations 1 to 26, while data from 2012 in the Porcupine Abyssal Plain (PAP) was reported by Vivo-Vilches et al. (2018). Water masses found during GEOVIDE cruise have been summarized and represented in same colour as the $^{129}$I concentration profiles. Acronyms are defined in Table 1.**

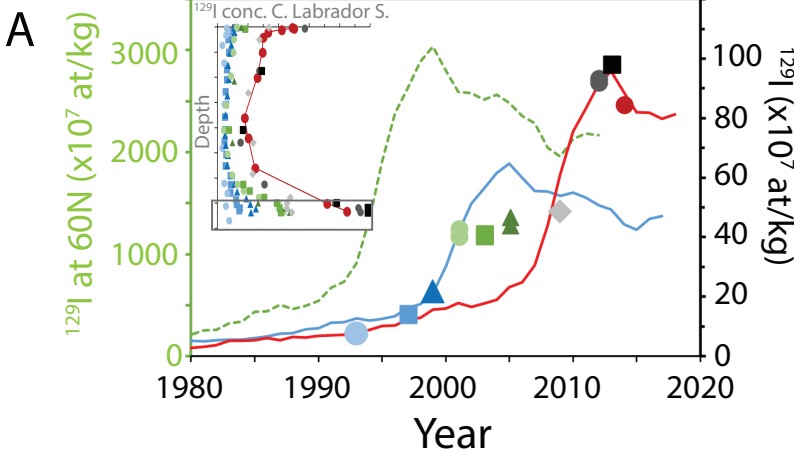

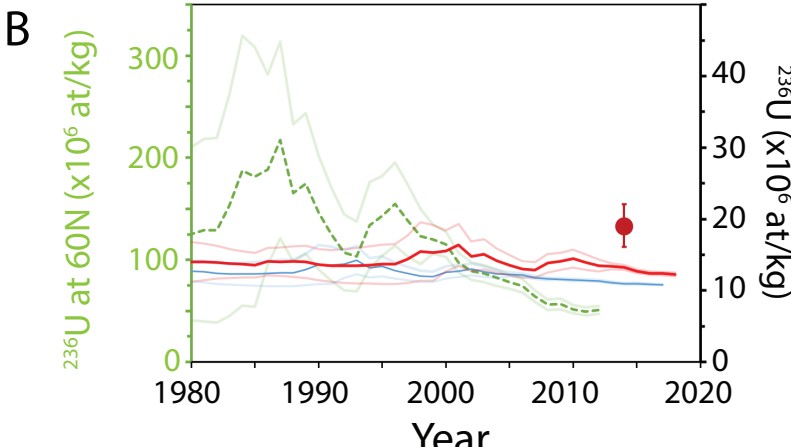

Figure 6. Comparison between measured (data points) and simulated (lines) radionuclide concentrations in bottom waters (DSOW) at the central Labrador Sea for (A) $^{129}$I and (B) $^{236}$U. The green y-axis on the left represents radionuclide concentrations over time (or the radionuclide input functions, also in green) at 60 ºN due to releases from the European Nuclear Reprocessing Plants and which were reported by Christl et al. (2015b). Note that for $^{236}$U, the average input function is accompanied with a lower and upper bound estimate to acknowledge the current uncertainties on the $^{236}$U releases from Sellafield and La Hague (Christl et al., 2015b). The black y-axis on the right represents radionuclide concentrations, measured or simulated, in bottom waters of the central Labrador Sea. Represented on these axes are the blue and red lines that represent two versions of the radionuclide input function at 60 ºN after dilution (DF=50 and 30 respectively) and application of a time delay (6 and 14 years respectively) that corresponds to the transport from 60 ºN to central Labrador Sea. The blue and red simulations include background concentrations for $^{129}$I (2.5 × $10^7$ at/kg; Edmonds et al., 2001) and $^{236}$U (10 × $10^6$ at/kg; e.g. Christl et al., 2012) due to the global fallout. In the black y-axis are also represented the radionuclide concentrations measured in 2014 that correspond to the bottom sample collected in GEOVIDE station 69. Earlier data on $^{129}$I concentrations at similar locations were reported for 1993 by Edmonds et al. (2001); for 1997, 1999 and 2001 by Smith et al. (2005); for 2003, 2005 and 2009 by Orre et al. (2010); and for 2012 and 2013 by Smith et al. (2016).