# Peer review of "Tracing water masses with 129I and 236U in the subpolar North Atlantic along the GEOTRACES GA01 section"

_Biogeosciences, 2018_

## Referee Comment (RC1) · Anonymous Referee #1 · 3 Jul 2018

This manuscript focus on the distribution of 129I and 236U along the GEOVIDE section (transect GEOTRACES GA01) in spring 2014. GEOVIDE cruise covered the subpolar North Atlantic Ocean and the Labrador Sea. It represents an important updated dataset and the authors successfully use 129I and 236/238U and 236U/129I atom ratios to describe water masses. The authors confirm with this study the major potential of the combination of 129I and 236U as circulation tracers, especially in the area of study and the Arctic Seas and I really enjoyed reading it.

However, I think that given that the combined use of 129I and 236U provide such rich information, some of the results provided could be discussed in more depth. My

impression is that the description and overall use of some the data still require a bit of discussion.

If I am not mistaken, the paper have three main objectives that should be emphasized and clarified in the abstract and the introduction.

1. Update and improve the database of 129I and 236, to be used for future studies and/or modelisation of the ocean circulation in the North Atlantic

2. Present new evidences of the advantages of using both radionuclides as dual tracers in the ocean. In this case, what I miss in the text is a more detailed explanation/introduction of why and how 236U, 129I and 236/129I combined provide different and complementary information. The authors reference previous works but should provide the reader with a bit of context and additional information about how these tracers/methodology work.

3. Use the tracers to understand ocean circulation in the area. This seems to be the main objective of the paper, however the conclusions from this part are mixed with the other two objectives, together with what is already known and what is novel in this paper. e.g. the final conclusion in the Abstract "Data of 129I and 236U from 2014 and the 129I time series in the Labrador Sea agrees with the hypothesis that Atlantic Waters follow at least two circulation loops from their source region [. . .] recirculation in the Arctic Eurasian Basin" is not new was already stated by Orre et al. (2010) with 129I and partially by Povinec et al., (2003) using other radioactive tracers such as 137Cs. But there is missing information in the abstract to emphasize that the other conclusions are indeed novel, i.e contribution of ISOW to eastern SPNA is quite recent.

A general comment on the paper is that it presents an impressive dataset and it would be desirable to make more clear which of the conclusions are confirmations of previous hypotheses/results. In the text it is indeed explained, however I think that the novel results, found mainly from the dual use of these radiotracers, are mixed with results that are confirmation of known facts and its relevance it is not explicitly enhanced,

which is a shame. Section 3.4 is basically where the novel features of these tracers are presented, in contrast with previous sections that basically use previous data and hypotheses and verify that the new 129I and 236U data are in agreement. However, this distinction is, in my opinion, not totally clear especially when presenting section 3.3. Novel and/or on discussion hypotheses reinforced by these dataset should be highlighted. I would also emphasize conclusions obtained by the use of 236U and 129I/236U, since they are novel tracers and the first time that they are measured simultaneously in the area. However, in this sense I find the Conclusion section very well structured.

Finally, it is assumed in the text that the reader knows well about the ocean circulation in the North Atlantic and Arctic Oceans and about 129I and 236U, if this is the case, the paper is quite straightforward to read. But in my opinion one can get easily lost if that is not the case, I have add a few examples of this in the specific comments below. To provide a general background to better understand the discussion of the results I suggest something like:

1. Presenting first a brief introduction to ocean circulation and water masses involved with the data.

2. Explain in more detail the role of 129I, 236U and 236U/129I as ocean tracers of the SPNA, making clear what we have learn so far using them i.e. provide context. It would be also good to better explain how to read and understand Figure 3. Which is extremely useful and provides a lot of information.

Please also note the supplement to this comment:
https://www.biogeosciences-discuss.net/bg-2018-228/bg-2018-228-RC1-supplement.pdf

**Supplement:**

***BG-2018-228. Tracing water masses with $^{129}$I and $^{236}$U in the subpolar North Atlantic along the GEOTRACES GA01 section***

This manuscript focus on the distribution of 129I and 236U along the GEOVIDE section (transect GEOTRACES GA01) in spring 2014. GEOVIDE cruise covered the subpolar North Atlantic Ocean and the Labrador Sea. This manuscript represents an important updated dataset and the authors successfully use 129I and 236/238U and 236U/129I atom ratios to describe water masses. The authors confirm with this study the major potential of the combination of 129I and 236U as circulation tracers, especially in the area of study and the Arctic Seas and I really enjoyed reading it.

However, I think that given that the combined use of 129I and 236U provide such rich information, some of the results provided could be discussed in more depth. My impression is that the description and overall use of some the data still require a bit of discussion.

If I am not mistaken, the paper have three main objectives that should be emphasized and clarified in the abstract and the introduction.

1. Update and improve the database of 129I and 236, to be used for future studies and/or modelisation of the ocean circulation in the North Atlantic

2. Present new evidences of the advantages of using both radionuclides as dual tracers in the ocean. In this case, what I miss in the text is a more detailed explanation/introduction of why and how 236U, 129I and 236/129I combined provide different and complementary information. The authors reference previous works but should provide the reader with a bit of context and additional information about how these tracers/methodology work.

3. Use the tracers to understand ocean circulation in the area. This seems to be the main objective of the paper, however the conclusions from this part are mixed with the other two objectives, together with what is already known and what is novel in this paper. e.g. the final conclusion in the Abstract "Data of 129I and 236U from 2014 and the 129I time series in the Labrador Sea agrees with the hypothesis that Atlantic Waters follow at least two circulation loops from their source region […] recirculation in the Arctic Eurasian Basin" is not new was already stated by Orre et al. (2010) with 129I and partially by Povinec et al., (2003) using other radioactive tracers such as 137Cs. But there is missing information in the abstract to emphasize that the other conclusions are indeed novel, i.e contribution of ISOW to eastern SPNA is quite recent.

A general comment on the paper is that it presents an impressive dataset and it would be desirable to make more clear which of the conclusions are confirmations of previous hypotheses/results. In the text it is indeed explained, however I think that the novel results, found mainly from the dual use of these radiotracers, are mixed with results that are confirmation of known facts and its relevance it is not explicitly enhanced, which is a shame. Section 3.4 is basically where the novel features of these tracers are presented, in contrast with previous sections that basically use previous data and hypotheses and verify that the new 129I and 236U data are in agreement. However, this distinction is, in my opinion, not totally clear especially when presenting section 3.3. Novel and/or on

discussion hypotheses reinforced by these dataset should be highlighted. I would also emphasize conclusions obtained by the use of 236U and 129I/236U, since they are novel tracers and the first time that they are measured simultaneously in the area. However, in this sense I find the Conclusion section very well structured.

Finally, it is assumed in the text that the reader knows well about the ocean circulation in the North Atlantic and Arctic Oceans and about 129I and 236U, if this is the case, the paper is quite straightforward to read. But in my opinion one can get easily lost if that is not the case, I have add a few examples of this in the specific comments below.

To provide a general background to better understand the discussion of the results I suggest something like:

1. Presenting first a brief introduction to ocean circulation and water masses involved with the data.

2. Explain in more detail the role of 129I, 236U and 236U/129I as ocean tracers of the SPNA, making clear what we have learn so far using them i.e. provide context. It would be also good to better explain how to read and understand Figure 3. Which is extremely useful and provides a lot of information.

**ABSTRACT**

I think that these lines "Results show that part of the effluents discharged from Sellafield and La Hague apparently enter the eastern SPNA directly through the Iceland-Scotland passage or the English Channel/Irish Sea, as it is shown by elevated 129I concentrations and 129I/238U ratios in shallow central waters flowing in the West European Basin (WEB)" are saying the same than these ones "The Iceland-Scotland Overflow Water spreading pathways into the eastern SPNA have been confirmed by the unequivocal transport of reprocessing 129I into the deep WEB".

When it is said "The Iceland-Scotland Overflow Water spreading pathways into the eastern SPNA have been confirmed by the *unequivocal transport* of reprocessing 129I into the deep WEB", it should be briefly explained why we find this transport unequivocal.

**INTRODUCTION**

When one reads from lines 15 (Page 3) to line 23 (Page 4) gets a very general idea about how 129I and 236U are distributed in the North Atlantic, but do not get a precise picture of what are the paths followed by the radionuclides when released by the RP. That information is given later in the text, the problem is that it is scattered in different sections of the manuscript.

Furthermore, lines 15 to 20 (Page 4) provides some information about previous results of 236U/129I however it does not explain what these numbers represent or why and how they change geographically or in time. For example, it is not explained *why* "Yet, LSW

and DSOW were *clearly* identified by 236U/238U >1000 x 10$^{-12}$"; or *why* the atom ratio varies *from* "$^{129}$I/$^{238}$U < *1 for GF to about 1 - 350* for European NRPs".

Lines 10-15 (Page 4). Why reference data are given here and not for 129I?

Line 18 (Page 5). No mention to deep water formation at the Greenland Sea? And ISOW formation? ISOW is later described (Line 7, Page 8), but it would be easier to follow the manuscript having the whole picture since the beginning.

**SECTION 3.1.**

Line 5 (Page 7). A brief introduction to 129I/236U ratios is missing to understand their values and the further discussions.

Line 10-30 (Page 7). I also miss a complete introduction to water mass structure. It will be easier to follow the discussion if first we understand water mass structure and then 129I and 236U/238U are given. This way, ISOW description (Lines 7 -11, Page 8) should be move to that introduction, and merge with description in Page 5.

Line 26 (Page 7). "SAIW probably incorporates 129I from precursor water masses (e.g., waters carried by LC and/or LSW) while forming in the western SPNA". Why is that? Some of the statements, like this one, are properly given but not explained in terms of 129I (or 236U/238U) values.

Line 5 (Page 8). "Thus, 2014 data probably reflects the dilution with old LSW and SPMW carrying less 129I and 236U than MW". How is it that waters from LSW and SPMW, both affected by NFRP, carry less 129I and 236U/238U than MW, also mainly affected by GF? Is it the influence of Marcule?

Lines 1-13 (Page 8). This is clearly explained, but it will be even easier to follow if the name of stations and references to Table 2 are given.

Lines 14 -18. As already said, previous brief introduction to the use of 129I/236U as tracer should be included to make this lines easier to follow.

This way it said "The highest 129I/236U ratios (> 100) are present in waters transported by the shallow EGC and LC. Overflow waters are also distinguishable by their relatively high 129I/236U ratios (60 to 110 for DSOW, 15 to 40 for ISOW)" Why is that?

**SECTION 3.2.**

Line 25- 30 (Page 8). I really like Figure 3. I contains lots of information, may be it could be further explained in the mentioned intro introducing the 129I/236U tracer?

**SECTION 3.3.**

Line 14 (Page 9). "129I discharge rate from European NRPs was observed in the whole water column, being more pronounced (about 10 times increase) in overflow waters". This actually an previously observed fact but an explanation should be given here.

Figure 4A. Indicate in the caption that Smith 2016 corresponds to 2012 and 2013 profiles. "The depth distribution of 129I concentrations in the Labrador Sea in 2014 (station 69), displays 129I concentrations in DSOW about 15 % lower than in 2012 – 2013 (Smith et al., 2016)". Is this because samples from 2012-2013 are measuring the peak in the NFRP releases? If this is the case, please mention that the explanation for that decrease will be given in Section 3.5.

As I said, it is a well-known fact DSOW present an increase in 129I concentrations for all years. This is already approached by previous works, but a brief discussion could be also given here.

Line 18 (Page 9). "The main difference between the 129I depth profiles in the Irminger Sea (station 44) and central Labrador Sea (station 69) in 2014 is the surface 129I peak in the latter one (Figure 4A). Which is probably caused by waters that split off from the boundary currents, either the West Greenland Current or the LC". I don't quite understand this. Splitting won't change 129I concentrations.

Line 26 (Page 9). "This similarity suggests little time variation and similar water mass composition for that region, although PAP might present slightly larger 129I concentrations because of its proximity to Sellafield and La Hague". And will support the later mentioned hypothesis of direct contribution of NFRP to SPNA without previous recirculation (Line 10, Page 10).

**SECTION 3.4.**

Line 17 (Page 10). "twice" instead of "two times"

Line 16 -17 (Page 10). "near-surface transport of 129I from European NRPs also across Iceland-Scotland into the eastern SPNA" is also clearly seen in Table 2. That shows that profiles 1, 13 and 21 strongly contrast from profiles 6 and 32. Not only due to ISOW (IcSPMW) contribution in intermediate depths but also at shallower depths.

Line 27 (Page 10). allowing to identify key circulation features such as the EGC/LC and the DWBC in the Labrador and Irminger Seas. Explain in terms of radioactive tracers.

Line 30 -30. Differences of 129I and 236U in boundary currents are mentioned but not explained. It should be further discussed in terms of radioactive tracers.

Line 1-2 (Page 11). "EGC shows particularly high 129I concentrations and 129I/236U ratios because it is carrying Arctic water of Atlantic origin (PIW-Atlantic) and RAW that have been largely influenced by NRP effluents". I assume the authors do not explain this further because this is well known from previous works. Nevertheless, a brief description should be given, may be in the previously mentioned introduction?

Line 5-6 (Page 11). "while its 236U/238U ratios are likely > 2000 10-12 due to GF and unconstrained Arctic rivers inputs". Influencing how? In 236U, 129I or both?

Line 12 (Page 11). "rise of 129I concentrations at certain depths on the Greenland slope (e.g., station 60; Figure 2 and Figure S1), and particularly in bottom waters of the Irminger Sea (station 44), which are probably related to the cascading of 129I-rich waters from the Greenland Shelf". And why not an increase in 236U?

Line 22-23 (Page 11). "The ISOW is best distinguished by its relative 129I concentration maxima". Explain origin of this maximum

Line 24 (Page 11). The differences can be more clearly seen in Table 2.

Line 24-25(Page 11). "Further, in the next years one can expect a stronger 129I signal associated with ISOW in the SPNA due to the releases from the NRPs". Explain this further.

Line 3 (Page 12). "The evolution of 129I (and 236U) in the SPNA is closely related to the effluents discharged from the two European NRPs". It sounds weird to mention this at the end of the paper.

Line 18 (Page 12). "Data reported in this study (2014) supports this 'Arctic loop' and suggests that the second 129I front probably peaked before the GEOVIDE cruise". Could Vivo et al. values be also used to support this "Arctic loop"?

---

## Referee Comment (RC2) · Anonymous Referee #2 · 15 Jul 2018

In general, this article presents new information about two circulation loops of Atlantic Waters which are tagged with nuclear reprocessing plant effluents from their source region based on the observations at stations from Lisbon (Portugal) to the southern tip of Greenland (Cape Farewell), and from Cape Farewell to St. John's (Newfoundland, Canada). The reviewer thinks that this article should be published in Biogeoscience, but there are several points should be revised before publication.

Major points: Page 8 line 25 The authors used a binary mixing model of which three end members are LB, GF and NRP. But, as the authors recognized and stated in the text, most of the samples can be explained by simple two end members model except

6 samples collected in the deeper layers (page 9, line 2) which towards the lithogenic background, LB. This means that in the surface to mid depths in this region, to discuss sources of 129I and 236U in the SPNA, the reviewer thinks that it is enough to use simple two end members mixing model and the authors can revise the discuss here.

Page12 Line 1 -27 The discussion about transit times and dilution factors in the paragraph is poor and difficult to understand how the authors calculate time scales of 8-10 years for shorter loop and 8-18 years for longer loop. This 8-18 years statement is also inconsistent the numbers stated "between the maximum 16-8 years (page 13 line 19)" in the conclusion.

The authors used 129I input function at 60 N deg. By Christle 2015 and compared observational peak. But the input function already includes several assumptions and based on the figure caption, no explanation in the main text, the authors expanded the function to fit the measurement. But as shown in Figures 4A and 4B, the reviewer observes inconsistency between input functions and observations for both 129I and 236U. Therefore, the reviewer suggests that the authors can and should collaborate with numerical modeling guys to get modeling results and compared with authors observation.

Minor points: page 2 line 25-29 The authors should add about 238U data in their study.

Page 5 line 25 and 24 12L Niskin bottles.→ and 24 of 12L Niskin bottles.?

Page 7 line 8 The authors used data marked * , but the uncertainties are so large for 236U/238U ratio 129I/236U ratio as 2350+- 370 and 200+-60, respectively. These numbers should be in the blanket ( ), and 2090+-140 and 140+-30 should be used. Due to larger uncertainty, 2350+-370 and 2090+-140 mean within the same and 200+-60 and 140+-30 locate are also within the same.

Page 8 line 3 andc∼ 1600 x . The reviewer can not understand the meaning of this part. Please clarifiy the meaning of this part.

Page 27 Figure4 Caption of Figure 4 is not enough and color coordinations for previous and current date are not good, eg. think open green circle in Fig.4B was hard to find in Fig.4D. Time series data in Fig.4A is also not good to undestand temporal changed of 129I concentration. In general, all figure captions did not contain enough information about meaning of each color and each mark. Please state more precisely.

End of comments.

---

## Author Comment (AC1) · 27 Jul 2018

Maxi Castrillejo et al., on behalf of all co-authors. maxic@phys.ethz.ch

Dear Reviewer 1,

We are grateful for the thorough review and the constructive comments.

We will gladly make most of the changes suggested. Special emphasis will be put on providing a more detailed introduction to provide a better background about the tracer sources/levels, their transport and distribution; the ocean circulation of the study area, and highlight the pursued objectives. The 'results and discussion' will include a first new

section with the complete description of the water mass structure. The overall discussion shall be revised to make clear what is the novel information obtained from these tracers and the GEOVIDE cruise and what are confirmations of earlier tracer/physical studies. The specific comments will also be addressed, as shown in the point by point answer (in bold) to the comments made by the reviewer (in italic).

On behalf of all coauthors,

Maxi Castrillejo

Please also note the supplement to this comment:
https://www.biogeosciences-discuss.net/bg-2018-228/bg-2018-228-AC1-supplement.pdf

**Supplement:**

Maxi Castrillejo et al., on behalf of all co-authors.
maxic@phys.ethz.ch

Dear Reviewer 1,

We are grateful for the thorough review and the constructive comments from the reviewer.

We will gladly make most of the changes suggested. Special emphasis will be put on providing a more detailed introduction to provide a better background about the tracer sources/levels, their transport and distribution; the ocean circulation of the study area and also, highlight the pursued objectives. The 'results and discussion' will include a first, new section with the complete description of the water mass structure. The overall discussion shall be revised to make clear what is the novel information obtained from these tracers and the GEOVIDE cruise and what are confirmations of earlier tracer/physical studies. The specific comments will also be addressed, as shown in the point by point answer (in bold) to the comments made by the reviewer (in italic).

On behalf of all coauthors,

Maxi Castrillejo

**Detailed point by point answers to reviewer 1.**

*This manuscript focus on the distribution of $^{129}$I and $^{236}$U along the GEOVIDE section (transect GEOTRACES GA01) in spring 2014. GEOVIDE cruise covered the subpolar North Atlantic Ocean and the Labrador Sea. This manuscript represents an important updated dataset and the authors successfully use $^{129}$I and $^{236}/^{238}$U and $^{236}$U/$^{129}$I atom ratios to describe water masses. The authors confirm with this study the major potential of the combination of $^{129}$I and $^{236}$U as circulation tracers, especially in the area of study and the Arctic Seas and I really enjoyed reading it. However, I think that given that the combined use of $^{129}$I and $^{236}$U provide such rich information, some of the results provided could be discussed in more depth. My impression is that the description and overall use of some the data still require a bit of discussion.*

*If I am not mistaken, the paper have three main objectives that should be emphasized and clarified in the abstract and the introduction.*

*1. Update and improve the database of $^{129}$I and $^{236}$, to be used for future studies and/or modelisation of the ocean circulation in the North Atlantic*

*2. Present new evidences of the advantages of using both radionuclides as dual tracers in the ocean. In this case, what I miss in the text is a more detailed explanation/introduction of why and how $^{236}$U, $^{129}$I and $^{236}/^{129}$I combined provide different and complementary information. The authors reference previous works but should provide the reader with a bit of context and additional information about how these tracers/methodology work.*

*3. Use the tracers to understand ocean circulation in the area. This seems to be the main objective of the paper, however the conclusions from this part are mixed with the other two objectives, together with what is already known and what is novel in this paper. e.g. the final conclusion in the Abstract "Data of $^{129}$I and $^{236}$U from 2014 and the $^{129}$I time series in the Labrador Sea agrees with the hypothesis that Atlantic Waters follow at least two circulation loops from their source region […] recirculation in the Arctic Eurasian Basin" is not new was already stated by Orre et al. (2010) with $^{129}$I and partially by Povinec et al., (2003) using other radioactive tracers such as $^{137}$Cs. But there is missing information in the abstract to emphasize that the other conclusions are indeed novel, i.e contribution of ISOW to eastern SPNA is quite recent.*

*A general comment on the paper is that it presents an impressive dataset and it would be desirable to make more clear which of the conclusions are confirmations of previous hypotheses/results. In the*

*text it is indeed explained, however I think that the novel results, found mainly from the dual use of these radiotracers, are mixed with results that are confirmation of known facts and its relevance it is not explicitly enhanced, which is a shame. Section 3.4 is basically where the novel features of these tracers are presented, in contrast with previous sections that basically use previous data and hypotheses and verify that the new $^{129}I$ and $^{236}U$ data are in agreement. However, this distinction is, in my opinion, not totally clear especially when presenting section 3.3. Novel and/or on discussion hypotheses reinforced by these dataset should be highlighted. I would also emphasize conclusions obtained by the use of $^{236}U$ and $^{129}I/^{236}U$, since they are novel tracers and the first time that they are measured simultaneously in the area. However, in this sense I find the Conclusion section very well structured.*

**The discussion will be modified in order to make clear which are the results that confirm hypotheses/results reported in the literature and we shall highlight the novel results. Special care will be taken in section 3.3.**

*Finally, it is assumed in the text that the reader knows well about the ocean circulation in the North Atlantic and Arctic Oceans and about $^{129}I$ and $^{236}U$, if this is the case, the paper is quite straightforward to read. But in my opinion one can get easily lost if that is not the case, I have add a few examples of this in the specific comments below.*

**A more complete background will be provided on the introduction section about the circulation in the North Atlantic, and about the origin and transport of these tracers.**

*To provide a general background to better understand the discussion of the results I suggest something like:*

*1. Presenting first a brief introduction to ocean circulation and water masses involved with the data.*

**The tracer transport will be explained in more detail considering the currents and water masses involved.**

*2. Explain in more detail the role of $^{129}I$, $^{236}U$ and $^{236}U/^{129}I$ as ocean tracers of the SPNA, making clear what we have learn so far using them i.e. provide context.*

**More detail will be provided on the use of these tracers and about the knowledge obtained from them.**

*It would be also good to better explain how to read and understand Figure 3. Which is extremely useful and provides a lot of information.*

**A better explanation will be provided in the main text and the caption of Figure 3 to facilitate the interpretation of the data.**

*ABSTRACT*
*I think that these lines "Results show that part of the effluents discharged from Sellafield and La Hague apparently enter the eastern SPNA directly through the Iceland-Scotland passage or the English Channel/Irish Sea, as it is shown by elevated $^{129}I$ concentrations and $^{129}I/^{238}U$ ratios in shallow central waters flowing in the West European Basin (WEB)" are saying the same than these ones "The Iceland-Scotland Overflow Water spreading pathways into the eastern SPNA have been confirmed by the unequivocal transport of reprocessing $^{129}I$ into the deep WEB".*

**The first sentence refers to the shallow tracer transport while the second refers to the deep transport of ISOW. This will be clarified.**

*When it is said "The Iceland-Scotland Overflow Water spreading pathways into the eastern SPNA have been confirmed by the unequivocal transport of reprocessing $^{129}I$ into the deep WEB", it should be briefly explained why we find this transport unequivocal.*

**The increase in $^{129}$I concentrations at those depths but not in overlying waters can only be explained by the intrusion of dense waters carrying the signal from the European nuclear reprocessing plants. This will be explained accordingly referring to tracer data.**

*INTRODUCTION*
*When one reads from lines 15 (Page 3) to line 23 (Page 4) gets a very general idea about how $^{129}$I and $^{236}$U are distributed in the North Atlantic, but do not get a precise picture of what are the paths followed by the radionuclides when released by the RP. That information is given later in the text, the problem is that it is scattered in different sections of the manuscript.*

**A complete and precise description of tracer transport pathways will be provided in the introduction instead of scattering the information in different sections.**

*Furthermore, lines 15 to 20 (Page 4) provides some information about previous results of $^{236}$U/$^{129}$I however it does not explain what these numbers represent or why and how they change geographically or in time. For example, it is not explained why "Yet, LSW and DSOW were clearly identified by $^{236}$U/$^{238}$U >1000 x 10$^{-12}$"; or why the atom ratio varies from "$^{129}$I/$^{238}$U < 1 for GF to about 1 - 350 for European NRPs".*

**The sources and geographical distribution of the tracers will be better explained so that the reader understands what their potential is for tracing water masses.**

*Lines 10-15 (Page 4). Why reference data are given here and not for $^{129}$I?*

**The referencing will be provided for both tracers.**

*Line 18 (Page 5). No mention to deep water formation at the Greenland Sea? And ISOW formation? ISOW is later described (Line 7, Page 8), but it would be easier to follow the manuscript having the whole picture since the beginning.*

**The deep-water formation and transport of DSOW and ISOW will be included in the aforementioned description of tracer transport pathways.**

*SECTION 3.1.*
*Line 5 (Page 7). A brief introduction to $^{129}$I/$^{236}$U ratios is missing to understand their values and the further discussions.*

**A more detailed description on the sources of $^{129}$I and $^{236}$U, and on the range of values for $^{129}$I/$^{236}$U will be provided in the introduction making the discussion more straightforward.**

*Line 10-30 (Page 7). I also miss a complete introduction to water mass structure. It will be easier to follow the discussion if first we understand water mass structure and then 129I and 236U/238U are given.*

**The discussion will begin with a new section to present the water mass structure in 2014.**

*This way, ISOW description (Lines 7 -11, Page 8) should be move to that introduction, and merge with description in Page 5.*

**The change will be made as recommended by the reviewer.**

*Line 26 (Page 7). "SAIW probably incorporates $^{129}$I from precursor water masses (e.g., waters carried by LC and/or LSW) while forming in the western SPNA". Why is that? Some of the statements, like this one, are properly given but not explained in terms of $^{129}$I (or $^{236}$U/$^{238}$U) values.*

**This sentence (and similar ones) will be clarified providing information on the water mass origin/transformation and how that can be observed by higher $^{129}$I and $^{129}$I/$^{236}$U. Also, the overall comprehension of the discussion will be improved with a more complete background about the tracer sources and transport in the introduction section.**

*Line 5 (Page 8). "Thus, 2014 data probably reflects the dilution with old LSW and SPMW carrying less $^{129}I$ and $^{236}U$ than MW". How is it that waters from LSW and SPMW, both affected by NFRP, carry less $^{129}I$ and $^{236}U/^{238}U$ than MW, also mainly affected by GF? Is it the influence of Marcule?*

**Yes. The $^{129}I$ concentrations and $^{236}U/^{238}U$ atom ratios are larger than expected in the Mediterranean Sea. Recent work showed that this is very likely due to the discharge of $^{129}I$ and $^{236}U$ from the Marcoule Facility (see Castrillejo et al 2017, Science of the Total Environment).**

**The sentence will be completed and clarified including information about Marcoule.**

*Lines 1-13 (Page 8). This is clearly explained, but it will be even easier to follow if the name of stations and references to Table 2 are given.*

**The station numbers will be provided, as well as the reference for Table 2.**

*Lines 14 -18. As already said, previous brief introduction to the use of $^{129}I/^{236}U$ as tracer should be included to make this lines easier to follow. This way it said "The highest $^{129}I/^{236}U$ ratios (> 100) are present in waters transported by the shallow EGC and LC. Overflow waters are also distinguishable by their relatively high $^{129}I/^{236}U$ ratios (60 to 110 for DSOW, 15 to 40 for ISOW)" Why is that?*

**That is because they have a larger contribution from the European nuclear fuel reprocessing plants than other waters which are affected only by global fallout. This will be clearly stated. Also, the tracer source(s), geographical distribution and transport will be better explained in the introduction to facilitate the discussion.**

*SECTION 3.2.*
*Line 25- 30 (Page 8). I really like Figure 3. I contains lots of information, may be it could be further explained in the mentioned intro introducing the $^{129}I/^{236}U$ tracer?*

**The range of values for $^{129}I/^{236}U$ and its potential to provide information on the source and origin of the water mass will be more clearly explained in the introduction. Also, the text and the caption associated with Figure 3 will be completed. This will clarify the overall use of the dual tracer to constrain the radionuclide sources and the ocean circulation.**

*SECTION 3.3.*
*Line 14 (Page 9). "$^{129}I$ discharge rate from European NRPs was observed in the whole water column, being more pronounced (about 10 times increase) in overflow waters". This actually an previously observed fact but an explanation should be given here.*

**The explanation shall be provided and the previously reported observation will be acknowledged.**

*Figure 4A. Indicate in the caption that Smith 2016 corresponds to 2012 and 2013 profiles. "The depth distribution of $^{129}I$ concentrations in the Labrador Sea in 2014 (station 69), displays $^{129}I$ concentrations in DSOW about 15 % lower than in 2012 – 2013 (Smith et al., 2016)". Is this because samples from 2012-2013 are measuring the peak in the NFRP releases? If this is the case, please mention that the explanation for that decrease will be given in Section 3.5.*

**We think that it is the most likely explanation. The change will be made as recommended by the reviewer.**

*As I said, it is a well-known fact DSOW present an increase in $^{129}I$ concentrations for all years. This is already approached by previous works, but a brief discussion could be also given here.*

**The manuscript will be modified to make clear which are confirmations of previous studies and to provide a brief discussion related to the cited literature.**

*Line 18 (Page 9). "The main difference between the $^{129}I$ depth profiles in the Irminger Sea (station 44) and central Labrador Sea (station 69) in 2014 is the surface $^{129}I$ peak in the latter one (Figure 4A).*

*Which is probably caused by waters that split off from the boundary currents, either the West Greenland Current or the LC". I don't quite understand this. Splitting won't change $^{129}$I concentrations.*

**The EGC and the LC carry particularly high $^{129}$I and $^{236}$U, respectively. We propose that the surface in the C. Labrador Sea might have been influenced by waters that separated from the mainstream of the EGC, which are characterized by specially high $^{129}$I. The sentence shall be changed to clarify the interpretation.**

*Line 26 (Page 9). "This similarity suggests little time variation and similar water mass composition for that region, although PAP might present slightly larger $^{129}$I concentrations because of its proximity to Sellafield and La Hague". And will support the later mentioned hypothesis of direct contribution of NFRP to SPNA without previous recirculation (Line 10, Page 10).*

**Indeed, it could be the case. We shall add the reviewers point to reinforce the hypothesis on the direct contribution from Sellafield and La Hague.**

*SECTION 3.4.*
*Line 17 (Page 10). "twice" instead of "two times"*

**The change will be made as pointed by the reviewer.**

*Line 16 -17 (Page 10). "near-surface transport of $^{129}$I from European NRPs also across Iceland-Scotland into the eastern SPNA" is also clearly seen in Table 2. That shows that profiles 1, 13 and 21 strongly contrast from profiles 26 and 32. Not only due to ISOW (IcSPMW) contribution in intermediate depths but also at shallower depths.*

**The station numbers and Table 2 will be referenced within the sentence.**

*Line 27 (Page 10). allowing to identify key circulation features such as the EGC/LC and the DWBC in the Labrador and Irminger Seas. Explain in terms of radioactive tracers.*

**The discussion will be modified here and whenever necessary to explain the circulation features in terms of tracer observations.**

*Line 30 -30. Differences of $^{129}$I and $^{236}$U in boundary currents are mentioned but not explained. It should be further discussed in terms of radioactive tracers.*

**As pointed above, the discussion will be modified whenever necessary to explain the circulation features in terms of tracer observations.**

*Line 1-2 (Page 11). "EGC shows particularly high $^{129}$I concentrations and $^{129}$I/$^{236}$U ratios because it is carrying Arctic water of Atlantic origin (PIW-Atlantic) and RAW that have been largely influenced by NRP effluents". I assume the authors do not explain this further because this is well known from previous works. Nevertheless, a brief description should be given, may be in the previously mentioned introduction?*

**The transport of Atlantic and Pacific waters from the Arctic will be better explained in the introduction or in this part of the section, taking care of acknowledging earlier findings.**

*Line 5-6 (Page 11). "while its $^{236}$U/$^{238}$U ratios are likely > 2000 $10^{-12}$ due to GF and unconstrained Arctic rivers inputs". Influencing how? In $^{236}$U, $^{129}$I or both?*

**Earlier studies showed that Pacific-Arctic water arriving to the Labrador Sea carry little $^{129}$I (Ellis and Smith 1999), while the $^{236}$U/$^{238}$U atom ratios are unexpectedly high in the realm of Pacific waters (Casacuberta et al., 2014). Although that source is not well constrained yet, it would appear that rivers might be a source, especially for $^{236}$U.**

*Line 12 (Page 11). "rise of $^{129}$I concentrations at certain depths on the Greenland slope (e.g., station 60; Figure 2 and Figure S1), and particularly in bottom waters of the Irminger Sea (station 44), which*

*are probably related to the cascading of $^{129}$I-rich waters from the Greenland Shelf". And why not an increase in $^{236}$U?*

**Our interpretation is that waters carried by the EGC may cascade from the shelf (station 53 and 61) over the slopes in the eastern (station 60) and western (station 64) sides of the southern tip of Greenland. The EGC carries about 10 times more $^{129}$I (the core presents about 250 10$^7$ at/kg, station 53 and 61, Table 2) than in surrounding off-shore waters (about 20-25 x10$^7$ at/kg, e.g. station 60 and 64). In contrast, the $^{236}$U concentration in the EGC (about 15 x10$^6$ at/kg) is only ½ times higher than in the mentioned off-shore waters. Thus, while the spike of $^{129}$I is easily observed in near bottom depths on the western and eastern slopes of Greenland (stations 60 and 64, Figure 2 and vertical profiles in the supplemental material), such spike is not distinguishable for $^{236}$U.**

*Line 22-23 (Page 11). "The ISOW is best distinguished by its relative $^{129}$I concentration maxima". Explain origin of this maximum.*

**The origin of the maximum will be explained and better understood thanks to a more elaborate background on tracer source and distribution provided in the introduction. $^{236}$U**

*Line 24 (Page 11). The differences can be more clearly seen in Table 2.*

**Table 2 will be referenced.**

*Line 24-25(Page 11). "Further, in the next years one can expect a stronger $^{129}$I signal associated with ISOW in the SPNA due to the releases from the NRPs". Explain this further.*

**A better explanation on the input function of $^{129}$I and its expected temporal evolution will be provided in the manuscript.**

*Line 3 (Page 12). "The evolution of $^{129}$I (and $^{236}$U) in the SPNA is closely related to the effluents discharged from the two European NRPs". It sounds weird to mention this at the end of the paper.*

**This shall be mentioned in the introduction also.**

*Line 18 (Page 12). "Data reported in this study (2014) supports this 'Arctic loop' and suggests that the second 129I front probably peaked before the GEOVIDE cruise". Could Vivo et al. values be also used to support this "Arctic loop"?*

**It is difficult to say given the different location and sampling time of the two studies. Our point of view is that the comparison should be kept to nearby stations measured repeatedly over time to avoid uncertainties related to transit times and water mass mixing. DSOW dilutes 1-2 times during the 0.3-2 years of transport from the Denmark Strait to the Central Labrador Sea (Smith et al., 2005). This makes difficult usingVivo-Vilches et al., (2018) data to confirm the Arctic loop in the context of the Labrador Sea. Vivo-Vilches et al., (2018) present $^{129}$I concentration of about 102 x10$^7$ at/kg in bottom depths at their station 9 in the Denmark Strait, typically occupied by the DSOW core, for samples collected in 2012.**

**\*All references provided in the above answers can be found in the original manuscript.**

---

## Author Comment (AC2) · 27 Jul 2018

Dear Reviewer 2,

We are grateful for your time and for the positive comments on the manuscript. We will gladly address the major and minor points and make the convenient changes.

Please click on the supplemental material to check specific answer to your comments.

On behalf of all co-authors,

Maxi Castrillejo

[Figure]

Please also note the supplement to this comment:
https://www.biogeosciences-discuss.net/bg-2018-228/bg-2018-228-AC2-
supplement.pdf

**Supplement:**

Maxi Castrillejo et al., on behalf of all co-authors.
maxic@phys.ethz.ch

Dear Reviewer 2,

We are grateful for your time and for the positive comments on the manuscript. We will gladly address the major and minor points and make the convenient changes.

On behalf of all co-authors,

Maxi Castrillejo

**Point by point answer to reviewer 2.**

*In general, this article presents new information about two circulation loops of Atlantic Waters which are tagged with nuclear reprocessing plant effluents from their source region based on the observations at stations from Lisbon (Portugal) to the southern tip of Greenland (Cape Farewell), and from Cape Farewell to St. John's (Newfoundland, Canada). The reviewer thinks that this article should be published in Biogeoscience, but there are several points should be revised before publication.*

*Major points:*
*Page 8 line 25 The authors used a binary mixing model of which three end members are LB, GF and NRP. But, as the authors recognized and stated in the text, most of the samples can be explained by simple two end members model except 6 samples collected in the deeper layers (page 9, line 2) which towards the lithogenic background, LB. This means that in the surface to mid depths in this region, to discuss sources of $^{129}$I and $^{236}$U in the SPNA, the reviewer thinks that it is enough to use simple two end members mixing model and the authors can revise the discuss here.*

**The reason why the lithogenic background is included is that despite its small contribution (by mass of radionuclide), it has a very distinct $^{129}$I/$^{236}$U and $^{236}$U/$^{238}$U atom ratios that makes this source clearly distinguishable from the two artificial radionuclide sources (NRP and the GF). Therefore, the third 'natural' source (i.e. Lithogenic background) allows identifying waters that have a very small anthropogenic impact. This is the case of NEADWL, a water mass that has a large presence in the eastern part of the section. Therefore and since it does not harm, we would prefer to keep the lithogenic background in the binary mixing model.**

*Page12 Line 1 -27 The discussion about transit times and dilution factors in the paragraph is poor and difficult to understand how the authors calculate time scales of 8-10 years for shorter loop and 8-18 years for longer loop.*

**We will provide more detailed information regarding water mass transport pathways and time scales in the introduction section and in section 3.4.4. Particular emphasis will be put also on the circulation within the Arctic Ocean (the longer loop) that was not explained in sufficient detail in the original manuscript and which might help on the discussion about the estimated transit times. We shall summarize earlier findings on this matter, making clear what are confirmations of previous findings and what is novel information obtained from this study. For instance, the existence of the two loops has been previously pointed by Smith et al., (2005, 2011 and 2016) and our results confirm such hypothesis. This will be stated clearly.**

*This 8-18 years statement is also inconsistent the numbers stated "between the maximum 16-8 years (page 13 line 19)" in the conclusion.*

**We acknowledge the reviewer for noting such inconsistency. The correct time scales are 8-10 years for the short loop and 16-18 years for the long loop. This will be corrected in the conclusion.**

*The authors used $^{129}I$ input function at 60 N deg. By Christle 2015 and compared observational peak. But the input function already includes several assumptions and based on the figure caption, no explanation in the main text, the authors expanded the function to fit the measurement. But as shown in Figures 4A and 4B, the reviewer observes inconsistency between input functions and observations for both $^{129}I$ and $^{236}U$.*

**We shall provide the assumptions made by Christl et al., (2015) for uranium-236 and for Iodine-129. The inconsistencies between the observed radionuclide values and the input functions will be further discussed briefly in terms of uncertainties related to the source input function and the circulation of water masses.**

*Therefore, the reviewer suggests that the authors can and should collaborate with numerical modeling guys to get modeling results and compared with authors observation.*

**The reviewer is certainly right that model simulations are desirable to better understand the observed tracer levels and their distribution. Indeed, we are exploring/looking for potential collaborations with ocean circulation modellers to simulate the release and transport of $^{129}I$ and $^{236}U$ in the subpolar North Atlantic. Yet, such comparison between experimental and modelling data would be out of the scope of this manuscript. The two main reasons are that the experimental data already provide a very rich information for one manuscript, and that finding a suitable model which can fit the tracer input in a well resolved ocean circulation for the subpolar North Atlantic is apparently not trivial.**

*Minor points:*
*page 2 line 25-29 The authors should add about $^{238}U$ data in their study.*
**The study focuses on the $^{129}I$ and $^{236}U$ which are largely of artificial origin and thus provide transient information about the ocean circulation. Nevertheless, we shall provide data on concentrations of $^{238}U$ (mostly natural) in the supplemental information in case this is of interest for the oceanographic community.**

*Page 5 line 25 and 24 12L Niskin bottles.! and 24 of 12L Niskin bottles.?*
**24 Niskin bottles of 12 L each. The sentence will be clarified accordingly.**

*Page 7 line 8 The authors used data marked * , but the uncertainties are so large for $^{236}U/^{238}U$ ratio $^{129}I/^{236}U$ ratio as 2350+- 370 and 200+-60, respectively. These numbers should be in the blanket ( ), and 2090+-140 and 140+-30 should be used. Due to larger uncertainty, 2350+-370 and 2090+-140 mean within the same and 200+-60 and 140+-30 locate are also within the same.*
**Thank you for noting the need for '()' in some of the data. Change will be done as requested. A small part of the dataset has larger uncertainties for uranium-236. This is because additional corrections had to be made to address limited contamination issues in those samples. We are confident that those data are valid and well represented as long as they are reported with the associated uncertainty.**

*Page 8 line 3 andc1600 x . The reviewer can not understand the meaning of this part. Please clarifiy the meaning of this part.*
**We apologize for this error. 'andc' will be replaced by 'and'.**

*Page 27 Figure4 Caption of Figure 4 is not enough and color coordinations for previous and current date are not good, eg. think open green circle in Fig.4B was hard to find in Fig.4D.*
**The colour and shape of the symbols, as well as figure captions will be revised in order to improve the overall comprehension of the data.**

*Time series data in Fig.4A is also not good to undestand temporal changed of $^{129}I$ concentration.*
**Please see the comment above.**

*In general, all figure captions did not contain enough information about meaning of each color and each mark. Please state more precisely.*
**Please see the comment above.**

*End of comments.*

---

## Author Response (AR1)

Dear Professor Catherine Jeandel,

It is my pleasure to provide the revised manuscript (with and without tracked changes). Please find also attached the detailed answer (in bold) to reviewers' comments (in italics). The latter file indicates the changes made in the revised manuscript with page and line numbering corresponding to the 'track changes' version.

Both reviewers agreed with the major scientific outcomes and with the interpretation of the data. Yet, some changes were suggested, both of more general and specific character. As requested, we have addressed them all. This resulted into a new version of the manuscript, in which new sections have been incorporated and others have been rewritten. Please, see these changes and the response to reviewers in the attached documents.

We hope that after addressing the comments of both reviewers, you will find the manuscript suitable for publication in the GEOVIDE special issue in Biogeosciences.

Maxi Castrillejo, on behalf of all co-authors.
maxic@phys.ethz.ch

Reviewer 1

**We are grateful for the thorough review and the constructive comments from the reviewer.**

**We have addressed most of the changes suggested by the reviewer. Special emphasis has been placed on providing a more detailed introduction on sources/levels of the tracers and their transport and distribution, the ocean circulation of the study area, and also highlighting the objectives. The 'results and discussion' section includes a first, new subsection with a full description of the water mass structure. The overall discussion has been revised to clarify what is new information obtained from these tracers during the GEOVIDE cruise and what are confirmations of earlier tracer/physical studies. All specific comments have also been addressed, as shown in the point by point answer (in bold) to the comments made by the reviewer (in italics).**

**On behalf of all coauthors,**

**Maxi Castrillejo**

Detailed point by point answers to Reviewer 1.

*This manuscript focus on the distribution of $^{129}$I and $^{236}$U along the GEOVIDE section (transect GEOTRACES GA01) in spring 2014. GEOVIDE cruise covered the subpolar North Atlantic Ocean and the Labrador Sea. This manuscript represents an important updated dataset and the authors successfully use $^{129}$I and $^{236}/^{238}$U and $^{236}$U/$^{129}$I atom ratios to describe water masses. The authors confirm with this study the major potential of the combination of $^{129}$I and $^{236}$U as circulation tracers, especially in the area of study and the Arctic Seas and I really enjoyed reading it. However, I think that given that the combined use of $^{129}$I and $^{236}$U provide such rich information, some of the results provided could be discussed in more depth. My impression is that the description and overall use of some the data still require a bit of discussion.*

*If I am not mistaken, the paper have three main objectives that should be emphasized and clarified in the abstract and the introduction.*

*1. Update and improve the database of $^{129}$I and $^{236}$, to be used for future studies and/or modelisation of the ocean circulation in the North Atlantic*

*2. Present new evidences of the advantages of using both radionuclides as dual tracers in the ocean. In this case, what I miss in the text is a more detailed explanation/introduction of why and how $^{236}$U, $^{129}$I and $^{236}/^{129}$I combined provide different and complementary information.*

*The authors reference previous works but should provide the reader with a bit of context and additional information about how these tracers/methodology work.*

*3. Use the tracers to understand ocean circulation in the area. This seems to be the main objective of the paper, however the conclusions from this part are mixed with the other two objectives, together with what is already known and what is novel in this paper. e.g. the final conclusion in the Abstract "Data of $^{129}$I and $^{236}$U from 2014 and the $^{129}$I time series in the Labrador Sea agrees with the hypothesis that Atlantic Waters follow at least two circulation loops from their source region [...] recirculation in the Arctic Eurasian Basin" is not new was already stated by Orre et al. (2010) with $^{129}$I and partially by Povinec et al., (2003) using other radioactive tracers such as $^{137}$Cs. But there is missing information in the abstract to emphasize that the other conclusions are indeed novel, i.e contribution of ISOW to eastern SPNA is quite recent.*

**The objectives of the paper are now clearly presented in both the abstract and the introduction.**

**Abstract (page 2, lines 1-7): 'Pathways and time scales of water mass transport in the Subpolar North Atlantic Ocean (SPNA) have been investigated by many studies due to their importance for Meridional Overturning Circulation and thus for the global ocean. In this sense, observational data on geochemical tracers provide complementary information to improve the current understanding of the circulation in the SPNA. To this end, we present the first simultaneous distribution of artificial $^{129}$I and $^{236}$U in 14 depth profiles and in surface waters along the GEOVIDE section covering a zonal transect through the SPNA in spring 2014.'.**

**Introduction (page 7, lines 1-15): 'In this study, we aim at using artificial $^{129}$I and $^{236}$U to investigate the transport pathways and time scales of water mass circulation in the SPNA. To this end, we present the first simultaneous distribution of $^{129}$I and $^{236}$U along the GEOVIDE cruise track in spring 2014 (Figure 1). The study pursues three specific objectives. Firstly, we study the zonal distribution of $^{129}$I and $^{236}$U and their relationship with the water mass structure. Although the distribution of $^{129}$I in the Irminger and Labrador Seas has been well studied in the last 30 years, there is a significant data gap east of the Reykjanes Ridge for $^{129}$I, and for most of the section for $^{236}$U. Secondly, we use the dual $^{129}$I/$^{236}$U - $^{236}$U/$^{238}$U tracer approach to distinguish the sources contributing to the presence of $^{129}$I and $^{236}$U in the SPNA. This information is then valuable to study the origin, mixing and spreading pathways of water masses participating in the AMOC. The combined use of $^{129}$I and $^{236}$U allows tracing circulation features that received significant attention in earlier modelling, tracer and physical studies, and helps validating recent interpretations on the ventilation of the North Atlantic by overflow waters. Thirdly, tracer data from 2014 are combined with the extensive $^{129}$I time series in the central Labrador Sea to further investigate the circulation time scales of AWs downstream of European NRPs.'.**

*A general comment on the paper is that it presents an impressive dataset and it would be desirable to make more clear which of the conclusions are confirmations of previous hypotheses/results. In the text it is indeed explained, however I think that the novel results, found mainly from the dual use of these radiotracers, are mixed with results that are confirmation of known facts and its relevance it is not explicitly enhanced, which is a shame.*

*Section 3.4 is basically where the novel features of these tracers are presented, in contrast with previous sections that basically use previous data and hypotheses and verify that the new $^{129}$I and $^{236}$U data are in agreement. However, this distinction is, in my opinion, not totally clear especially when presenting section 3.3. Novel and/or on discussion hypotheses reinforced by these dataset should be highlighted. I would also emphasize conclusions obtained by the use of $^{236}$U and $^{129}$I/$^{236}$U, since they are novel tracers and the first time that they are measured simultaneously in the area. However, in this sense I find the Conclusion section very well structured.*

**The discussion has been modified (as shown in specific comments below) in order to clarify which are the novel results and which confirm the hypotheses/results reported in the literature. Special care has been taken to revise section 3.3.**

*Finally, it is assumed in the text that the reader knows well about the ocean circulation in the North Atlantic and Arctic Oceans and about $^{129}$I and $^{236}$U, if this is the case, the paper is quite straightforward to read. But in my opinion one can get easily lost if that is not the case, I have add a few examples of this in the specific comments below.*

*To provide a general background to better understand the discussion of the results I suggest something like:*

*1. Presenting first a brief introduction to ocean circulation and water masses involved with the data.*

**In this revised version, we include a brief introduction to the ocean circulation and water masses involved in the tracer transport. For example:**

 **Page 5, lines 2-17: 'The schematic transport of NRP effluents and water masses in the SPNA-Artic Ocean region is displayed in Figure 1. NRP-labelled AWs are first transported by surface currents into the North Sea and then carried poleward by the Norwegian Coastal Current (NCC) into the Nordic Seas (Edmonds et al., 1998; Raisbeck and Yiou, 2002) while mixing with the Norwegian Atlantic Current (NwAC) (Gascard et al., 2004; Kershaw and Baxter, 1995). The current splits in two branches north of Norway, one branch entering the Barents Sea as Barents Sea Branch Water (BSBW) and the other branch approaching the Fram Strait west of Spitsbergen where it bifurcates again. One branch joins the East Greenland Current (EGC) and recirculates southwards as Return Atlantic Water (RAW) (Fogelqvist et al., 2003) mixing with IrSPMW and PIW (modified AW that has recirculated in the Arctic Ocean; Rudels et al., 1999b). The other branch, the West Spitsbergen Current (WSC), transports the remaining AWs at shallow to intermediate depths into the Arctic Ocean via the Fram Strait Branch Water (FSBW), where they recirculate in the Arctic Eurasian Basin before outflowing back through the Fram Strait and continuing southwards carried by the EGC (Rudels, 2015). The NRP signal also penetrates deep in the water column due to the formation of dense water north of the Greenland-Iceland and Iceland-Scotland passages, providing means of tracing the deep overflows that ventilate the deep North Atlantic Ocean (e.g. Smith et al., 2005).'.**

*2. Explain in more detail the role of $^{129}$I, $^{236}$U and $^{236}$U/$^{129}$I as ocean tracers of the SPNA, making clear what we have learn so far using them i.e. provide context.*

In the revised manuscript we provide more detail on sources which is necessary to understand the values of $^{129}$I, $^{236}$U and $^{236}$U/$^{129}$I. We also explain why the combination of both tracers is important. For example:

Page 5, lines 26-28; and page 6, lines 1-3:'the presence of $^{129}$I in those regions is dominated by the liquid discharge from European NRPs, which has a well-documented release history (> 5700 kg; He et al., 2013a; Raisbeck et al., 1995), while the contribution from GF is comparably negligible (~ 90 kg worldwide release; Hou, 2004; Raisbeck and Yiou, 1999; Wagner et al., 1996). Consequently, the seawater affected by NRPs may present $^{129}$I concentrations 1 – 4 orders of magnitude above the background due to GF (~ $2.5 \times 10^7$ at/kg; Edmonds et al., 1998).'.

Page 6, lines 15-22: 'Surface seawaters of the northern hemisphere present $^{236}$U/$^{238}$U atom ratios of about $1000 \times 10^{-12}$ (e.g. Christl et al., 2012) in the unique presence of GF (about 900 kg released worldwide; Sakaguchi et al., 2009). However, the $^{236}$U/$^{238}$U ratios can be significantly higher in the Arctic and North Atlantic Oceans due to the liquid discharge of $^{236}$U from European NRPs (about 100 kg, Christl et al., 2015a). This has allowed tracing the waters carrying NRP-$^{236}$U with $^{236}$U/$^{238}$U ratios up to $3800 \times 10^{-12}$ in the Arctic Ocean in 2011 – 2012 (Casacuberta et al., 2016), and up to $1400 \times 10^{-12}$ in LSW and DSOW in the western SPNA in 2010 (Casacuberta et al., 2014).'.

Page 6, lines 23-26: In addition, both $^{236}$U and $^{129}$I can be combined as the dual tracer, $^{129}$I/$^{236}$U - $^{236}$U/$^{238}$U, to identify the radionuclide source(s) present in a given water mass (Casacuberta et al., 2016; Christl et al., 2015b). This is possible because the GF and the European NRPs introduced different amounts of $^{236}$U and $^{129}$I (see above) to the environment and tagged the waters with characteristic $^{129}$I/$^{236}$U and $^{236}$U/$^{238}$U atom ratios depending on the proximity from the source(s).

*It would be also good to better explain how to read and understand Figure 3. Which is extremely useful and provides a lot of information.*
We have modified the structure of section 3.3 to include a paragraph that facilitates the interpretation of Figure 4 (former Figure 3). This is achieved by including the expected values of $^{129}$I/$^{236}$U and $^{236}$U/$^{238}$U atom ratios in the subpolar North Atlantic water (page 14, lines 8-15): 'As done in earlier studies (Casacuberta et al., 2016), we can estimate the contribution to our samples from the LB, GF and NRP by combining the $^{129}$I/$^{236}$U and $^{236}$U/$^{238}$U on a dual tracer approach (Figure 4). This is possible because the atom ratios of $^{129}$I/$^{236}$U and $^{236}$U/$^{238}$U display a wide range of values due to the different input of $^{129}$I and $^{236}$U from the three sources. For example, the GF introduced about 10 times more $^{236}$U than $^{129}$I, thus this endmember is characterized by $^{129}$I/$^{236}$U < 1 and $^{236}$U/$^{238}$U surface ratios in the (1000–2000) $\times 10^{-12}$ range. On the contrary, the total amount of $^{236}$U introduced from European NRPs was much smaller than for $^{129}$I. Therefore, a water mass with the additional influence from the European NRP may present $^{129}$I/$^{236}$U on the 1–350 range and $^{236}$U/$^{238}$U above the GF.'.

We also explain with an example what it means to fall in one part or the other of the chart (page 14, lines 22- 23): 'For instance, a sample falling in the 1 % value on the GF-NRP

**binary mixing line would be composed of waters carrying largely GF and about 1 % of the NRPs signal.'.**

*ABSTRACT*
*1) I think that these lines "Results show that part of the effluents discharged from Sellafield and La Hague apparently enter the eastern SPNA directly through the Iceland-Scotland passage or the English Channel/Irish Sea, as it is shown by elevated $^{129}I$ concentrations and $^{129}I/^{236}U$ ratios in shallow central waters flowing in the West European Basin (WEB)" are saying the same than these ones "The Iceland-Scotland Overflow Water spreading pathways into the eastern SPNA have been confirmed by the unequivocal transport of reprocessing $^{129}I$ into the deep WEB".*

**The abstract has been re-written to clarify, among other things, that the signals from the reprocessing plant may follow alternative pathways, not only through overflow waters, but also on the surface through the Iceland-Scotland passage. For example (page 2, lines 15-17): 'Nevertheless, our results show that the effluents from NRPs may also directly enter the surface eastern SPNA through the Iceland-Scotland passage or the English Channel/Irish Sea.'.**

*2) When it is said "The Iceland-Scotland Overflow Water spreading pathways into the eastern SPNA have been confirmed by the unequivocal transport of reprocessing $^{129}I$ into the deep WEB", it should be briefly explained why we find this transport unequivocal.*

**It has been briefly explained in the abstract (page 2, lines 24-27): 'Several depth profiles also show an increase in $^{129}I$ concentrations in near bottom waters in the Iceland and the West European Basins that are very likely associated to the transport of the NRP signal by the Iceland-Scotland Overflow Water (ISOW). This novel result would support current modelling studies indicating the transport of ISOW into the eastern SPNA.'.**

*INTRODUCTION*
*3) When one reads from lines 15 (Page 3) to line 23 (Page 4) gets a very general idea about how $^{129}I$ and $^{236}U$ are distributed in the North Atlantic, but do not get a precise picture of what are the paths followed by the radionuclides when released by the RP. That information is given later in the text, the problem is that it is scattered in different sections of the manuscript.*

**In the revised version, we explain the precise path(s) followed by the radionuclides when released by the RP (page 5, lines 2-17): 'The schematic transport of NRP effluents and water masses in the SPNA-Artic Ocean region is displayed in Figure 1. NRP-labelled AWs are first transported by surface currents into the North Sea and then carried poleward by the Norwegian Coastal Current (NCC) into the Nordic Seas (Edmonds et al., 1998; Raisbeck and Yiou, 2002) while mixing with the Norwegian Atlantic Current (NwAC) (Gascard et al., 2004; Kershaw and Baxter, 1995). The current splits in two branches north of Norway, one branch entering the Barents Sea as Barents Sea Branch Water (BSBW) and the other branch approaching the Fram Strait west of Spitsbergen where it bifurcates again. One branch joins the East Greenland Current (EGC) and recirculates southwards as Return Atlantic Water (RAW) (Fogelqvist et al., 2003) mixing with IrSPMW and PIW (modified AW that has recirculated in the Arctic Ocean; Rudels et al., 1999b). The other branch, the West Spitsbergen Current (WSC), transports the remaining AWs at shallow to intermediate depths into the Arctic Ocean via the Fram Strait Branch Water (FSBW), where they**

recirculate in the Arctic Eurasian Basin before outflowing back through the Fram Strait and continuing southwards carried by the EGC (Rudels, 2015). The NRP signal also penetrates deep in the water column due to the formation of dense water north of the Greenland-Iceland and Iceland-Scotland passages, providing means of tracing the deep overflows that ventilate the deep North Atlantic Ocean (e.g. Smith et al., 2005).'.

*4) Furthermore, lines 15 to 20 (Page 4) provides some information about previous results of $^{236}U/^{129}I$ however it does not explain what these numbers represent or why and how they change geographically or in time. For example, it is not explained why "Yet, LSW and DSOW were clearly identified by $^{236}U/^{238}U > 1000 \times 10^{-12}$"; or why the atom ratio varies from "$^{129}I/^{238}U < 1$ for GF to about 1 - 350 for European NRPs".*

**This is now better explained by providing more information on sources, oceanic levels and the usefulness of using these tracers alone and in combination. For example:**

**Page 5, lines 26-28; and page 6, lines 1-3:'the presence of $^{129}I$ in those regions is dominated by the liquid discharge from European NRPs, which has a well-documented release history (> 5700 kg; He et al., 2013a; Raisbeck et al., 1995), while the contribution from GF is comparably negligible (~ 90 kg worldwide release; Hou, 2004; Raisbeck and Yiou, 1999; Wagner et al., 1996). Consequently, the seawater affected by NRPs may present $^{129}I$ concentrations 1 – 4 orders of magnitude above the background due to GF (~ $2.5 \times 10^7$ at/kg; Edmonds et al., 1998).'.**

**Page 6, lines 15-22: 'Surface seawaters of the northern hemisphere present $^{236}U/^{238}U$ atom ratios of about $1000 \times 10^{-12}$ (e.g. Christl et al., 2012) in the unique presence of GF (about 900 kg released worldwide; Sakaguchi et al., 2009). However, the $^{236}U/^{238}U$ ratios can be significantly higher in the Arctic and North Atlantic Oceans due to the liquid discharge of $^{236}U$ from European NRPs (about 100 kg, Christl et al., 2015a). This has allowed tracing the waters carrying NRP-$^{236}U$ with $^{236}U/^{238}U$ ratios up to $3800 \times 10^{-12}$ in the Arctic Ocean in 2011 – 2012 (Casacuberta et al., 2016), and up to $1400 \times 10^{-12}$ in LSW and DSOW in the western SPNA in 2010 (Casacuberta et al., 2014).'.**

**Page 6, lines 23-26: In addition, both $^{236}U$ and $^{129}I$ can be combined as the dual tracer, $^{129}I/^{236}U$ - $^{236}U/^{238}U$, to identify the radionuclide source(s) present in a given water mass (Casacuberta et al., 2016; Christl et al., 2015b). This is possible because the GF and the European NRPs introduced different amounts of $^{236}U$ and $^{129}I$ (see above) to the environment and tagged the waters with characteristic $^{129}I/^{236}U$ and $^{236}U/^{238}U$ atom ratios depending on the proximity from the source(s).**

*5) Lines 10-15 (Page 4). Why reference data are given here and not for $^{129}I$?*
**In the revised version we provide reference data also for I-129. For example;**

**Page 6, lines 1-3: 'the seawater affected by NRPs may present $^{129}I$ concentrations 1 – 4 orders of magnitude above the background due to GF (~ $2.5 \times 10^7$ at/kg; Edmonds et al., 1998).'.**

*6) Line 18 (Page 5). No mention to deep water formation at the Greenland Sea? And ISOW formation? ISOW is later described (Line 7, Page 8), but it would be easier to follow the manuscript having the whole picture since the beginning.*

**This is now described in the text of the introduction (page 5, lines 14-17): 'The NRP signal also penetrates deep in the water column due to the formation of dense water north of the Greenland-Iceland and Iceland-Scotland passages, providing means of tracing the deep overflows that ventilate the deep North Atlantic Ocean (e.g. Smith et al., 2005).'.**

*SECTION 3.1.* **(Now section 3.2)**
*7) Line 5 (Page 7). A brief introduction to $^{129}I/^{236}U$ ratios is missing to understand their values and the further discussions.*
**The revised introduction includes more detailed information on sources and oceanic levels of these tracers, which then used to introduce that ratio:'**

**Page 5, lines 26-28; and page 6, lines 1-3: 'the presence of $^{129}I$ in those regions is dominated by the liquid discharge from European NRPs, which has a well-documented release history (> 5700 kg; He et al., 2013a; Raisbeck et al., 1995), while the contribution from GF is comparably negligible (~ 90 kg worldwide release; Hou, 2004; Raisbeck and Yiou, 1999; Wagner et al., 1996). Consequently, the seawater affected by NRPs may present $^{129}I$ concentrations 1 – 4 orders of magnitude above the background due to GF (~ $2.5 \times 10^7$ at/kg; Edmonds et al., 1998).'**

**Page 6, lines 15-22: 'Surface seawaters of the northern hemisphere present $^{236}U/^{238}U$ atom ratios of about $1000 \times 10^{-12}$ (e.g. Christl et al., 2012) in the unique presence of GF (about 900 kg released worldwide; Sakaguchi et al., 2009). However, the $^{236}U/^{238}U$ ratios can be significantly higher in the Arctic and North Atlantic Oceans due to the liquid discharge of $^{236}U$ from European NRPs (about 100 kg, Christl et al., 2015a). This has allowed tracing the waters carrying NRP-$^{236}U$ with $^{236}U/^{238}U$ ratios up to $3800 \times 10^{-12}$ in the Arctic Ocean in 2011 – 2012 (Casacuberta et al., 2016), and up to $1400 \times 10^{-12}$ in LSW and DSOW in the western SPNA in 2010 (Casacuberta et al., 2014).'.**

**Page 6, lines 23-26: 'In addition, both $^{236}U$ and $^{129}I$ can be combined as the dual tracer, $^{129}I/^{236}U$ - $^{236}U/^{238}U$, to identify the radionuclide source(s) present in a given water mass (Casacuberta et al., 2016; Christl et al., 2015b). This is possible because the GF and the European NRPs introduced different amounts of $^{236}U$ and $^{129}I$ (see above) to the environment and tagged the waters with characteristic $^{129}I/^{236}U$ and $^{236}U/^{238}U$ atom ratios depending on the proximity from the source(s).'.**

**In addition, we have modified the structure of section 3.3 to include a paragraph that facilitates the interpretation of Figure 4 (former Figure 3). This is achieved by including the expected values of $^{129}I/^{236}U$ and $^{236}U/^{238}U$ atom ratios in the subpolar North Atlantic water (page 14, lines 8-15): 'As done in earlier studies (Casacuberta et al., 2016), we can estimate the contribution to our samples from the LB, GF and NRP by combining the $^{129}I/^{236}U$ and $^{236}U/^{238}U$ on a dual tracer approach (Figure 4). This is possible because the atom ratios of $^{129}I/^{236}U$ and $^{236}U/^{238}U$ display a wide range of values due to the different input of $^{129}I$ and $^{236}U$ from the three sources. For example, the GF introduced about 10 times more $^{236}U$ than $^{129}I$, thus this endmember is characterized by $^{129}I/^{236}U$ < 1 and**

**$^{236}$U/$^{238}$U surface ratios in the (1000–2000) × 10$^{-12}$ range. On the contrary, the total amount of $^{236}$U introduced from European NRPs was much smaller than for $^{129}$I. Therefore, a water mass with the additional influence from the European NRP may present $^{129}$I/$^{236}$U on the 1–350 range and $^{236}$U/$^{238}$U above the GF.**'.

*8) Line 10-30 (Page 7). I also miss a complete introduction to water mass structure. It will be easier to follow the discussion if first we understand water mass structure and then 129I and 236U/238U are given.*

*This way, ISOW description (Lines 7 -11, Page 8) should be move to that introduction, and merge with description in Page 5.*

**The discussion now begins with a new section 3.1 (pages 10 and 11), which describes the structure of water masses:**

**'3.1 Water mass structure in 2014**

[revised manuscript text omitted]

This section is supported by a new figure (Figure 3 in page 38) presenting the distribution of hydrographic properties:

[Figure]

**Figure 2. Vertical distribution of (A) salinity, (B) potential temperature and (C) dissolved oxygen along the GEOVIDE section in spring 2014. Water mass acronyms are defined in Table 1.'**

9) Line 26 (Page 7). "SAIW probably incorporates [129]I from precursor water masses (e.g., waters carried by LC and/or LSW) while forming in the western SPNA". Why is that? Some of the statements, like this one, are properly given but not explained in terms of [129]I (or [236]U/[238]U) values.

In the revised manuscript we explain better that LC carries a large NRP signal, that SAIW forms in the LC and then is transported into the GEOVIDE transect, resulting in a large tracer signal due to the influence from NRPs. For example:

Page 12, lines 15-18: 'The highest $^{129}$I concentrations ($\sim 250 \times 10^7$ at/kg) and $^{236}$U/$^{238}$U ratios ($\sim 2300 \times 10^{-12}$) are present in PIW and RAW carried by the EGC and LC over the shelves and slopes of Greenland and Canada. This water admixture, largely influenced by NRPs (e.g. Alfimov, 2004), …'.

Page 11, lines 3-6: 'SAIW forms in the western boundary of the SPNA (i.e. the LC) by mixing between LSW and subtropical waters carried by the NAC (Arhan, 1990; Read, 2000), before subducting at about 400 m depth and being advected within the northern branch of NAC (Figure 2).'.

Page 13, lines 2-3: 'SAIW also presents relatively high $^{129}$I concentrations ($\sim 20 \times 10^7$ at/kg) at stations 26 and 32, probably because of the influence of waters carried by the LC.'.

*10) Line 5 (Page 8). "Thus, 2014 data probably reflects the dilution with old LSW and SPMW carrying less $^{129}$I and $^{236}$U than MW". How is it that waters from LSW and SPMW, both affected by NFRP, carry less $^{129}$I and $^{236}$U/$^{238}$U than MW, also mainly affected by GF? Is it the influence of Marcule?*
Yes. The $^{129}$I concentrations and $^{236}$U/$^{238}$U atom ratios are higher than expected from the GF in the Mediterranean Sea. Recent work showed that this is very likely due to the discharge of $^{129}$I and $^{236}$U from the Marcoule reprocessing plant (see Castrillejo et al., 2017, Science of the Total Environment). The sentence now reads (page 13, lines 6-11): 'Similar depths are also influenced by MW (stations 1 and 13), yet, its $^{129}$I concentrations ($\sim 3 \times 10^7$ at/kg) and $^{236}$U/$^{238}$U ratios ($\sim 1000 \times 10^{-12}$) in 2014 were significantly lower than average $^{129}$I concentrations ($9 \times 10^7$ at/kg) and $^{236}$U/$^{238}$U ratios ($1600 \times 10^{-12}$) reported in the outflow region of MW at the Strait of Gibraltar in 2013 (Castrillejo et al., 2017). Thus, 2014 data probably reflects the dilution of MW, which is largely affected by inputs from the Marcoule nuclear facility (Castrillejo et al., 2017), with old LSW and SPMW carrying a diluted NRP signal.'.

*11) Lines 1-13 (Page 8). This is clearly explained, but it will be even easier to follow if the name of stations and references to Table 2 are given.*
The reader is referred to Table 2 and station numbers throughout section 3.2 (pages 12 and 13).

*12) Lines 14 -18. As already said, previous brief introduction to the use of $^{129}$I/$^{236}$U as tracer should be included to make these lines easier to follow. This way it said "The highest $^{129}$I/$^{236}$U ratios (> 100) are present in waters transported by the shallow EGC and LC. Overflow waters are also distinguishable by their relatively high $^{129}$I/$^{236}$U ratios (60 to 110 for DSOW, 15 to 40 for ISOW)" Why is that?*
That is because they have a greater contribution from the European nuclear fuel reprocessing plants than other waters that are only affected by global radioactive fallout.

**This kind of sentences are now easier to follow, since a more detailed information on sources and transport of the tracers and their use is provided in the Introduction section.**

*SECTION 3.2.*
*13) Line 25- 30 (Page 8). I really like Figure 3. I contains lots of information, may be it could be further explained in the mentioned intro introducing the $^{129}I/^{236}U$ tracer?*
**As previously stated, in the revised version we include further explanation on the sources which helps better understanding the values found for the $^{129}I/^{236}U$ ratio.**

**We have also modified the structure of section 3.3 to include a paragraph that facilitates the interpretation of Figure 4 (former Figure 3). This is achieved by including the expected values of $^{129}I/^{236}U$ and $^{236}U/^{238}U$ atom ratios in the subpolar North Atlantic water (page 14, lines 8-15): 'As done in earlier studies (Casacuberta et al., 2016), we can estimate the contribution to our samples from the LB, GF and NRP by combining the $^{129}I/^{236}U$ and $^{236}U/^{238}U$ on a dual tracer approach (Figure 4). This is possible because the atom ratios of $^{129}I/^{236}U$ and $^{236}U/^{238}U$ display a wide range of values due to the different input of $^{129}I$ and $^{236}U$ from the three sources. For example, the GF introduced about 10 times more $^{236}U$ than $^{129}I$, thus this endmember is characterized by $^{129}I/^{236}U < 1$ and $^{236}U/^{238}U$ surface ratios in the $(1000–2000) \times 10^{-12}$ range. On the contrary, the total amount of $^{236}U$ introduced from European NRPs was much smaller than for $^{129}I$. Therefore, a water mass with the additional influence from the European NRP may present $^{129}I/^{236}U$ on the 1–350 range and $^{236}U/^{238}U$ above the GF.'.**

*SECTION 3.3.*
*14) Line 14 (Page 9). "$^{129}I$ discharge rate from European NRPs was observed in the whole water column, being more pronounced (about 10 times increase) in overflow waters". This actually an previously observed fact but an explanation should be given here.*
**Some parts of this section 3.4 (former section 3.3) have been rewritten to emphasize that data from the GEOVIDE cruise in 2014 is compared to the literature (e.g., page 15, lines 13-14):' In this section we compare radionuclide concentrations reported in the literature with those measured at nearby stations during GEOVIDE (Figure 5).'.**

**In addition, the section about I-129 begins with a summary of previous findings (e.g., page 15, lines 16-22): 'In the case of $^{129}I$, the existing time series for the central Labrador Sea (1993–2013, Figure 5A) demonstrated that most of the tracer transport was carried by overflow waters (e.g. DSOW) and that the temporal evolution of $^{129}I$ concentrations in those waters could be associated with the tracer release from the European NRPs some years earlier (Edmonds et al., 2001; Orre et al., 2010; Smith et al., 2005, 2016). For instance, the literature on $^{129}I$ shows a rise in tracer concentrations due to the increased $^{129}I$ discharge rate from European NRPs in the whole water column, being more pronounced (about 10 times increase) in overflow waters (Figure 5A).'.**

*15) Figure 4A. Indicate in the caption that Smith 2016 corresponds to 2012 and 2013 profiles. "The depth distribution of $^{129}I$ concentrations in the Labrador Sea in 2014 (station 69), displays $^{129}I$ concentrations in DSOW about 15 % lower than in 2012 – 2013 (Smith et al., 2016)". Is this because samples from 2012-2013 are measuring the peak in the NFRP*

*releases? If this is the case, please mention that the explanation for that decrease will be given in Section 3.5.*

**The figure caption has been amended to reflect the source of the data (page 41, lines 1-10).**

**Yes, the interpretation is that the 2014 data suggests that the peak occurred in 2013. As suggested by the reviewer, we refer to this in the text (page 15, lines 23-24): 'The depth distribution of [129]I concentrations in the Labrador Sea in 2014 (station 69) displays [129]I concentrations in DSOW about 15 % lower (see section 3.5.4) than in 2012 – 2013 (Smith et al., 2016),'.**

**We give the explanation in page 20, lines 22-24: 'Thus, 2014 data reported in this study supports the current interpretation on the 'Arctic loop' (e.g. Smith et al., 2016) and suggests that the second [129]I front probably peaked before the GEOVIDE cruise.'.**

*16) As I said, it is a well-known fact DSOW present an increase in [129]I concentrations for all years. This is already approached by previous works, but a brief discussion could be also given here.*

**Please see answer to comment 14.**

*17) Line 18 (Page 9). "The main difference between the [129]I depth profiles in the Irminger Sea (station 44) and central Labrador Sea (station 69) in 2014 is the surface [129]I peak in the latter one (Figure 4A). Which is probably caused by waters that split off from the boundary currents, either the West Greenland Current or the LC". I don't quite understand this. Splitting won't change [129]I concentrations.*

**The EGC and the LC are characterised by particularly high [129]I and [236]U concentrations, respectively. We propose that the surface waters in the central Labrador Sea may have been influenced by waters that separated from the mainstream of the WGC/LC. Indeed, such surface peak is also observed in the profile of U-236 concentrations represented in Figure S1. The sentence has been changed to clarify the interpretation (page 15, lines 27 and page 16, lines 1-4): 'Considering the 30 m deep freshwater surface layer observed between station 69 and Greenland (not shown here), we suggest that waters carried by the West Greenland Current (continuation of the EGC) may have separated from the main western boundary transport and entered the Central Labrador Sea (Cuny and Rhines, 2002). This can also explain the peak in [236]U/[238]U ratios observed at the same location (Figure S1).'**

*18) Line 26 (Page 9). "This similarity suggests little time variation and similar water mass composition for that region, although PAP might present slightly larger [129]I concentrations because of its proximity to Sellafield and La Hague". And will support the later mentioned hypothesis of direct contribution of NFRP to SPNA without previous recirculation (Line 10, Page 10).*

**Indeed, this is probably the case. We have added the reviewers' point in the revised text (page 15, lines 15-18): 'These results suggest a similar water mass composition for that region, yet the offset in deep [129]I concentrations would support the hypothesis that effluents from the nearby Sellafield and/or La Hague NRPs may enter directly into the SPNA without previous circulation in the Nordic Seas (see section 3.5.1).'.**

*SECTION 3.4.* **(now section 3.5.1)**
*19) Line 17 (Page 10). "twice" instead of "two times"*
**This has been changed (page 17, line 17).**

*20) Line 16 -17 (Page 10). "near-surface transport of $^{129}I$ from European NRPs also across Iceland-Scotland into the eastern SPNA" is also clearly seen in Table 2. That shows that profiles 1, 13 and 21 strongly contrast from profiles 26 and 32. Not only due to ISOW (IcSPMW) contribution in intermediate depths but also at shallower depths.*
**The station numbers and Table 2 have been referenced (page 17, lines 19-20): 'Consequently, $^{129}I$ concentrations in shallow waters at stations 1, 13 and 21 strongly contrast with those at stations 26 and 32 located west of the SAF (Table 2)'.**

*21) Line 27 (Page 10). allowing to identify key circulation features such as the EGC/LC and the DWBC in the Labrador and Irminger Seas. Explain in terms of radioactive tracers.*
**This comment and the following ones (22 and 23) have been addressed by: i) adding a brief introduction to the section on the circulation of the EGC and LC and the subsequent supply of waters of Atlantic and Canada origin; and ii) discussing their different tracer levels (page 18, lines 2-17): 'It is well known that the eastern coast of Greenland receives RAW and PIW-Atlantic injected in the EGC (Figure 1). Similarly, the shelf of Newfoundland is bathed by the LC, which carries EGC waters and PIW-Canada, this last one being supplied through the Nares Strait (Curry et al., 2014). The tracer levels are particularly high in such waters residing on the shelves, slopes and very deep waters around Greenland and Newfoundland (Figure 3). Further, the tracer content differs between Arctic waters of Atlantic and Canadian origin enriched in $^{129}I$ and $^{236}U$, respectively. Thus, one may use them to distinguish key circulation features such as the EGC/LC and the DWBC in the Labrador and Irminger Seas (Figure 1). For example, at shallow depths, the EGC (stations 53 to 64) presents remarkably high $^{129}I$ concentrations (up to ~ $250 \times 10^7$ at/kg) and $^{129}I/^{236}U$ ratios (up to 200), while both values are significantly lower in the LC (station 78) which is characterized by comparably higher $^{236}U/^{238}U$ ratios (up to $2350 \times 10^{-12}$) (Figure 3). Such differences on the $^{129}I$ and $^{236}U$ composition of the two shallow boundary currents are likely due to the fact that waters of Atlantic origin (PIW-Atlantic and RAW) have been largely influenced by NRP effluents (high $^{129}I$). On the contrary, the LC records lower $^{129}I$ concentrations due to the influence of PIW-Canada waters with mainly GF signal (Ellis and Smith, 1999; Smith et al., 1998), and a large $^{236}U$ content ($^{236}U/^{238}U$ ratios are likely > $2000 \times 10^{-12}$) from both the GF and unconstrained Arctic rivers inputs (Casacuberta et al., 2016).'.**

*22) Line 30 -30. Differences of $^{129}I$ and $^{236}U$ in boundary currents are mentioned but not explained. It should be further discussed in terms of radioactive tracers.*
**Please see answer to comment 21.**

*23) Line 1-2 (Page 11). "EGC shows particularly high $^{129}I$ concentrations and $^{129}I/^{236}U$ ratios because it is carrying Arctic water of Atlantic origin (PIW-Atlantic) and RAW that have been largely influenced by NRP effluents". I assume the authors do not explain this further because this is well known from previous works. Nevertheless, a brief description should be given, may be in the previously mentioned introduction?*

**Please see answer to comment 21.**

*24) Line 5-6 (Page 11). "while its $^{236}U/^{238}U$ ratios are likely > 2000 $10^{-12}$ due to GF and unconstrained Arctic rivers inputs". Influencing how? In $^{236}U$, $^{129}I$ or both?*
**Previous studies showed that the Arctic-Canada water arriving to the Labrador Sea carry low concentrations of $^{129}I$ (Ellis and Smith, 1999), while the $^{236}U/^{238}U$ atom ratios are unexpectedly high for those waters (Casacuberta et al., 2014). Although that source is not well constrained yet, it would appear that Arctic rivers might be a source, especially for $^{236}U$.**

**This is now explained in page 22, lines 1-4: 'For example, the LC presents mainly the GF signal and unconstrained Arctic river inputs (more $^{236}U$ relative to $^{129}I$) indicating the contribution from PIW-Canada through the Canadian Archipelago, while the EGC, largely influenced by the NRPs (more $^{129}I$ relative to $^{236}U$), indicates the contribution of RAW and PIW-Atlantic.'.**

*25) Line 12 (Page 11). "rise of $^{129}I$ concentrations at certain depths on the Greenland slope (e.g., station 60; Figure 2 and Figure S1), and particularly in bottom waters of the Irminger Sea (station 44), which are probably related to the cascading of $^{129}I$-rich waters from the Greenland Shelf". And why not an increase in $^{236}U$?*
**Our interpretation is that waters carried by the EGC may cascade from the continental shelf (station 53 and 61) over the slopes in the eastern (station 60) and western (station 64) sides of the southern tip of Greenland. The EGC transports about 10 times more $^{129}I$ (the core presents about 250 x $10^7$ at kg$^{-1}$, station 53 and 61, Table 2) than in surrounding offshore waters (about 20-25 x $10^7$ at kg$^{-1}$, e.g. station 60 and 64). In contrast, $^{236}U$ concentrations in the EGC (about 15 x $10^6$ at kg$^{-1}$) are only 50% higher than in the mentioned offshore waters. Thus, while the spike of $^{129}I$ is easily observed near the bottom on the western and eastern slopes of Greenland (stations 60 and 64, Figure 2 and vertical profiles in the supplemental material), such increase is not distinguishable for $^{236}U$.**

*26) Line 22-23 (Page 11). "The ISOW is best distinguished by its relative $^{129}I$ concentration maxima". Explain origin of this maximum.*
**The origin of the maximum has been explained in page 19, lines 8-9: 'The ISOW is best distinguished by its relative $^{129}I$ concentration maxima and $^{129}I/^{236}U$ ratios of 15 – 40 due to NRPs,'.**

*27) Line 24 (Page 11). The differences can be more clearly seen in Table 2.*
**Table 2 has been referenced (page 19, line 11).**

*28) Line 24-25(Page 11). "Further, in the next years one can expect a stronger $^{129}I$ signal associated with ISOW in the SPNA due to the releases from the NRPs". Explain this further.*
**A better explanation is now provided (page 19, lines 14-17): 'In the comming years, one can expect a stronger $^{129}I$ signal carried by ISOW because tracer concentrations in ISOW precursor waters have increased from $7 \times 10^7$ at/kg to $63 \times 10^7$ at/kg at the Iceland–**

**Scotland Sill from 1993 to 2012 in response to releases from the NRPs (Alfimov et al., 2004; Edmonds et al., 2001; Vivo-Vilches et al., 2018).'.**

*29) Line 3 (Page 12). "The evolution of $^{129}$I (and $^{236}$U) in the SPNA is closely related to the effluents discharged from the two European NRPs". It sounds weird to mention this at the end of the paper.*

**This is now mentioned in the Introduction and particularly when the temporal evolution of I-129 is discussed in section 3.4 (e.g. page 15, lines 16-20): 'In the case of $^{129}$I, the existing time series for the central Labrador Sea (1993–2013, Figure 5A) demonstrated that most of the tracer transport was carried by overflow waters (e.g. DSOW) and that the temporal evolution of $^{129}$I concentrations in those waters could be associated with the tracer release from the European NRPs few years earlier (Edmonds et al., 2001; Orre et al., 2010; Smith et al., 2005, 2016).'.**

*30) Line 18 (Page 12). "Data reported in this study (2014) supports this 'Arctic loop' and suggests that the second 129I front probably peaked before the GEOVIDE cruise". Could Vivo et al. values be also used to support this "Arctic loop"?*

**It is difficult to do so, given the different locations and sampling times of the two studies. We think that the comparison should be kept to nearby stations measured repeatedly over time to avoid uncertainties related to transit times and mixing of water mass. For instance, the DSOW dilutes 1 - 2 times during the 0.3 – 2.0 years of transport from the Denmark Strait to the Central Labrador Sea (Smith et al., 2005).**

Reviewer 2,
*In general, this article presents new information about two circulation loops of Atlantic Waters which are tagged with nuclear reprocessing plant effluents from their source region based on the observations at stations from Lisbon (Portugal) to the southern tip of Greenland (Cape Farewell), and from Cape Farewell to St. John's (Newfoundland, Canada). The reviewer thinks that this article should be published in Biogeoscience, but there are several points should be revised before publication.*

*Major points:*
*1) Page 8 line 25 The authors used a binary mixing model of which three end members are LB, GF and NRP. But, as the authors recognized and stated in the text, most of the samples can be explained by simple two end members model except 6 samples collected in the deeper layers (page 9, line 2) which towards the lithogenic background, LB. This means that in the surface to mid depths in this region, to discuss sources of $^{129}$I and $^{236}$U in the SPNA, the reviewer thinks that it is enough to use simple two end members mixing model and the authors can revise the discuss here.*

**We acknowledge that using a two end-member model could render similar results in most instances. Yet, the reason for including the lithogenic background is that, despite its small contribution (by mass of radionuclide), it has a very distinct $^{129}$I/$^{236}$U and $^{236}$U/$^{238}$U atom ratio that makes this source clearly distinguishable from the two artificial radionuclide sources (NRP and the GF). Therefore, the third (natural) source (i.e. Lithogenic background) allows identifying waters that have a very small anthropogenic impact. This is the case of NEADW$_L$, a water mass that has a large presence in the eastern part of the**

**section. Therefore, we prefer to keep the lithogenic background in the binary mixing model.**

*2) Page12 Line 1 -27 The discussion about transit times and dilution factors in the paragraph is poor and difficult to understand how the authors calculate time scales of 8-10 years for shorter loop and 8-18 years for longer loop.*
**This section 3.5.4 (pages 19-21) has been rewritten to better explain: i) what was known prior to this study about the circulation loops of Atlantic Waters and their time scales, ii) what are the calculations we have made and how they are done, and, iii) the reasons for possible inconsistencies. We also improved the caption of Figure 6 (former figure 5) to facilitate understanding the interpretation of the data. We believe that the discussion in now clearer for the reader.**

*3) This 8-18 years statement is also inconsistent the numbers stated "between the maximum 16-8 years (page 13 line 19)" in the conclusion.*
**We thank the reviewer for noting such inconsistency. The correct time scales are now stated in page 22, lines 13-15: ' This study supports the current interpretations on the circulation of AWs, which apparently follow a short loop trough the Nordic Seas (8 – 10 years in this study) and a longer loop including the recirculation in the Arctic Eurasian Basin (16 – 18 years in this study).'.**

*4) The authors used $^{129}$I input function at 60 N deg. By Christl 2015 and compared observational peak. But the input function already includes several assumptions and based on the figure caption, no explanation in the main text, the authors expanded the function to fit the measurement. But as shown in Figures 4A and 4B, the reviewer observes inconsistency between input functions and observations for both $^{129}$I and $^{236}$U.*
**We provide the assumptions made by Christl et al. (2015) which followed previous modelling studies using tracer inputs from the European NRPs. For example (page 20, lines ): 'we took the $^{129}$I input function at 60 °N for the northern North Sea (Figure 6A, green dashed line) used in earlier studies (e.g. Christl et al., 2015b; Orre et al., 2010; Smith et al., 2005). This input function is estimated assuming that the signal of both NRPs mixes in the North Sea and then is advected to 60 °N in 2 years from La Hague and in 4 years from Sellafield.'.**

**The inconsistencies between measured and estimated U-236 concentrations are are briefly mentioned in page 21, lines 5–10: 'Although the $^{236}$U data would agree with the hypothesis of a second delayed $^{129}$I pulse arriving from the Arctic Ocean, there are significant inconsistencies between the simulated and measured concentrations. These might be attributed to, among other factors, the large uncertainty of the used $^{236}$U data point for 2014, uncertainties on the amount released by the Sellafield NRP (Christl et al., 2015), missing information on other sources, or unaccounted features on the water mass circulation downstream NRP.'.**

*5) Therefore, the reviewer suggests that the authors can and should collaborate with numerical modeling guys to get modeling results and compared with authors observation.*
**The reviewer is certainly right that model simulations are desirable to better understand the observed tracer levels and their distribution. Indeed, we initiated collaborations with**

ocean circulation modellers to understand better the release and transport of $^{129}$I and $^{236}$U in the subpolar North Atlantic. Yet, such modelling studies are highly complex and the results not at the stage of being ready for publication. Furthermore, the focus of this manuscript is the presentation and interpretation of the measured data. A model vs data study would certainly be out of the scope of this manuscript. The two main reasons are that the experimental data already provide a very rich information for one manuscript, and that finding a suitable model which can fit the tracer input in a well resolved ocean circulation for the subpolar North Atlantic is apparently not trivial.

*Minor points:*
*6) page 2 line 25-29 The authors should add about $^{238}$U data in their study.*
**Now, we provide the $^{238}$U in a supplemental table.**

*7) Page 5 line 25 and 24 12L Niskin bottles.! and 24 of 12L Niskin bottles.?*
**The sentence has been clarified (page 8, line 1): ' 24 Niskin bottles of 12L each'.**

*8) Page 7 line 8 The authors used data marked \* , but the uncertainties are so large for $^{236}$U/$^{238}$U ratio $^{129}$I/$^{236}$U ratio as 2350+- 370 and 200+-60, respectively. These numbers should be in the blanket ( ), and 2090+-140 and 140+-30 should be used. Due to larger uncertainty, 2350+-370 and 2090+-140 mean within the same and 200+-60 and 140+-30 locate are also within the same.*
**We think the symbols '(' and ')' are used correctly in the original manuscript (now page 12, lines 1-7). A small part of the dataset has large uncertainties for U-236. This is because additional corrections had to be made to address limited contamination issues in those samples. We are confident that these data are valid and well represented as long as they are reported with the appropriate uncertainty.**

*9) Page 8 line 3 andc1600 x . The reviewer can not understand the meaning of this part. Please clarifiy the meaning of this part.*
**The error has been corrected in page 13, line 8: 'average $^{129}$I concentrations ($9 \times 10^7$ at/kg) and $^{236}$U/$^{238}$U ratios ($1600 \times 10^{-12}$) reported in'.**

*10) Page 27 Figure4 Caption of Figure 4 is not enough and color coordinations for previous and current date are not good, eg. think open green circle in Fig.4B was hard to find in Fig.4D.*
**We have improved the Figure by: i) enlarging symbols in Figure 5D; ii) changing the colours in Figure 5A, iii) adding sampling years in Figures A, B and C; and iv) by further explaining the figure caption (page 41, lines 2-10): 'Figure 5. Vertical profiles of $^{129}$I concentrations at locations shown in (D) of selected GEOVIDE stations and of those reported in nearby locations by earlier studies. (A) $^{129}$In the Labrador and Irminger Seas, red profiles represent data from this work (2014). Data from 1993 was reported southwest of GEOVIDE station 69 by Edmonds et al. (2001). Data from 1997 to 2013 were reported at Station 17 of the AR7W line: for 1997, 1999 and 2001 by Smith et al. (2005); for 2003, 2005 and 2009 by Orre et al. (2010); and for 2012 and 2013 by Smith et al. (2016). (B) In the Icelandic Basin, green profiles show data from GEOVIDE stations 32 and 38 in 2014, while black profiles represent data from 1993 reported by Edmonds et al. (2001). (C) In the West European Basin, blue profiles represent data from GEOVIDE**

**stations 1 to 26, while data from 2012 in the Porcupine Abyssal Plain (PAP) was reported by Vivo-Vilches et al. (2018). Water masses found during GEOVIDE cruise have been summarized and represented in same colour as the $^{129}$I concentration profiles. Acronyms are defined in Table 1.'.**

**We hope the interpretation of Figure 5 (former figure 4) is now clear.**

*11) Time series data in Fig.4A is also not good to undestand temporal changed of $^{129}$I concentration.*
**Please see answer to comment 10.**

*12) In general, all figure captions did not contain enough information about meaning of each color and each mark. Please state more precisely.*
**All the figure captions have been revised and completed to make figure interpretation more straightforward in pages 38-42.**

*End of comments.*

[revised manuscript text omitted]

* * *
Margin annotations:

Núria Casacuberta 14/8/2018 11:38

María Isabel García Ib…, 13/8/2018 19:19

Microsoft Office User 8/8/2018 9:46

María Isabel García Ib…, 13/8/2018 19:19

Microsoft Office User 8/8/2018 9:51

María Isabel García Ib…, 13/8/2018 19:19

Microsoft Office User 22/8/2018 11:50

Microsoft Office User 7/8/2018 11:12

Microsoft Office User 22/8/2018 11:52

Núria Casacuberta 14/8/2018 11:43

Microsoft Office User 8/8/2018 13:18

María Isabel García Ib…, 13/8/2018 19:20

Microsoft Office User 8/8/2018 13:22

Microsoft Office User 8/8/2018 13:19
**Bajado [11]:** (Edmonds et al., 2001; Orr … [139]

Microsoft Office User 8/8/2018 13:19
**Movido (inserción) [11]**

Núria Casacuberta 14/8/2018 11:46

Microsoft Office User 8/8/2018 13:33

María Isabel García Ib…, 13/8/2018 19:21

Microsoft Office User 7/8/2018 10:18

5A) and central Labrador Sea (station 69, red circles in Figure 5A) in 2014 is the surface [129]I peak in the latter one. Considering the 30 m deep freshwater surface layer observed between station 69 and Greenland (not shown here), we suggest that waters carried by the West Greenland Current (continuation of the EGC) may have separated from the main western boundary transport and entered the Central Labrador Sea (Cuny and Rhines, 2002). This may also explain the peak in [236]U/[238]U ratios observed at the same location (Figure S1).

A similar assessment of [129]I concentrations is now possible for the water column over the Reykjanes Ridge (station 38) and the Icelandic Basin (station 32) (Figure 5B), which were first studied in 1993 (Edmonds et al., 2001). The [129]I concentrations in the water column are 5 - 7 times higher in 2014 than in 1993. The most pronounced increase occurs in the upper 1000 m filled by SPMWs and in the deep Icelandic Basin dominated by ISOW. This novel result shows that the [129]I tracer could potentially be used to trace the transformation of ENACWs into SPMWs and the evolution of ISOW. The depth profiles of [129]I concentration measured in the WEB in 2014 (particularly station 21) resemble the one sampled at the Porcupine Abyssal Plain (PAP) in 2012 by Vivo-Vilches et al. (2018) (Figure 5C). The [129]I distribution in the upper 1000 m at PAP is very similar to station 21 located 365 km to the southwest, while below that depth [129]I concentrations are about $2.5 \times 10^7$ at/kg higher in the PAP. These results suggest a similar water mass composition for that region, yet the offset in deep [129]I concentrations would support the hypothesis that effluents from the nearby Sellafield and/or La Hague NRPs may enter directly into the SPNA without previous circulation in the Nordic Seas (see section 3.5.1).

**3.5 Tracing water mass circulation in the SPNA using [129]I and [236]U**

We use the above information on the distribution, sources and time evolution of [129]I and [236]U to investigate the circulation of nuclear reprocessing effluents and in return, provide more insight on composition, spreading pathways and transport time scales of water masses in the SPNA.

[revised manuscript text omitted]

Microsoft Office User 10/8/2018 10:03

[revised manuscript text omitted]